# TAMING IMPERFECT PROCESS VERIFIERS: A SAMPLING PERSPECTIVE ON BACKTRACKING

**Dhruv Rohatgi**
MIT
drohatgi@mit.edu

**Abhishek Shetty**
MIT
shetty@mit.edu

**Donya Saless**
TTIC
donya@ttic.edu

**Yuchen Li**
Google Research
yuchenl4@alumni.cmu.edu

**Ankur Moitra**
MIT
moitra@mit.edu

**Andrej Risteski**
CMU
aristesk@andrew.cmu.edu

**Dylan J. Foster**
Microsoft Research
dylanfoster@microsoft.com

## ABSTRACT

Test-time algorithms that combine the *generative* power of language models with *process verifiers* that assess the quality of partial generations offer a promising lever for eliciting new reasoning capabilities, but the algorithmic design space and computational scaling properties of such approaches are still opaque, and their benefits are far from apparent when one accounts for the cost of learning a high-quality verifier. Our starting point is the observation that seemingly benign errors in a learned verifier can lead to catastrophic failures for standard decoding techniques due to *error amplification* during the course of generation. We then ask: can this be improved with more sophisticated decoding strategies?

We introduce a new process-guided test-time sampling algorithm, VGB, which uses theoretically grounded *backtracking* to achieve *provably* better robustness to verifier errors. VGB interprets autoregressive generation as a random walk on a tree of partial generations, with transition probabilities guided by the process verifier and base model; crucially, backtracking occurs probabilistically. This process generalizes the seminal *Sinclair-Jerrum random walk* (Sinclair & Jerrum, 1989) from the literature on approximate counting and sampling in theoretical computer science, and a conceptual contribution of our work is to highlight parallels with this literature. Empirically, we demonstrate on both synthetic and real language modeling tasks that VGB outperforms baselines on a variety of metrics.

## 1 INTRODUCTION

Test-time compute provides a powerful lever for scaling and improving language models, driving substantial improvements in reasoning capabilities (Brown et al., 2024; Snell et al., 2024; Wu et al., 2024; OpenAI, 2024; DeepSeek-AI, 2025). At the heart of these advances lies a fundamental principle: combining the *generative* power of language models with *verifiers* that can evaluate and guide their outputs. Even simple approaches like best-of-$N$ sampling, where a verifier selects the highest-scoring response from multiple candidates, can yield non-trivial performance gains (Brown et al., 2024). If one has access to a *process verifier* that can assess the quality of *partial* generations, additional gains can be unlocked (Polu & Sutskever, 2020; Uesato et al., 2022; Lample et al., 2022; Lightman et al., 2024; Wang et al., 2024; 2025)—and the *space of possible algorithmic strategies* becomes considerably broader and less well-understood.

Empirical generation methods that incorporate process verifiers range from simple token-wise reweighting (Yang & Klein, 2021; Mudgal et al., 2024; Khanov et al., 2024; Rashid et al., 2024) to more complex strategies that expend additional parallel (Mudgal et al., 2024; Wang et al., 2025; Puri et al., 2025) or sequential (Botta et al., 2025) computation. Many of these strategies are implicitly trying to address the problem that **process verifiers are imperfect**; often, they are learned from data (Yang & Klein, 2021; Lightman et al., 2024; Wang et al., 2025), and therefore seem subject to the *curse of horizon*—a folklore principle in many subcommunities of machine learning, including imitation learning (Ross & Bagnell, 2010), model-based reinforcement learning (Janner et al., 2019), multi-hop reasoning (Jiang et al., 2022), dynamical systems (Somalwar et al., 2025), PDEs (Lippe

et al., 2023; Molinaro et al., 2024), and language modelling (Bengio et al., 2015; Cundy & Ermon, 2024), that **long horizons tend to amplify learning errors**. Understanding the extent to which different strategies mitigate this issue—and designing better strategies—requires making this problem *explicit*: in what ways are learned process verifiers imperfect? And when—if ever—can amplification of these imperfections during generation be algorithmically avoided?

**In this paper, we propose a concrete model for imperfect process verifiers, in which we show error amplification *can be algorithmically mitigated*.** To do so, we connect the above problem with the classical algorithmic toolkit for sampling—specifically, a Markov chain Monte Carlo (MCMC) technique from theoretical computer science that leverages *approximate counting* oracles to do approximate sampling (Sinclair & Jerrum, 1989). This machinery lets us implement the empirical heuristic of *backtracking* (Botta et al., 2025; Yang et al., 2025; von Rütte et al., 2025)—i.e. occasionally "erasing" generated tokens—in a mathematically justified manner, and to establish rigorous guarantees for guiding generation with an imperfect process verifier. Broadly, we believe that perspectives from classical theory on design and analysis of Markov chains may bear additional fruits for language model reasoning, and we hope our paper will stimulate further work to connect these areas. See Appendix A for additional related work.

## 1.1 THE PROMISE: TEST-TIME ALIGNMENT WITH VALUE FUNCTIONS

Let $\pi_{\mathsf{ref}} : \mathcal{X} \to \Delta(\mathcal{Y})$ be a pre-trained language model (or *base policy*) that maps a prompt $x$ to a distribution over responses $y = (y_1, \ldots, y_H) \in \mathcal{Y} := \mathcal{A}^H$, where each $y_h$ is an *action* (either a single token, or more generally, a chunk of tokens). A basic goal of both post-training and test-time interventions is to "align" $\pi_{\mathsf{ref}}$ according to a *reward function* $r^\star : \mathcal{X} \times \mathcal{Y} \to [0, R_{\mathsf{max}}]$, while maintaining the capabilities of the base policy. In post-training pipelines, this task is typically accomplished by learning the policy $\pi^\star$ that maximizes KL-regularized reward (Ziegler et al., 2019), which for a temperature parameter $\beta > 0$ and a fixed prompt $x$ is defined as

$$J_\beta(\pi; x) := \mathop{\mathbb{E}}_{y \sim \pi(\cdot|x)} [r^\star(x, y)] - \beta \cdot D_{\mathsf{KL}}(\pi(\cdot \mid x) \| \pi_{\mathsf{ref}}(\cdot \mid x)). \tag{1}$$

Given prompt $x$, the analogous test-time problem is to sample from $\pi^\star(\cdot \mid x)$, which has the form

$$\pi^\star(y \mid x) \propto \pi_{\mathsf{ref}}(y \mid x) \cdot \tau(x, y) \tag{2}$$

where $\tau(x, y) := \exp(\beta^{-1} r^\star(x, y))$ (Korbak et al., 2022; Geuter et al., 2025). For a general tilt function $\tau : \mathcal{X} \times \mathcal{Y} \to \mathbb{R}_{\geq 0}$, we call this problem *test-time alignment*: given sample access to $\pi_{\mathsf{ref}}$, query access to $\tau$, and a fixed prompt $x$, how do we sample from the tilted distribution $\pi^\star(\cdot \mid x)$ defined via Eq. (2)? This central algorithmic problem subsumes many natural tasks like *constrained generation* (Scholak et al., 2021), *posterior inference* (Zhao et al., 2024a; Puri et al., 2025), and *test-time reinforcement learning* (Foster et al., 2025). Unfortunately, since the response space $\mathcal{Y}$ is exponentially large, it can be computationally intractable without further information (see e.g. Proposition G.5 or Botta et al. (2025)).

**True value functions mitigate computational intractability.** While many formalizations of process verifiers have been explored in the literature (Lightman et al., 2024; Wang et al., 2024; 2025), arguably the most principled of these are *value functions*. The (ground truth) value of a partial response $y_{1:h}$ for prompt $x$ is the conditional expectation[1]

$$V^\star_{\mathsf{tilt}}(x, y_{1:h}) := \mathbb{E}^{\pi_{\mathsf{ref}}}[\tau(x, y_{1:H}) \mid y_{1:h}]. \tag{3}$$

In the KL-regularized setting, $V^\star_{\mathsf{tilt}}$ is closely related to the *regularized value function* $Q^\star_\beta$ (Zhou et al., 2025). In general, $V^\star_{\mathsf{tilt}}$ enables exactly computing the next-action conditional probabilities of $\pi^\star$ (Lemma F.5)—and, thus, efficiently sampling from $\pi^\star$ via autoregressive generation:

$$\pi^\star(y_h \mid x, y_{1:h-1}) \propto \pi_{\mathsf{ref}}(y_h \mid x, y_{1:h-1}) \cdot V^\star_{\mathsf{tilt}}(x, y_{1:h}).$$

We call this method *action-level rejection sampling* with $V^\star_{\mathsf{tilt}}$; see also Yang & Klein (2021).

---

[1]We use the notation $V^\star_{\mathsf{tilt}}$ to avoid confusion with the optimal value function for the autoregressive MDP, which in this setting would be $V^\star(y_{1:h}) := \max_{y_{h+1:H} \in \mathcal{A}^{H-h}} \tau(x, y_{1:H})$.

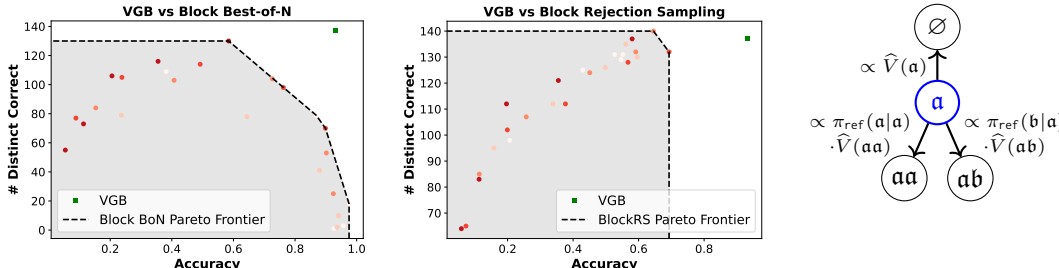

Figure 1: **Left/middle:** Accuracy (**x**) and diversity (**y**) of VGB vs. Block Best-of-$N$ and Block Rejection Sampling on Dyck grammar task (Section 5.2) with pre-trained base model and trained value function. Each circle is a baseline with specific hyperparameters (block length $\in \{1, 2, 4, 8, 16\}$ and # of candidates $\in \{2, 4, 8, 16, 32\}$); darker red indicates larger block length. **Right:** Snippet of the generation tree on which VGB walks, with transition probabilities from "$\mathfrak{a}$" (self-loop not shown).

## 1.2 THE CHALLENGE: A TRUE VALUE FUNCTION IS HARD TO FIND

The preceding discussion paints an optimistic picture: given the value function $V_{\mathtt{tilt}}^\star$, test-time alignment can be efficiently and exactly solved via action-level rejection sampling. But there is no free lunch: typically, the task of evaluating $V_{\mathtt{tilt}}^\star$ is itself computationally intractable. In practice, it is more common to use some heuristic progress metric (Khanov et al., 2024; Loula et al., 2025)—or to *learn an approximation* to $V_{\mathtt{tilt}}^\star$. A popular recent approach is to fit an approximate value function $\widehat{V}$ through regression on Monte-Carlo rollouts (Yang & Klein, 2021; Mudgal et al., 2024; Chakraborty et al., 2024; Setlur et al., 2025; Wang et al., 2025), solving some variant of the program

$$\widehat{V} := \arg\min_{V \in \mathcal{V}} \widehat{\mathbb{E}}\Big[\big(V(x, y_{1:h}) - \tau(x, y_{1:H})\big)^2\Big], \tag{4}$$

which theoretically converges to $V_{\mathtt{tilt}}^\star$ in the well-specified large-sample limit, but in practice will incur errors due to misspecification, optimization bottlenecks, and finite samples. How do we design algorithms that mitigate *amplification* of these errors across long horizons?

## 1.3 OUR CONTRIBUTIONS

**Pitfalls of naive action-level sampling (Section 3).** As a starting point, we theoretically demonstrate that when all we have is an approximate value function, **action-level rejection sampling is no longer the right alignment algorithm**: even in the presence of evidently benign errors in the approximate value function $\widehat{V}$ (Examples 3.1 and 3.2), action-level rejection sampling with $\widehat{V}$ *amplifies these errors*, degrading catastrophically with long generations. Other common process-guided algorithms such as beam search (Wang et al., 2025) are also insufficient even without errors (Appendix C).

**A new value-guided alignment strategy: VGB (Section 4).** We introduce VGB, a simple one-line modification to action-level rejection sampling that **provably mitigates error amplification** for a broad class of value function errors—entirely avoiding degradation in the generation length—through *backtracking*. VGB interprets the space of possible responses to a prompt as a tree (with internal nodes as partial responses) and casts generation as a random walk on the tree (Figure 1), which stochastically adds actions or backtracks, guided by the value function and base model—and is designed so that the *stationary distribution* of the random walk is exactly proportional to $\pi^\star$ at the leaves of the tree, even with imperfect value estimates at internal nodes.

On a technical level, VGB can be viewed as a generalization of the seminal *Sinclair-Jerrum random walk* (Sinclair & Jerrum, 1989), a tool originally developed in the approximate sampling community of theoretical computer science. A core conceptual contribution of our work is to highlight parallels between test-time alignment and approximate sampling and counting, opening the door for further transfer of ideas. Beyond showing that the Sinclair-Jerrum walk adapts to test-time alignment, a key technical contribution of our work is to establish correctness of the algorithm in the presence of average-case errors—more akin to standard assumptions from statistical learning theory—using modern techniques such as local mixing (Liu et al., 2024b).

**Empirical results (Section 5).** We evaluate VGB with a trained value function $\widehat{V}$ on Dyck grammar generation, Python test case generation, and several instructive synthetic tasks. We find that **VGB**

**outperforms naive sampling strategies on a variety of axes measuring distributional accuracy**. Also, in the classical constrained text generation setting where *there is no trained value function*, VGB produces more coherent generations than the predominant algorithm (Scholak et al., 2021). Our empirical results corroborate our theoretical thesis that VGB mitigates error amplification.

## 2    PRELIMINARIES

**General framework: Test-time alignment.**    Fix sets $\mathcal{X}$ and $\mathcal{Y} := \mathcal{A}^H$. Let $\pi_{\mathsf{ref}} : \mathcal{X} \to \Delta(\mathcal{Y})$ be a base model. Let $\tau : \mathcal{X} \times \mathcal{Y} \to \mathbb{R}_{\geq 0}$ be a tilt function, and let $\widehat{V} : \mathcal{X} \times \mathcal{A}^\star \to \mathbb{R}_{\geq 0}$ be an approximate value function (in subsequent sections, we will make assumptions about how $\widehat{V}$ relates to the true value function $V^\star_{\mathsf{tilt}}$ defined in Eq. (3)). We assume that, for any $x \in \mathcal{X}$, $h \in [H]$, and $y_{1:h} \in \mathcal{A}^h$, we can (1) sample from the conditional distribution $\pi_{\mathsf{ref}}(\cdot \mid x, y_{1:h-1})$, (2) query the density $\pi_{\mathsf{ref}}(y_h \mid x, y_{1:h-1})$, and (3) query $\widehat{V}(x, y_{1:h})$. We discuss both the setting where $\tau$ is known (in which case we set $\widehat{V}(x, y_{1:H}) := \tau(x, y_{1:H})$) and the setting where $\tau$ is unknown. Given $x \in \mathcal{X}$, our primary goal is to sample from a distribution close to $\pi^\star(\cdot \mid x)$ as defined in Eq. (2).

**Special case: KL-regularized optimization.**    Suppose $\tau(x, y) \propto \exp(\beta^{-1} r^\star(x, y))$ for some reward function $r^\star : \mathcal{X} \times \mathcal{Y} \to [0, R_{\mathsf{max}}]$ and temperature parameter $\beta > 0$. Given $x \in \mathcal{X}$, the goal of KL-regularized optimization is to sample from a distribution $\pi(\cdot \mid x)$ with low *KL-regularized regret* $J_\beta(\pi^\star; x) - J_\beta(\pi; x)$ (see Eq. (1)). Since $J_\beta(\pi^\star; x) - J_\beta(\pi; x) = \beta D_{\mathsf{KL}}(\pi(\cdot \mid x) \,\|\, \pi^\star(\cdot \mid x))$ (Lemma F.4), this goal is equivalent to approximate sampling.    In this setting, we have that $V^\star_{\mathsf{tilt}}(x, y_{1:h}) = \exp(\beta^{-1} Q^\star_\beta(x, y_{1:h}))$ where $Q^\star_\beta$ is the *regularized value function* (see Appendix F.2).

**Special case: Constrained sampling.**    In this particularly common setting, $\tau(x, y) = r^\star(x, y)$ for some binary-valued reward function $r^\star : \mathcal{X} \times \mathcal{Y} \to \{0, 1\}$, so $\pi^\star$ is a conditional distribution (Yang & Klein, 2021; Scholak et al., 2021; Lew et al., 2023; Wang et al., 2025).

### 2.1    BASELINE ALGORITHMS FOR TEST-TIME ALIGNMENT

**Outcome-level rejection sampling with $\tau$.**    Given prompt $x$, the algorithm OutcomeLevelRS repeatedly draws $y_{1:H} \sim \pi_{\mathsf{ref}}(\cdot \mid x)$ and accepts with probability proportional to $\tau(x, y_{1:H})$.

**Action-level rejection sampling with $\widehat{V}$.**    Given prompt $x$, the algorithm ActionLevelRS autoregressively samples $y_1, y_2, \ldots, y_H$: for each $h \in [H]$, after sampling $y_{1:h-1}$, it uses rejection sampling to sample $y_h$ from the distribution defined by $\mu_h(y_h) \propto \pi_{\mathsf{ref}}(y_h \mid x, y_{1:h-1}) \cdot \widehat{V}(x, y_{1:h})$.

See Appendix G for formal pseudocode and analyses. OutcomeLevelRS is an exact sampler for $\pi^\star$ (and doesn't need $\widehat{V}$)[2] but its runtime scales with the *sequence-level coverage coefficient* $\mathcal{C}_{\mathsf{seq}}(x) := \max_{y \in \mathcal{Y}} \pi^\star(y \mid x)/\pi_{\mathsf{ref}}(y \mid x)$. When $\widehat{V} = V^\star_{\mathsf{tilt}}$, ActionLevelRS is also exact, and has runtime $\widetilde{O}(\mathcal{C}_{\mathsf{act}}(x)H)$, where $\mathcal{C}_{\mathsf{act}}(x) := \max_{h, y_{1:h}} \pi^\star(y_h \mid x, y_{1:h-1})/\pi_{\mathsf{ref}}(y_h \mid x, y_{1:h-1})$ is the *action-level coverage coefficient*.[3] We always have $\mathcal{C}_{\mathsf{act}}(x) \leq \mathcal{C}_{\mathsf{seq}}(x)$, but one can have $\mathcal{C}_{\mathsf{seq}}(x) \geq 2^H$ even if $\mathcal{C}_{\mathsf{act}}(x) \leq 2$, so ActionLevelRS can be exponentially more efficient than OutcomeLevelRS.[4]

**Remark 2.1** (Sampling vs reward maximization). *As discussed in Section 1, we cast guided generation as a* sampling *problem, not a pure search/reward maximization problem, and thus largely consider sampling baselines. This framing is motivated by similar considerations in post-training (Ziegler et al., 2019), applications to constrained sampling (Scholak et al., 2021; Yang & Klein, 2021; Botta et al., 2025), and empirical evidence that sampling is a more robust formalism for exploiting imperfect process verifiers (Puri et al., 2025).*

## 3    PITFALLS OF ACTION-LEVEL SAMPLING

Unfortunately, the computational benefits of ActionLevelRS may not be realized in the presence of estimation errors. Suppose that we have access to an approximate value function $\widehat{V}$ that is accurate up to some multiplicative error $1 + \varepsilon_V$: $\frac{\widehat{V}(x, y_{1:h})}{V^\star_{\mathsf{tilt}}(x, y_{1:h})} \in [1/(1 + \varepsilon_V), 1 + \varepsilon_V]$.    The following

---

[2]More precisely, it can achieve approximation error $\varepsilon$ in time $O(\log(1/\varepsilon))$—see Appendix G.

[3]When $|\mathcal{A}|$ is small, one can explicitly sample from $\mu_h$ in time $|\mathcal{A}|$, for overall time complexity of $O(|\mathcal{A}|H)$.

[4]See Setlur et al. (2025); Botta et al. (2025) for similar observations.

---

**Algorithm 1** VGB: Value-Guided Sampling with Stochastic Backtracking (illustration in Figure 21)

---

**Input:** base model $\pi_{\mathsf{ref}}$; appx. value function $\widehat{V}$, prompt $x \in \mathcal{X}$, horizon $H \in \mathbb{N}$, step count $T \in \mathbb{N}$.

1: If outcome-level reward/tilt is available, set $\widehat{V}(x, y_{1:H}) := \tau(x, y_{1:H})$.
2: Initialize $y^{(0)} := \varnothing$.
3: **for** $0 \leq t < T$ **do**
4:   Set $h := |y^{(t)}|$, and define $p^{(t)}$ as the distribution over neighborhood $\mathcal{N}(y_{1:h}^{(t)})$ of $y_{1:h}^{(t)}$ where

$$p^{(t)}(y_{1:h-1}^{(t)}) \propto \begin{cases} \widehat{V}(x, y_{1:h}^{(t)}) & \text{if } h > 0, \\ 0 & \text{if } h = 0, \end{cases}$$

*Backtracking probability*

and for each $y_{h+1} \in \mathcal{A}$,

$$p^{(t)}(y_{1:h}^{(t)}, y_{h+1}) \propto \begin{cases} \pi_{\mathsf{ref}}(y_{h+1} \mid x, y_{1:h}^{(t)}) \, \widehat{V}(x, y_{1:h}^{(t)}, y_{h+1}) & \text{if } h < H, \\ 0 & \text{if } h = H. \end{cases}$$

5:   With probability $1/2$, set $y^{(t+1)} := y^{(t)}$, else sample $y^{(t+1)} \sim p^{(t)}$.   `// See Remark 4.1.`
6: **return** $y^{(T)}$ (Thm. 4.1) or $y^{(i)}$ for $i \sim \mathsf{Unif}([T])$ (Thm. 4.2).   `// For analysis only.`

---

didactic example shows that even for small $\varepsilon_V = o(1)$, `ActionLevelRS` can fail entirely (unless $\varepsilon_V$ is inverse-polynomial in the sequence length $H$). See Appendix H.1 for more details.[5]

**Example 3.1** (Failure of `ActionLevelRS` under perturbation). *Fix $\varepsilon_V \in (0,1)$ and $H \in \mathbb{N}$. Let $\mathcal{X} := \{\bot\}$ (henceforth omitted from notation) and action space $\mathcal{A} := \{\mathfrak{a}, \mathfrak{b}, \mathfrak{c}\}$. Let $\pi_{\mathsf{ref}} := \mathsf{Unif}(\{\mathfrak{a}, \mathfrak{b}, \mathfrak{c}\}^H)$ and $\tau(y_{1:H}) := \mathbb{I}\{\mathfrak{c} \notin y_{1:H}\}$. Then $V_{\mathsf{tilt}}^{\star}(y_{1:h}) = (2/3)^{H-h}\mathbb{I}[\mathfrak{c} \notin y_{1:h}]$. Let us define $\widehat{V}(y_{1:h}) := (1 + \varepsilon_V)V_{\mathsf{tilt}}^{\star}(y_{1:h})\mathbb{I}\{y_h = \mathfrak{a}\} + V_{\mathsf{tilt}}^{\star}(y_{1:h})\mathbb{I}\{y_h \neq \mathfrak{a}\}$. Then the output distribution $\widehat{\pi}$ of `ActionLevelRS` satisfies $D_{\mathsf{TV}}(\widehat{\pi}, \pi^{\star}) \geq \Omega(\min(\varepsilon_V \sqrt{H}, 1))$.* ◁

Example 3.1 shows seemingly benign perturbations to the estimated value function $\widehat{V}$ can compound during the execution of `ActionLevelRS`. While it is tempting to imagine that this might be a fundamental limitation (i.e., the approximate value simply doesn't contain enough information to meaningfully guide the sampling process), we will see in Section 4 that this is not the case.

We give a second example in the language of the KL-regularized setting, which demonstrates that `ActionLevelRS` can suffer $\Omega(1)$ KL-regularized regret even when there are no "perturbations" in $\widehat{V}$—but it is simply *delayed* one step compared to the truth. See Appendix H.1 for more details.

**Example 3.2** (Failure of `ActionLevelRS` under delay). *Let $\mathcal{X} := \{\bot\}$ (omitted from notation) and $\mathcal{A} := \{0, 1\}$. Let $\pi_{\mathsf{ref}} := \mathsf{Unif}(\mathcal{A}^H)$, $\beta := 1/H$, and $r^{\star}(y_{1:H}) := \frac{1}{H}\sum_{h=1}^{H} y_h$, so that $\pi^{\star}(y_{1:H}) \propto \exp(\sum_{h=1}^{H} y_h)$ and, via Eq. (3), $V_{\mathsf{tilt}}^{\star}(y_{1:h}) = \left(\frac{1+e}{2}\right)^{H-h}\exp(\sum_{k=1}^{h} y_k)$. We define $\widehat{V}$ by delaying the true value function by one step: $\widehat{V}(y_{1:h}) := V_{\mathsf{tilt}}^{\star}(y_{1:h-1})$, which satisfies $\varepsilon_V := 1$. Since $\widehat{V}(y_{1:h})$ does not depend on $y_h$, `ActionLevelRS` samples from the uniform distribution $\widehat{\pi} = \pi_{\mathsf{ref}}$, which has $J_\beta(\pi^{\star}) - J_\beta(\widehat{\pi}) = \frac{1}{H}D_{\mathsf{KL}}(\widehat{\pi} \| \pi^{\star}) \geq \frac{1}{10}$.* ◁

Example 3.2 is a negative result for `ActionLevelRS`, but also hints at an algorithmic intervention: $\widehat{V}(x, y_{1:h})$ gives no guidance in selecting $y_h$, but *can* help detect whether the *previous* action $y_{h-1}$ was good or bad. Can we improve `ActionLevelRS` by propagating information backwards?

## 4 TAMING IMPERFECT PROCESS VERIFIERS: STOCHASTIC BACKTRACKING

We now present a value-guided sampling algorithm, VGB, that provably mitigates error amplification through a principled *stochastic backtracking* strategy. We first describe the algorithm and motivation (Section 4.1), then provide theoretical guarantees (Sections 4.2 and 4.3).

## 4.1 MAIN ALGORITHM: VGB

Our main algorithm, VGB, is displayed in Algorithm 1. To explain the algorithm, we view guided sampling algorithms as implicitly traversing the exponentially-large tree $\mathcal{T}$ with node set $\bigcup_{h=0}^{H} \mathcal{A}^h$, where $y_{1:h-1}$ is the parent of $y_{1:h}$. ActionLevelRS implements a random walk from the root $\varnothing$ to a leaf $y_{1:H}$, which only goes *down* the tree. As we saw in Example 3.2, such strategies seem limited in their ability to correct for systematic errors in the value function estimates.

**VGB as a one-line change to ActionLevelRS.** VGB addresses the limitations of ActionLevelRS by augmenting the random walk with the ability to *backtrack up the tree*. Formally, let $\mathcal{N}(y_{1:h})$ denote the neighborhood of $y_{1:h}$ in $\mathcal{T}$, which contains its parent $y_{1:h-1}$ and all its children $\{y_{1:h+1}\}_{y_{h+1} \in \mathcal{A}}$.

VGB now proceeds as follows. At each step $t$, given the current node $y_{1:h}^{(t)}$, we first (Line 4) define a probability distribution $p^{(t)}$ over the neighborhood $\mathcal{N}(y_{1:h}^{(t)})$. The probability of selecting a child node $(y_{1:h}^{(t)}, y_{h+1})$ is proportional to $\pi_{\text{ref}}(y_{h+1} \mid x, y_{1:h}^{(t)}) \widehat{V}(x, y_{1:h}^{(t)}, y_{h+1})$—just as in ActionLevelRS—while the probability of selecting the parent $y_{1:h-1}^{(t)}$ is proportional to $\widehat{V}(x, y_{1:h}^{(t)})$. The random walk proceeds by sampling a new node $y^{(t+1)}$ from $p^{(t)}$ with probability $1/2$, and staying at the current node otherwise (setting $y^{(t+1)} \leftarrow y^{(t)}$).[6] After $T$ steps, in the version of VGB that we analyze first, it returns the final node of the random walk.[7]

**VGB as a generalization of the Sinclair-Jerrum walk.** VGB can be interpreted as a generalization of the Sinclair-Jerrum walk (Sinclair & Jerrum, 1989), which corresponds to the special case where $\pi_{\text{ref}}$ is uniform and $\tau$ is binary-valued.[8] This walk was originally used to show that approximate *sampling* reduces to approximate *counting* in self-reducible problems such as SAT, without incurring the error amplification of prior reductions (Jerrum et al., 1986). Value functions are precisely the generalization of counting oracles to our setting.

Under Assumption 4.1 on $\widehat{V}$, stated below, VGB inherits three key properties from the Sinclair-Jerrum walk, formalized in Theorem 4.1: (1) the walk rapidly mixes to a stationary distribution $\widetilde{\pi}$; (2) $\widetilde{\pi}$ puts $\Omega(1/H)$ mass on the leaves of $\mathcal{T}$; and (3) as long as exact outcome-level rewards are used (i.e. $\widehat{V}(x, y_{1:H}) = \tau(x, y_{1:H})$; see Remark 4.2), $\widetilde{\pi}$ is proportional to $\pi^\star$ at the leaves.

**Remark 4.1** (Efficient implementation for large action spaces). *When the action space $\mathcal{A}$ is small (e.g., in token-level generation), the distribution $p^{(t)}$ in Line 5 of VGB can be constructed explicitly by enumerating all actions. When the action space is large (e.g., in block-level generation), this enumeration is computationally infeasible, but we can still sample efficiently from $p^{(t)}$ using rejection sampling (in time $\widetilde{O}(\mathcal{C}_{\text{act}}(x))$—see Theorem 4.1). See Algorithm 8 for the detailed implementation.*

## 4.2 THEORETICAL GUARANTEES: UNIFORM ERROR

We present the first of our theoretical guarantees for VGB—the strongest and simplest—under the assumption that the learned value function $\widehat{V}$ satisfies a *uniform* error bound with respect to the true value function $V_{\text{tilt}}^\star$ (as discussed in Section 3), and that we have query access to the outcome-level reward $\tau$; we will relax both assumptions in the sequel. The assumption is for a fixed prompt $x \in \mathcal{X}$:

**Assumption 4.1** (Uniform bound on value errors). *We assume that the exact outcome-level reward is available (i.e., $\widehat{V}(x, y_{1:H}) = \tau(x, y_{1:H})$), and for each $1 \leq h < H$, for each $y_{1:h} \in \mathcal{A}^h$,*

$$\frac{\widehat{V}(x, y_{1:h})}{V_{\text{tilt}}^\star(x, y_{1:h})} \in [1/(1 + \varepsilon_V), 1 + \varepsilon_V]. \tag{5}$$

Under this assumption, we show that VGB rapidly converges to the target distribution $\pi^\star$. The proof closely follows the analysis of the Sinclair-Jerrum walk (Sinclair & Jerrum, 1989), which bounds the mixing time of the walk by analyzing its conductance. See Appendix I for the full proof.

---

[5]We remark that Example 3.1 also has an analogue in the KL-regularized setting; see Example H.1.

[6]This technique (staying at the current node with probability $1/2$) is called *laziness*, and is needed to ensure that the random walk has a stationary distribution. See e.g. Levin et al. (2009) for background on Markov chains.

[7]The chosen node may not be a leaf, in which case we re-run the algorithm; this can occur at most $\widetilde{O}(H)$ times.

[8]See Bakshi et al. (2024) for a related generalization, applied to quantum Gibbs state preparation.

---

**Algorithm 2** Instantiation of VGB for KL-regularized reinforcement learning

---

1: **Input:** Base model $\pi_{\mathrm{ref}}$; approximate regularized value function $\widehat{Q}$, prompt $x \in \mathcal{X}$, horizon $H \in \mathbb{N}$,
   regularization parameter $\beta > 0$, step count $T \in \mathbb{N}$.
2: If outcome-level reward is available, set $\widehat{Q}(x, y_{1:H}) = r^\star(x, y_{1:H})$.
3: Initialize $y^{(0)} := \varnothing$.
4: **for** $0 \le t < T$ **do**
5:  Set $h := |y^{(t)}|$, and define $p^{(t)}$ as the distribution over neighborhood $\mathcal{N}(y_{1:h}^{(t)})$ of $y_{1:h}^{(t)}$ where

$$p^{(t)}(y_{1:h-1}^{(t)}) \propto \begin{cases} \exp\!\big(\beta^{-1}\widehat{Q}(x, y_{1:h}^{(t)})\big) & \text{if } h > 0, \\ 0 & \text{if } h = 0, \end{cases} \qquad \textit{Backtracking} \atop \textit{probability}$$

and for each $y_{h+1} \in \mathcal{A}$,

$$p^{(t)}(y_{1:h}^{(t)}, y_{h+1}) \propto \begin{cases} \pi_{\mathrm{ref}}(y_{h+1} \mid x, y_{1:h}^{(t)}) \exp\!\big(\beta^{-1}\widehat{Q}(x, y_{1:h}^{(t)}, y_{h+1})\big) & \text{if } h < H, \\ 0 & \text{if } h = H. \end{cases}$$

6:  With probability $1/2$, set $y^{(t+1)} := y^{(t)}$, else sample $y^{(t+1)} \sim p^{(t)}$.  `// See Remark 4.1.`
7: **return** $y^{(T)}$ (Thm. 4.1) or $y^{(i)}$ for $i \sim \mathsf{Unif}([T])$ (Thm. 4.2).  `// For analysis only.`

---

**Theorem 4.1** (Main guarantee for VGB). *For any prompt $x \in \mathcal{X}$ and $\delta > 0$, under Assumption 4.1, let $\widehat{\pi}$ be the output distribution of VGB with step count*

$$T := \widetilde{O}\big(H^2 \cdot (1 + \varepsilon_V)^4 \cdot \log(\delta^{-1})\big),$$

*and let $\mathcal{E}_{\mathsf{leaf}}$ be the event that $y \sim \widehat{\pi}$ is a leaf node. Then $\mathbb{P}[\mathcal{E}_{\mathsf{leaf}}] \ge 1/(8(1 + \varepsilon_V)H)$, and the error conditioned on this event is $D_{\mathsf{TV}}(\widehat{\pi}|_{\mathcal{E}_{\mathsf{leaf}}}, \pi^\star) \le \delta$. The total runtime (including evaluations of $\widehat{V}$ and queries/conditional generations of $\pi_{\mathrm{ref}}$) is bounded by $O(|\mathcal{A}| \cdot T)$ in the small-action case and $\widetilde{O}(\mathcal{C}_{\mathsf{act}}(x)) \cdot T$ in the large-action case.*[9]

In the KL-regularized setting, we achieve an analogous bound on regret—namely, $J_\beta(\pi^\star; x) - J_\beta(\widehat{\pi}|_{\mathcal{E}_{\mathsf{leaf}}}; x) \le \beta\delta$; see Corollary I.1 for the full statement. While these guarantees are conditioned on the event $\mathcal{E}_{\mathsf{leaf}}$, we can ensure that this occurs with high probability by simply rerunning the algorithm $\widetilde{O}(H)$ times. Compared to ActionLevelRS, **VGB avoids the error amplification demonstrated in Examples 3.1 and 3.2**, because these examples satisfy $\varepsilon_V = O(1)$. Compared to OutcomeLevelRS, VGB is much more efficient: it avoids dependence on the sequence-level coverage coefficient $\mathcal{C}_{\mathsf{seq}}(x)$, which should be thought of as potentially exponential in $H$.

To be clear, there is room for improvement, as the overall time required for VGB to output a leaf node is $\widetilde{O}(H^3)$.

**Remark 4.2** (Access to outcome-level tilt $\tau$). *When VGB has exact access to $\tau$, its output distribution (conditioned on $\mathcal{E}_{\mathsf{leaf}}$) converges to $\pi^\star$ as $\delta \to 0$, even for fixed value error $\varepsilon_V$. Without such access, convergence to $\pi^\star$ is impossible, but VGB still provably avoids error amplification, as shown below.*

### 4.3 GUARANTEES UNDER WEAKER AVERAGE-CASE ERROR

Theorem 4.1 shows that VGB can significantly improve over naive sampling in the presence of value function errors, but the result requires the uniform error condition of Assumption 4.1. Such worst-case bounds may not hold for value functions derived from standard learning techniques like Monte Carlo estimation. To address this, we present a more refined analysis of VGB under a weaker, average-case error assumption. We also drop the assumption that $\tau$ is known.

**Assumption 4.2** (Average-case bound on value errors). *We assume that for each $h \in [H]$,*

$$\mathbb{E}_{y_{1:h} \sim \pi^\star(\cdot|x)}\left[\frac{\widehat{V}(x, y_{1:h})}{V_{\mathtt{tilt}}^\star(x, y_{1:h})}\right] \le 1 + \varepsilon_V, \qquad \mathbb{E}_{y_{1:h} \sim \pi^\star(\cdot|x)}\left[\frac{V_{\mathtt{tilt}}^\star(x, y_{1:h})}{\widehat{V}(x, y_{1:h})}\right] \le 1 + \varepsilon_V. \qquad (6)$$

---

[9]This case requires (approximate) knowledge of $\mathcal{C}_{\mathsf{act}}(x)$ and $1 + \varepsilon_V$; see Remark I.1.

In the KL-regularized setting, this condition controls the moment generating function under $\pi^\star$ of the error $Q_\beta^\star(x, y_{1:h}) - \widehat{Q}(x, y_{1:h})$, where $\widehat{Q}(x, y_{1:h}) := \beta \log \widehat{V}(x, y_{1:h})$. While still requiring control over tail behavior, this significantly relaxes Assumption 4.1, which corresponds to a uniform bound on that error. Similar conditions have been studied in Aminian et al. (2025); Yang & Wibisono (2022).

Under this weaker assumption, VGB still provides strong guarantees, though of a different nature than Theorem 4.1. Instead of a direct bound on suboptimality, we show VGB provides good *coverage* (Huang et al., 2025b; Foster et al., 2025) of typical responses from the target policy $\pi^\star$:

**Theorem 4.2.** *For any prompt $x \in \mathcal{X}$ and $\delta > 0$, under Assumption 4.2, the runtime of* VGB *with step count $T := \widetilde{O}\big(H^5 \cdot (1 + \varepsilon_V)^6 \cdot \delta^{-4}\big)$ is $O(T \cdot |\mathcal{A}|)$,[10] and the output distribution $\widehat{\pi}$ satisfies*

$$\Pr_{y_{1:H} \sim \pi^\star(x)} \left[ \frac{\pi^\star(y_{1:H} \mid x)}{\widehat{\pi}(y_{1:H})} > \frac{48H(1 + \varepsilon_V)^2}{\delta} \right] \le \delta. \tag{7}$$

That is, with high probability over $y_{1:H} \sim \pi^\star(\cdot \mid x)$, the policy $\widehat{\pi}$ induced by VGB assigns $y_{1:H}$ not much less probability than $\pi^\star$. This guarantee does not directly imply low regularized regret in the KL-regularized setting; however, it *does* enable achieving low unregularized regret against $\pi^\star$, by composing VGB with standard Best-of-$N$ sampling (Cobbe et al., 2021); see Appendix B. We remark that in Examples 3.1 and 3.2, ActionLevelRS does *not* achieve Eq. (7) (see Appendix H.1).

**Overview of analysis.** The analysis of VGB under the average-case error assumption in Assumption 4.2 is considerably more challenging than in the uniform error setting. Classical techniques such as global conductance are no longer applicable. Instead, our analysis leverages modern tools for analyzing slowly mixing Markov chains, particularly the notion of *local stationarity* (Liu et al., 2024b). We view this analysis as a key technical contribution, as it bridges the average-case guarantees typical in statistical learning theory with the uniform guarantees traditionally required in the analysis of approximate counting and sampling algorithms. We defer the full proof to Appendix I.4.

**Insufficiency of alternative assumptions.** We show that several alternative assumptions (which a priori may seem more natural than Assumption 4.2) are insufficient for *any* efficient algorithm to achieve non-trivial sampling guarantees: average-case multiplicative error under $\pi_{\text{ref}}$ (Appendix D.1), additive error bounds on $\widehat{V}$ (Appendix D.2), and MSE bounds on $\widehat{Q}$ (Appendix D.3).

## 5 EMPIRICAL RESULTS

Our experiments focus on *constrained sampling*—perhaps the most common, concrete application of test-time alignment (Yang & Klein, 2021; Scholak et al., 2021; Lew et al., 2023; Wang et al., 2025)—where $\tau := r^\star$ is a binary-valued reward function. We train value functions for tasks where $\pi_{\text{ref}}$ is analytic (Section 5.1) or a pre-trained language model (Section 5.2), and demonstrate benefits of VGB on a variety of axes measuring distributional fidelity. We also show that *constrained text generation* (Scholak et al., 2021) can be interpreted as value-guided sampling with a *weak* value function—and that VGB improves coherence compared to the standard baseline for this setting (Section 5.3). These benefits largely do come at added computational cost, but we view our findings as a promising step towards understanding trade-offs between efficiency and error mitigation—and therefore a step towards reliable *long-horizon* reasoning. For all omitted experiment details, and additional results (e.g. improving the efficiency of VGB with *momentum*), see Appendix E.

**Implementation details.** The value functions are parametrized as MLPs—where the input features are either one-hot encodings of the input sequence (for tasks with small $|\mathcal{A}|$) or pooled hidden states from the pre-trained language model (for tasks with large $|\mathcal{A}|$)—and trained via regression onto Monte Carlo roll-outs (Eq. (4)). To improve wall-clock efficiency, the implementation of VGB differs in two ways from Algorithm 1. First, instead of fixing a step count $T$ we simply stop the Markov chain as soon as it reaches a leaf. Second, in settings where the alphabet is too large to efficiently enumerate, we use a heuristic version of rejection sampling to implement each transition. In experiments where the true rewards are provided at the final position, we allow ActionLevelRS to restart if all actions have reward 0. See Appendix E for pseudocode and task-by-task implementation/training details.

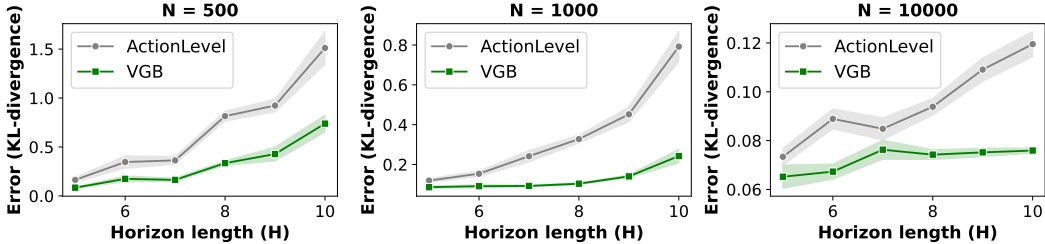

Figure 2: Estimated KL-divergence of VGB and ActionLevelRS to $\pi^\star$ in ABC task (Section 5.1) for varied horizon length $H$ and # of value function training samples $N$. We repeat the experiment 10 times for each $(H, N)$ and report the mean and standard error. See Appendix E.2 for details.

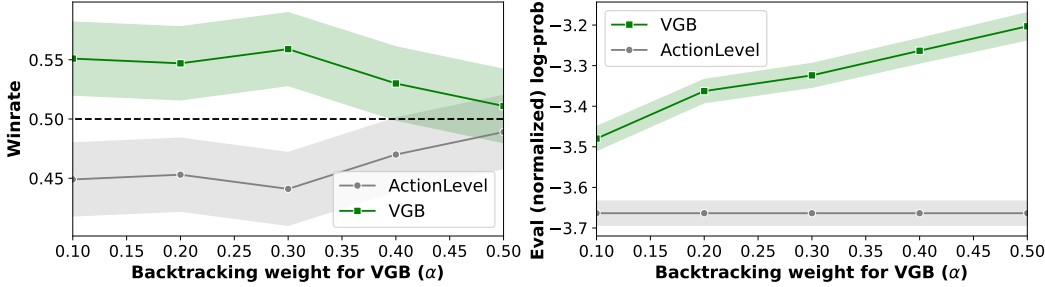

Figure 3: Comparison of VGB against ActionLevelRS for letter avoidance task (Section 5.3), with varied backtracking weight $\alpha$ for VGB. **Left:** Winrate of VGB against ActionLevelRS under pairwise comparison of responses by GPT-4o-mini (judging for coherence). **Right:** Average horizon-normalized log-probabilities evaluated by Qwen-2.5-1.5B. See Appendix E.6 for details.

## 5.1 SYNTHETIC GENERATOR WITH TRAINED VALUE FUNCTION

Our first tasks are synthetic tasks that we use to stress-test the theory; the base distribution $\pi_{\tt ref}$ is analytic rather than defined by a pre-trained language model, and the problem instances are small enough that we can easily compute relevant metrics, but the value functions are trained via Eq. (4).

**ABC task (Appendix E.2): distributional benefits.** We follow the task defined in Example 3.1, where we proved that errors in a specific $\widehat{V}$ compound with ActionLevelRS, whereas VGB provably avoids this error compounding. We show that the benefits of VGB persist when $\widehat{V}$ is trained. We implement VGB and ActionLevelRS using the true rewards at the final position, so both algorithms produce only reward-1 responses, but may have biased distributions. As shown in Figure 2, **VGB incurs lower KL-divergence to $\pi^\star$ than ActionLevelRS**, with increasing benefits for large $H$.

The following task provides a setting where *systematic* training errors arise (a la Example 3.2)—and induce an even starker separation between VGB and ActionLevelRS.

**Parity task (Appendix E.3): wall-clock time benefits.** This task instantiates common intuition that *estimating values may often be much harder at some positions than at others* (Li et al., 2024). Specifically, we design $\pi_{\tt ref}$ and $r^\star$ so that at many positions $h$, $V^\star_{\tt tilt}(y_{1:h})$ is a *parity function*—which is notoriously challenging for gradient descent to learn (Barak et al., 2022)—but many other positions "reveal" the most recent parity. Intuitively, this effectuates *delayed* feedback in the trained value function, as in Example 3.2. Indeed, on this task, **VGB requires less wall-clock time than ActionLevelRS to find a reward-1 response, with superior scaling as $H$ increases** (Figure 6).

## 5.2 LANGUAGE MODEL GENERATOR WITH TRAINED VALUE FUNCTION

We now turn to tasks where $\pi_{\tt ref}$ is a Transformer-based language model. In these tasks, while it is infeasible to empirically estimate KL-divergence as before (due to larger response space), we demonstrate that VGB has benefits over baselines according to natural subjective metrics.

---

[10]We have not sought to optimize the polynomial dependence on $H$. Also, for this result, we omit analysis of the large-$|\mathcal{A}|$ case; see Remark I.2 for discussion.

**Dyck grammar task (Appendix E.4).** The *Dyck grammar* (Schützenberger, 1963) is a classic context-free grammar consisting of balanced parenthesis strings (e.g., `[()]` is valid but `([)]` is not), and a common theoretical sandbox for language modeling (Hewitt et al., 2020; Yao et al., 2021; Liu et al., 2023b;a; Wen et al., 2023; Botta et al., 2025). Using code from Botta et al. (2025), we train (from scratch) a Transformer language model $\pi_{\texttt{ref}}$ on Dyck grammar sequences of length 32 with two bracket types. The training data is sampled from a distribution where square brackets `[` and `]` occur with probability 0.8 whereas round brackets `(` and `)` occur with probability 0.2. While $\pi_{\texttt{ref}}$ has near-perfect accuracy at completing in-distribution prefixes, it has much worse accuracy at completing out-of-distribution (OOD) prefixes, i.e. prefixes with many round brackets.

First, we fix a randomly-chosen length-16 prefix consisting of only round brackets, and train a value function for 40 epochs on 10000 completions from $\pi_{\texttt{ref}}$. We compare VGB on two metrics—*accuracy* and *diversity* (i.e. # of correct responses)—against two popular sampling algorithms: Block Best-of-$N$ (Mudgal et al., 2024) and Block Rejection Sampling (a generalization of ActionLevelRS). We sweep over block sizes and number of candidate blocks, allowing up to 32 candidate blocks—which equalizes the number of queries to $\pi_{\texttt{ref}}$ made by VGB and the baselines (Table 1). VGB achieves a **robustly non-dominated point on the Pareto frontier between accuracy and diversity** (Figure 1).

Second, on 12 OOD prefixes, we measure how well VGB and ActionLevelRS match $\pi^\star$ in distribution, using a tractable *proxy* for distributional error. True rewards are used at the final position, and the value function is trained for a varied number of epochs. After initial epochs, **VGB achieves lower $\ell_1$ error than ActionLevelRS while being more query-efficient than OutcomeLevelRS** (Figure 12).

**Code generation task (Appendix E.5).** In this task, the goal is to generate a valid test case for a randomly-named Python function that implements append and is given in the prompt (Botta et al., 2025). With base model Qwen-2.5-0.5B, a trained value function $\widehat{V}$, and true rewards at the final position, we evaluate VGB and ActionLevelRS against $\pi^\star$ on three distributional metrics: (1) the histogram of test case lengths, (2) the distribution of object types in the test cases, and (3) the number of distinct test cases generated. We find that **VGB substantially improves upon ActionLevelRS for (1), and is comparable for (2) and (3)**; see Table 3 for full results and discussion.

## 5.3 Constrained Text Generation

In our final experiment, we demonstrate that VGB can be gainfully applied **even without training a value function**. In many applications of constrained sampling, the reward function $r^\star$ itself is used as a process verifier (Shin et al., 2021; Scholak et al., 2021; Poesia et al., 2022; Lipkin et al., 2025). The dominant algorithm is *locally constrained decoding* (Scholak et al., 2021), which is precisely ActionLevelRS with $\widehat{V} \propto r^\star$. Recent works have observed that this algorithm does not sample from $\pi^\star$ (Lew et al., 2023; Park et al., 2024), and correcting this bias is an active research area.

**Letter avoidance task (Appendix E.6).** In this task, the goal is to generate an English sentence with $H = 32$ tokens that does not use the letter "e". We use base model Qwen-2.5-0.5B and approximate value function $\widehat{V}(y_{1:h}) := \alpha^{H-h} r^\star(y_{1:h})$ for varied $\alpha \in (0, 1)$. Intuitively, a good choice of $\alpha$ should roughly represent the "average probability that $\pi_{\texttt{ref}}$ errs at an average position"; the choice of $\alpha$ effectively controls the *backtracking probability* of VGB, but is irrelevant to ActionLevelRS (it is equivalent to locally constrained decoding regardless of $\alpha$). We show that across a broad range of $\alpha$, **VGB achieves superior coherence and log-probabilities to ActionLevelRS** (Figure 3).

ETHICS STATEMENT

We view our work as fundamental research on guiding Large Language Models (LLMs). While LLMs have capacity for harm, we believe that guided generation will ultimately be an important part of the toolkit for aligning LLM behaviors towards societal interests—and, therefore, that the benefits of this research outweigh potential harms.

REPRODUCIBILITY STATEMENT

We provide full proofs for all theoretical results in the appendix. We also provide our code in the supplementary material, along with extensive implementation details for all experimental results in Appendix E.

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

CONTENTS OF APPENDIX

# Part I

# Additional Discussion and Results

## A  ADDITIONAL RELATED WORK

### A.1  EMPIRICAL LITERATURE

Early empirical investigations into benefits of process verification for language models (Polu & Sutskever, 2020; Cobbe et al., 2021; Lightman et al., 2024; Uesato et al., 2022) focused on intuitive notions of intermediate progress, and explored various schemes for aggregating step-wise verification judgments into a final outcome-level score, including taking products of probabilities, majority voting (Li et al., 2023), or minimums of scores (Wang et al., 2024). A recurring theme in this line of work is the question of how to define the "steps" or "blocks" for verification, and at what level of granularity process-level feedback is most effective.

Many of these earlier works learned process rewards merely as a building block to more effectively learn outcome rewards (and used the learned rewards with reinforcement learning rather than at test time), but many recent works on language model reasoning have also directly used process reward models at test time, with beam search or other tree search algorithms (Yang et al., 2022; Snell et al., 2024; Wu et al., 2024; Wang et al., 2025). Several works have demonstrated empirical shortcomings of greedy methods such as beam search with process rewards, and suggest introducing stochasticity (Yu et al., 2025a; Puri et al., 2025) or uncertainty-awareness (Yu et al., 2025b) to robustify generation.

**Value functions as process verifiers.**  A growing body of empirical work uses learned value functions as (a specific instantiation of) process verifiers to guide language model decoding. These works can roughly be categorized as employing either guided search-based methods (primarily for mathematical reasoning or similar reward maximization tasks) (Snell et al., 2024; Setlur et al., 2025; Wang et al., 2025) or guided sampling-based methods (for alignment or constrained text generation) (Yang & Klein, 2021; Mudgal et al., 2024; Chakraborty et al., 2024; Khanov et al., 2024; Rashid et al., 2024; Li et al., 2024; Liu et al., 2025). Both lines of work typically learn the value function through Monte Carlo rollouts (a method that has also been employed in reinforcement learning pipelines (Wang et al., 2024)) and then use it within search schemes such as beam search or (block-level) best-of-$N$, or sampling schemes such as token/action-level sampling or block-level sampling.

Our work is most closely related to the sampling literature, and provides a formal theoretical model for analyzing different value-guided sampling methods, but we hope that it may provide insights for improving search as well. Empirically, we expect that VGB, by incorporating more deliberate search and backtracking while preserving the base model's diversity, could be beneficially combined with the value function learning schemes from this literature.

**Backtracking and tree search with process verifiers.**  While many search-based approaches are based on greedy decoding or beam search, several empirical works have explored more advanced backtracking or tree search schemes with process verifiers (Scialom et al., 2021; Chaffin et al., 2022; Liu et al., 2024a; Hao et al., 2023; Snell et al., 2024; Singh et al., 2025; Botta et al., 2025). These methods aim to overcome the limitations of purely left-to-right generation by allowing the model to revise earlier decisions.[11] Compared to VGB, these methods may not preserve the base model's diversity, as they are designed with pure reward maximization in mind (cf. Remark 2.1). Our work demonstrates that backtracking can be applied effectively (and in a mathematically justified manner) even for sampling.

**Controlled decoding without learned process verifiers.**  Our work connects to a long line of empirical and methodological work on constrained/controlled decoding that has developed largely in parallel with the literature discussed above, but that does not actually *learn* a process verifier (Shin et al., 2021; Scholak et al., 2021; Poesia et al., 2022; Lew et al., 2023; Ahmed et al., 2025; Gonzalez et al., 2025; Lipkin et al., 2025; Loula et al., 2025). This line of work primarily focuses on the setting where the goal is to condition a base model $\pi_{\mathrm{ref}}$ on some sequence-level constraint $r^\star(x,y) = 1$ that can also be evaluated on partial generations. These works forgo training a value function (which can be computationally expensive), and instead use the reward function $r^\star(x, y_{1:h})$ itself as a process

---

[11]We remark that other recent works explore *training* models to backtrack, which is orthogonal to the focus of this work (Qin et al., 2025).

verifier for partial generations. The dominant method in this literature is *locally constrained decoding* (Shin et al., 2021; Scholak et al., 2021) (see also Lipkin et al. (2025) for additional references), which—as we discuss in Section 5—is equivalent to `ActionLevelRS` for a natural approximate value function $\widehat{V}$.

**Sequential Monte Carlo and particle filtering.** Our work also connects to a body of literature on using Sequential Monte Carlo (SMC) and particle filtering methods for inference-time decoding under process-level guidance (Lew et al., 2023; Zhao et al., 2024a; Loula et al., 2025; Puri et al., 2025).[12] These algorithms bear some resemblance to `VGB`, in that they maintain a diverse set of candidate sequences, sampling successors based on importance weights. However, they typically do not incorporate backtracking, and lack finite-time/compute performance guarantees. It is an interesting open question whether these methods achieve comparable guarantees to `VGB`.

**Additional context: diffusion models.** Our work broadly shares some high-level motivations with a few strands of work on diffusion models. Since the earliest days of diffusion models, strategies like predictor-corrector (Song et al., 2021) were used at inference-time to tame the error accumulation at inference time, accrued from simulating the reverse SDE. Variants of these strategies have been developed also for discrete diffusions (Zhao et al., 2024b). Moreover, for the most popular type of discrete diffusion model for language—absorbing noise diffusion models—the natural inference-time sampling strategies will not revise a token once it has been set to some value. This has naturally led to work on allowing the model to backtrack in a principled manner (von Rütte et al., 2025), with promising gains in performance.

Another related area in diffusion models is sampling from tilted distributions of a distribution that we have trained a diffusion model for. Notable applications of this flavor are classifier guidance (Dhariwal & Nichol, 2021; Ho & Salimans, 2021) and inverse problems (Bruna & Han, 2024; Pokle et al., 2023). However, the conceptual problem in this area is somewhat different: namely, we have access to the denoiser for the base distribution (at any level of noise), but not for the tilted distribution—and these two cannot be easily related to each other.

### A.2 THEORETICAL LITERATURE

**Theoretical understanding of process verifiers.** On the theoretical front, our work is most closely related to Botta et al. (2025), who show provable benefits of access to an exact process verifier in a binary outcome setting without KL-regularization, a result analogous to our findings regarding action-level versus sequence-level coverage in Section 3. However, their analysis is restricted to perfect verifiers. Botta et al. (2025) also show empirical benefits of a backtracking heuristic with a trained process verifier; one key obstacle that they face is deciding *when* and *how often* to backtrack, which their method addresses by introducing several ad-hoc hyperparameters.

Foster et al. (2025) give guarantees for KL-regularized reinforcement learning in a setting where $Q_\beta^\star(x, y_{1:h}) = \langle \theta^\star, \phi(x, y_{1:h}) \rangle$ is linear in a known feature map. They show that by using deliberate exploration, it is possible to provably learn a high-quality value function, with runtime controlled by the action-level coverage coefficient $\mathcal{C}_{\mathsf{act}}$. Conceptually this result highlights that process verifiers can be learned end-to-end while maintaining the computational benefits we sketch in Section 3, but the algorithm and analysis are quite specialized to the linear setting.

Other related works include Huang et al. (2025a;b), who provide algorithms and theoretical guarantees for imperfect *outcome-level* verifiers, and Balcan et al. (2025), who study the sample complexity of learning outcome-level verifiers for chain-of-thought reasoning. Finally, we mention in passing the work of Jia et al. (2025), which shows that for general MDPs, the sequence-level coverage coefficient we consider is controlled by a state-level coverage coefficient;[13] we remark that these quantities are equivalent for language models, i.e. the state-level coverage coefficient will also typically be impractically large.

**Approximate sampling in theoretical computer science.** As discussed in Section 4, `VGB` can be interpreted as a generalization of the Sinclair-Jerrum random walk (Sinclair & Jerrum, 1989) from

---

[12]SMC is also a common method for tilting diffusion models at inference time; see e.g. Skreta et al. (2025).

[13]Jia et al. (2025) use this observation to show that for offline reinforcement learning in general MDPs where a reward $r_h$ is available at every step, learning from the sum $\sum_{h=1}^{H} r_h$ is no harder than learning from the individual rewards $(r_1, \ldots, r_H)$.

theoretical computer science. Sinclair & Jerrum (1989) developed this walk to show that for any self-reducible constraint problem such as SAT, there is an efficient complexity-theoretic reduction from approximately sampling from the uniform distribution on satisfying assignments to approximately counting the number of satisfying assignments. Crucially, the error in the approximate counter is not amplified; to the contrary, the approximate sampler can achieve arbitrary (inverse-polynomial) accuracy even when the approximate counter has polynomially-large error.

At a technical level, Sinclair and Jerrum (in the above work and Jerrum & Sinclair (1988)) developed the foundations of (now classical) theory on the mixing time of Markov chains: specifically, the method of using conductance to bound the spectral gap, which implies fast mixing. Our proof of Theorem 4.1 follows the same method; however, our proof of Theorem 4.2 requires more modern techniques such as local stationarity (Balasubramanian et al., 2022; Liu et al., 2024b), as discussed in Appendix I.4.

# B   APPROXIMATE COVERAGE ENABLES REWARD MAXIMIZATION

In Theorem 4.2, we showed that the law of the output of VGB approximately *covers* the optimal distribution under Assumption 4.2. As a result, it is competitive with the optimal distribution in terms of diversity (i.e., under Eq. (7), drawing $O(nH/\delta)$ samples from VGB will tend to cover almost as many responses as $O(n)$ samples from $\pi^\star$). Approximate coverage also enables approximate (unregularized) reward maximization. This is achieved simply by applying the Best-of-$N$ algorithm to multiple samples from VGB. In this section, we provide a formal statement of this implication for completeness.

Formally, for proposal distribution $\pi \in \Delta(\mathcal{Y})$, reward function $r^\star : \mathcal{Y} \to \mathbb{R}$, and positive integer $N$, the Best-of-$N$ algorithm draws $N$ samples $y^{(1)}, \ldots, y^{(N)} \sim \pi$ and outputs $y^{(j)}$ where $j := \arg\max_{i \in [N]} r^\star(y^{(i)})$. The following proposition shows that approximate coverage implies regret bounds for any binary-valued reward function; see e.g. Huang et al. (2025b) for more general guarantees.

**Proposition B.1** (Best-of-$N$ guarantee under approximate coverage). *Let $\pi, \pi^\star$ be two distributions over the set $\mathcal{Y}$, and let $r^\star : \mathcal{Y} \to \{0, 1\}$ be a binary-valued reward function. Suppose that $\pi, \pi^\star$ satisfy*

$$\mathbb{P}_{y \sim \pi^\star} \left[ \frac{\pi^\star(y)}{\pi(y)} \geq F(\varepsilon) \right] \leq \varepsilon \tag{8}$$

*for some function $F : \mathbb{R}_{>0} \to \mathbb{R}_{>0}$. Then for any $\varepsilon, \delta > 0$, the output $\widehat{y} \in \mathcal{Y}$ of the Best-of-$N$ algorithm with proposal distribution $\pi$, reward function $r^\star$, and parameter $N := \frac{1}{F(\varepsilon)} \log \frac{1}{\delta}$ satisfies*

$$\mathbb{E}[r^\star(\widehat{y})] \geq \mathbb{E}^{\pi^\star}[r^\star(y)] - \varepsilon - \delta.$$

Note that this is interesting in the context of Theorem 4.2 since the output of VGB satisfies the above guarantee with $F(\varepsilon) = \frac{48H(1+\varepsilon_V)^2}{\varepsilon}$. Thus, by applying the Best-of-$N$ algorithm to $O(H)$ samples from VGB, we can compete with the optimal distribution $\pi^\star$ in expected reward (for any $r^\star$ that we can evaluate).

**Proof.** Let $\mathcal{E}_1$ be the event that $r^\star(y) = 1$, and let $\mathcal{E}_2$ be the event that $\pi^\star(y)/\pi(y) < F(\varepsilon)$. By assumption, we have $\pi^\star(\mathcal{E}_2) \geq 1 - \varepsilon$, and thus $\pi^\star(\mathcal{E}_1 \cap \mathcal{E}_2) \geq \mathbb{E}^{\pi^\star}[r^\star(y)] - \varepsilon$. It follows that

$$\pi(\mathcal{E}_1 \cap \mathcal{E}_2) \geq \frac{1}{F(\varepsilon)}(\mathbb{E}^{\pi^\star}[r^\star(y)] - \varepsilon).$$

Now recall the definition of the Best-of-$N$ algorithm; let $y^{(1)}, \ldots, y^{(N)}$ denote the candidates that it draws from $\pi$. The algorithm's output $\widehat{y}$ satisfies

$$\begin{aligned}
\mathbb{E}[r^\star(\widehat{y})] &\geq \mathbb{P}[\exists i \in [N] : y^{(i)} \in \mathcal{E}_1] \\
&\geq 1 - (1 - \pi(\mathcal{E}_1 \cap \mathcal{E}_2))^N \\
&\geq 1 - e^{-\frac{N}{F(\varepsilon)}(\mathbb{E}^{\pi^\star}[r^\star(y)] - \varepsilon)} \\
&\geq 1 - \delta^{\mathbb{E}^{\pi^\star}[r^\star(y)] - \varepsilon} \\
&\geq \mathbb{E}^{\pi^\star}[r^\star(y)] - \varepsilon - \delta
\end{aligned}$$

by choice of $N$ and the inequality $1 - y^x \geq x - y$ for all $x, y \in [0, 1]$.   $\square$

## C  PITFALLS OF BEAM SEARCH

The main focus of our work is on the task of *sampling*, which is often used to balance reward maximization and diversity / general capability (Ziegler et al., 2019). In practice, sometimes the *only* goal is reward maximization: given a base policy $\pi_{\mathsf{ref}} : \mathcal{X} \to \mathcal{Y}$ and query access to a reward function $r^\star : \mathcal{X} \times \mathcal{Y} \to [0, R_{\mathsf{max}}]$, the goal—given some prompt $x \in \mathcal{X}$—is to compute some $y \in \mathcal{Y}$ achieving maximal reward.

Given some process guidance function $G : \mathcal{X} \times \mathcal{A}^\star \to \mathbb{R}$, two classical, widely-used algorithms for this task are greedy decoding and beam search (Khanov et al., 2024; Wang et al., 2025). For some width parameter $W$, beam search with guidance function $G$ proceeds as follows. At each $h \in [H]$, it maintains $W$ *beams* $y_{1:h-1}^{(1)}, \ldots, y_{1:h-1}^{(W)} \in \mathcal{A}^{h-1}$ of length $h-1$. It then computes the $W|\mathcal{A}|$ beams that are all possible one-token extensions (if $|\mathcal{A}|$ is large, then instead the algorithm draws some number of candidate tokens from $\pi_{\mathsf{ref}}$). Finally, from these $W|\mathcal{A}|$ extended beams, it selects the $W$ extended beams $(y_{1:h-1}^{(i)}, y_h)$ with the maximal values of $G(y_{1:h-1}^{(i)}, y_h)$. After step $H$, the output is selected again by maximizing $G$. Greedy decoding is simply beam search with width $W = 1$.

No matter the choice of guidance function, it is clear that these algorithms are not sampling from the distribution $\pi^\star$ defined in Eq. (2), nor do they maintain diversity of generations in any sense, as their goal is pure reward maximization. In certain settings, they are provably effective at reward maximization: in particular, it is straightforward to see that with guidance function

$$G(x, y_{1:h}) := \max_{y_{h+1:H} \in \mathcal{A}^{H-h}} r^\star(x, y_{1:H}),$$

beam search finds some $y_{1:H}^\star \in \arg\max r^\star(x, y_{1:H})$ for any $W \geq 1$. This guidance function is the classical (optimal, unregularized) value function in the autoregressive MDP defined by the deterministic dynamics $(y_{1:h-1}, y_h) \mapsto y_{1:h}$.

However, our setting studies the benefits of the guidance function

$$G(x, y_{1:h}) := V_{\mathsf{tilt}}^\star(x, y_{1:h}) = \mathbb{E}^{\pi_{\mathsf{ref}}}[r^\star(x, y_{1:H}) \mid x, y_{1:h}],$$

which is the value function for $\pi_{\mathsf{ref}}$ in the autoregressive MDP. This "averaged" guidance function is natural since it can be directly estimated via regression onto Monte Carlo roll-outs, unlike the preceding function, and this is commonly done in practice (Yang & Klein, 2021; Wang et al., 2025). The following example demonstrates that with this value function, *even in the absence of estimation errors*, beam search is not the "right" algorithm for reward maximization: it can be substantially worse than algorithms that sample from $\pi^\star(y \mid x) \propto \pi_{\mathsf{ref}}(y \mid x) r^\star(x, y)$.[14]

**Example C.1** (Greedy decoding and beam search are suboptimal with $V_{\mathsf{tilt}}^\star$). *Let $\mathcal{X} := \{\perp\}$ (we omit dependencies on the prompt space henceforth) and $\mathcal{Y} := \{0, 1\}^H$ for some even $H$. Define $\pi_{\mathsf{ref}} := \mathsf{Unif}(\mathcal{Y})$. Define $r^\star : \mathcal{Y} \to [0, 1]$ by*

$$r^\star(y_{1:H}) := \begin{cases} 1 & \text{if } y_1 = y_{H/2+1:H} = 0 \\ 2^{1-H/2} & \text{if } y_1 = 1 \\ 0 & \text{otherwise} \end{cases}.$$

*Then the distribution $\pi^\star(y) \propto \pi_{\mathsf{ref}}(y) r^\star(y)$ satisfies*

$$\mathbb{E}^{\pi^\star}[r^\star(y)] \geq \sum_{y_{2:H/2}} \pi^\star(0, y_{2:H/2}, 0^{H/2}) = \frac{2^{H/2-1}2^{-H}}{2^{H/2-1}2^{-H} + 2^{H-1}2^{-H}2^{1-H/2}} = \frac{1}{3}.$$

*However, consider the execution of beam search for any $W \leq 2^{H/2-1}$. For every $h \leq H/2$, we have*

$$V_{\mathsf{tilt}}^\star(y_{1:h}) = \begin{cases} 2^{-H/2} & \text{if } y_1 = 0 \\ 2^{1-H/2} & \text{if } y_1 = 1 \end{cases}.$$

*In particular, $V_{\mathsf{tilt}}^\star(0, y_{2:h}) < V_{\mathsf{tilt}}^\star(1, y_{2:h}')$ for all $y_{2:h}, y_{2:h}'$. Thus, by step $H/2$, beam search will reject all beams with $y_1 = 0$. So it will find a response with reward $2^{1-H/2}$, which is $\Omega(1)$-suboptimal compared to $\pi^\star$.* ◁

---

[14]Of course, in some instances it can be better, so the algorithms are incomparable when the goal is reward maximization.

We can prove an analogous result in the KL-regularized optimization setting. In this setting, for a fixed base policy and reward function, the regularized value function $Q_\beta^\star(x, y_{1:h}) = \beta \log V_{\texttt{tilt}}^\star(x, y_{1:h})$ converges to the classical unregularized value function as $\beta \to 0$, so in this limit beam search will succeed. However, the following example demonstrates that even for $\beta = O(1/H)$, beam search can fail unless it uses width $W = 2^{\Omega(H)}$.

**Example C.2** (Greedy decoding and beam search fail with $Q_\beta^\star$). *Let $\mathcal{X} := \{\bot\}$ (we ignore the prompt space henceforth) and $\mathcal{Y} := \{0,1\}^H$ for some even $H > 2$. Define $\pi_{\texttt{ref}} := \mathsf{Unif}(\mathcal{Y})$. Set $\beta := \frac{1}{(H-2)\log(2)}$ so that $e^{\beta^{-1}/2} = 2^{H/2-1}$. Define $r^\star : \mathcal{Y} \to [0,1]$ by*

$$r^\star(y_{1:H}) := \begin{cases} 1 & \text{if } y_1 = y_{H/2+1:H} = 0 \\ \frac{1}{2} & \text{if } y_1 = 1 \\ 0 & \text{otherwise} \end{cases}$$

*similar to the preceding example. Then the distribution $\pi_\beta^\star(y) \propto \pi_{\texttt{ref}}(y) \exp(\beta^{-1} r^\star(y))$ satisfies*

$$\mathbb{E}^{\pi_\beta^\star}[r^\star(y)] = \frac{2^{H/2-1}2^{-H}e^{\beta^{-1}} + \frac{1}{2}2^{H-1}2^{-H}e^{\beta^{-1}/2}}{2^{H/2-1}2^{-H}e^{\beta^{-1}} + 2^{H-1}2^{-H}e^{\beta^{-1}/2} + (2^{H-1} - 2^{H/2-1})2^{-H}} = \frac{2}{3} - o(1).$$

*However, for every $h \leq H/2$, we have*

$$V_{\texttt{tilt}}^\star(y_{1:h}) = \mathbb{E}^{\pi_{\texttt{ref}}}[\exp(\beta^{-1} r^\star(y_{1:H})) \mid y_{1:h}] = \begin{cases} 2^{H/2-2} & \text{if } y_1 = 0 \\ 2^{H/2-1} & \text{if } y_1 = 1 \end{cases}.$$

*Thus, as in the preceding example, beam search will reject all beams with $y_1 = 0$ unless the width exceeds $2^{H/2-1}$. Therefore it will (deterministically) produce a response with reward only $1/2$.* ◁

## D INSUFFICIENCY OF ALTERNATIVE ERROR CONDITIONS

In this section, we discuss issues with three natural alternative error conditions to Assumption 4.2: an average-case error bound under $\pi_{\mathrm{ref}}$ rather than $\pi^\star$ (Appendix D.1), an additive error bound rather than multiplicative (Appendix D.2), and a mean-squared error bound on $\widehat{Q} = \beta \log \widehat{V}$ rather than an exponential MGF bound on $\widehat{Q}$ (Appendix D.3).

### D.1 AVERAGE-CASE MULTIPLICATIVE ERROR BOUND UNDER $\pi_{\mathrm{ref}}$

One plausible alternative to Assumption 4.2 is the following assumption, which is average-case over $\pi_{\mathrm{ref}}$ rather than $\pi^\star$: for each position $h \leq H$,

$$
\mathop{\mathbb{E}}_{y_{1:h} \sim \pi_{\mathrm{ref}}(\cdot|x)} \left[ \frac{\widehat{V}(x, y_{1:h})}{V^\star_{\mathrm{tilt}}(x, y_{1:h})} \right] \leq 1 + \varepsilon_V, \qquad \mathop{\mathbb{E}}_{y_{1:h} \sim \pi_{\mathrm{ref}}(\cdot|x)} \left[ \frac{V^\star_{\mathrm{tilt}}(x, y_{1:h})}{\widehat{V}(x, y_{1:h})} \right] \leq 1 + \varepsilon_V. \tag{9}
$$

This modification seems particularly natural if $\widehat{V}$ is estimated via regression onto Monte Carlo roll-outs from $\pi_{\mathrm{ref}}$, since typical statistical learning guarantees will be average-case over $\pi_{\mathrm{ref}}$. Unfortunately, as the following example shows, test-time alignment is information-theoretically impossible with only this assumption, unless $\varepsilon_V$ is exponentially small. The basic reason is that distribution shift between $\pi_{\mathrm{ref}}$ and $\pi^\star$ at earlier positions $h$ can lead to the error bound being essentially vacuous on the effective support of $\pi^\star$ at later positions $h' > h$.

**Example D.1.** *Let $\mathcal{X} := \{\perp\}$ (we omit dependencies on prompt henceforth) and $\mathcal{Y} := \mathcal{A}^H$ where $H \in \mathbb{N}$ is even and $\mathcal{A} := \{0, 1\}$. Let $\pi_{\mathrm{ref}} := \mathsf{Unif}(\mathcal{Y})$. We define a family of test-time alignment instances $I_g$ parametrized by a sequence $g = g_{H/2+1:H} \in \mathcal{A}^{H/2}$. Each instance has base policy $\pi_{\mathrm{ref}}$ and tilt function $\tau_g : \mathcal{Y} \to \{0, 1\}$ defined by*

$$
\tau_g(y_{1:H}) = \mathbb{I}[y_{1:H/2} = 0 \wedge y_{H/2+1:H} = g] + \lambda \cdot \mathbb{I}[y_{1:H/2} = 0],
$$

*for a parameter $\lambda > 0$ to be determined. The optimal distribution $\pi^\star_g \propto \pi_{\mathrm{ref}} \cdot \tau_g$ satisfies $\pi^\star_g(0, g) = \frac{1+\lambda}{1+2^{H/2}\lambda}$, and*

$$
V^\star_{\mathrm{tilt},g}(y_{1:h}) = \begin{cases} 2^{h-H}\mathbb{I}[y_{1:h} = 0^h] + 2^{h-H/2}\lambda\mathbb{I}[y_{1:h} = 0^h] & \text{if } h \leq H/2 \\ 2^{h-H}\mathbb{I}[y_{1:h} = (0, g)_{1:h}] + \lambda\mathbb{I}[y_{1:H/2} = 0^{H/2}] & \text{otherwise} \end{cases}.
$$

*We define*

$$
\widehat{V}(y_{1:h}) := \begin{cases} V^\star_{\mathrm{tilt},g}(y_{1:h}) & \text{if } h \leq H/2 \\ \lambda\mathbb{I}[y_{1:H/2} = 0^{H/2}] & \text{otherwise} \end{cases}.
$$

*Then for any $h > H/2$, we have $\widehat{V}(y_{1:h}) \leq V^\star_{\mathrm{tilt},g}(y_{1:h})$ for all $y_{1:h}$, and*

$$
\mathop{\mathbb{E}}_{y_{1:h} \sim \pi_{\mathrm{ref}}} \left[ \frac{V^\star_{\mathrm{tilt},g}(y_{1:h})}{\widehat{V}(y_{1:h})} \right] \leq 1 + \frac{2^{h-H}}{\lambda} \mathop{\Pr}_{y_{1:h} \sim \pi_{\mathrm{ref}}} [y_{1:h} = (0, g)_{1:h}] = 1 + \frac{2^{-H}}{\lambda}.
$$

*Set $\lambda := 2^{-H/2}$. Then Eq. (9) is satisfied with $\varepsilon_V := 2^{-H/2}$. Moreover, $\widehat{V}$ does not depend on $g$, so for any alignment algorithm, the law $\widehat{\pi}$ of the output on instance $I_g$ is independent of $g$. It follows that there is some $g$ such that $\widehat{\pi}(0, g) \leq 1/2^{H/2}$. But by choice of $\lambda$, we have $\pi^\star_g(0, g) \geq 1/2$ for all $g$. Thus, there is some $g$ such that*

$$
\mathop{\Pr}_{y_{1:H} \sim \pi^\star_g} \left[ \frac{\pi^\star_g(y_{1:H})}{\widehat{\pi}(y_{1:H})} \geq 2^{H/2-1} \right] \geq \frac{1}{2},
$$

*i.e. there is a constant fraction of the mass of $\pi^\star_g$ where $\widehat{\pi}$ exponentially undercovers the responses.* ◁

### D.2 ADDITIVE ERROR BOUND FOR $\widehat{V}$

Typical statistical learning guarantees for regression give additive error bounds rather than multiplicative error bounds. Thus, it may seem natural to instead assume that for each position $h \in [H]$, for each $y_{1:h} \in \mathcal{A}^h$,

$$
|\widehat{V}(x, y_{1:h}) - V^\star_{\mathrm{tilt}}(x, y_{1:h})| \leq \varepsilon_V \tag{10}
$$

for a parameter $\varepsilon_V > 0$—or an analogous average-case bound. For simplicity, let us consider the worst-case version; the same issues apply to the weaker average-case versions.

The basic issue is that Eq. (10) is not appropriately scale-invariant: rescaling $\tau$, $V^\star_{\mathtt{tilt}}$, and $\widehat{V}$ by $1/2$ does not change the problem, but halves the additive error. Moreover, the following simple example demonstrates that even in the constrained sampling setting, where $\tau$ is binary-valued and therefore has a natural scale, Eq. (10) is insufficient unless $\varepsilon_V$ is exponentially small.

**Example D.2.** Let $\mathcal{X} = \{\perp\}$ and $\mathcal{Y} = \mathcal{A}^H$ where $H \in \mathbb{N}$ and $\mathcal{A} = \{0, 1\}$. Let $\pi_{\mathtt{ref}} := \mathsf{Unif}(\mathcal{Y})$. We define a family of test-time alignment instances $I_g$ parametrized by a sequence $g = g_{1:H} \in \mathcal{A}^H$. Each instance has base policy $\pi_{\mathtt{ref}}$ and tilt function $\tau_g : \mathcal{Y} \to \{0, 1\}$ defined by

$$\tau_g = \mathbb{I}[y = g].$$

Thus, the optimal distribution $\pi^\star_g \propto \pi_{\mathtt{ref}} \cdot \tau_g$ satisfies $\pi^\star_g(g) = 1$, and $V^\star_{\mathtt{tilt},g}(y_{1:h}) = 2^{h-H}\mathbb{I}[y_{1:h} = g_{1:h}]$. We define

$$\widehat{V}(y_{1:h}) := \begin{cases} 0 & \text{if } h \le H/2 \\ V^\star_{\mathtt{tilt},g}(y_{1:h}) & \text{otherwise} \end{cases}.$$

Then Eq. (10) is satisfied with $\varepsilon_V := 2^{-H/2}$. However, it is evident that observing a non-zero value of $\widehat{V}$ requires $2^{\Omega(H)}$ queries, so for any algorithm that makes $2^{o(H)}$ queries, there will be some instance $I_g$ on which its output distribution $\widehat{\pi}_g$ will exponentially undercover $\pi^\star_g$ (in the sense of the previous example) with high probability. We omit formal details of the query lower bound as such arguments are standard. ◁

**A scale-invariant additive assumption?** The failure mode of the above example is that $\pi_{\mathtt{ref}}$ has extremely low average reward—and it seems natural that the accuracy of the value function estimates in such a setting may necessarily have to be correspondingly more accurate. Motivated by this observation, we briefly discuss a scale-invariant version of Eq. (10), which scales the error according to the inherent "difficulty" of the problem:

**Assumption D.1.** For each position $h \in [H]$, for each $y_{1:h} \in \mathcal{A}^h$,

$$|\widehat{V}(x, y_{1:h}) - V^\star_{\mathtt{tilt}}(x, y_{1:h})| \le \varepsilon_V V^\star_{\mathtt{tilt}}(x).$$

The following proposition shows that Assumption D.1 actually *implies* Assumption 4.2, with a slight adjustment to the approximate value function—that can be tractably computed if $V^\star_{\mathtt{tilt}}(x)$ is known (i.e. we have some estimate of the "difficulty" of prompt $x$ for $\pi_{\mathtt{ref}}$).

**Proposition D.1.** Let $\varepsilon_V > 0$. Under Assumption D.1 with parameter $\varepsilon_V$, the (modified) approximate value function $\widehat{V}'(x, y_{1:h}) := \widehat{V}(x, y_{1:h}) + \varepsilon_V V^\star_{\mathtt{tilt}}(x)$ satisfies Assumption 4.2 with parameter $2\varepsilon_V$.

**Proof.** Since $\pi^\star(y_{1:h} \mid x) = \frac{1}{V^\star_{\mathtt{tilt}}(x)}\pi_{\mathtt{ref}}(y_{1:h} \mid x)V^\star_{\mathtt{tilt}}(x, y_{1:h})$, we have

$$\underset{y_{1:h}\sim\pi^\star(\cdot|x)}{\mathbb{E}}\left[\frac{\widehat{V}'(x, y_{1:h})}{V^\star_{\mathtt{tilt}}(x, y_{1:h})}\right] = 1 + \underset{y_{1:h}\sim\pi^\star(\cdot|x)}{\mathbb{E}}\left[\frac{\widehat{V}'(x, y_{1:h}) - V^\star_{\mathtt{tilt}}(x, y_{1:h})}{V^\star_{\mathtt{tilt}}(x, y_{1:h})}\right]$$

$$= 1 + \underset{y_{1:h}\sim\pi_{\mathtt{ref}}(\cdot|x)}{\mathbb{E}}\left[\frac{\widehat{V}'(x, y_{1:h}) - V^\star_{\mathtt{tilt}}(x, y_{1:h})}{V^\star_{\mathtt{tilt}}(x)}\right]$$

$$= 1 + \varepsilon_V + \underset{y_{1:h}\sim\pi_{\mathtt{ref}}(\cdot|x)}{\mathbb{E}}\left[\frac{\widehat{V}(x, y_{1:h}) - V^\star_{\mathtt{tilt}}(x, y_{1:h})}{V^\star_{\mathtt{tilt}}(x)}\right]$$

$$\le 1 + 2\varepsilon_V$$

and, for every $y_{1:h} \in \mathcal{A}^h$, $\widehat{V}'(x, y_{1:h}) = \widehat{V}(x, y_{1:h}) + \varepsilon_V V^\star_{\mathtt{tilt}}(x) \ge V^\star_{\mathtt{tilt}}(x, y_{1:h})$, so that

$$\underset{y_{1:h}\sim\pi^\star(\cdot|x)}{\mathbb{E}}\left[\frac{V^\star_{\mathtt{tilt}}(x, y_{1:h})}{\widehat{V}'(x, y_{1:h})}\right] \le 1$$

which suffices. $\square$

This result provides additional motivation for Assumption 4.2, and suggests a potential algorithmic intervention to make an approximate value function more useful—namely, adding some estimate of $V^\star_{\text{tilt}}(x)$ to the approximation. However, it remains unclear whether Assumption D.1 is sufficient when $V^\star_{\text{tilt}}(x)$ is unknown. It is also unclear whether an average-case variant of Assumption D.1 could suffice for efficient test-time alignment (in the proof of Proposition D.1, the first part goes through with an average-case error bound under $\pi_{\text{ref}}$, but the second part does not)—and in what regimes of $\varepsilon_V$ it would enable improving upon the error guarantees of ActionLevelRS. We leave these interesting open questions for future work.

## D.3   MEAN-SQUARED ERROR BOUND FOR $\widehat{Q}$

As discussed in Section 4.3, in the KL-regularized setting, Assumption 4.2 corresponds to an MGF bound on $\Delta(x, y_{1:h}) := Q^\star_\beta(x, y_{1:h}) - \widehat{Q}(x, y_{1:h})$; it is implied by:

$$\mathbb{E}_{y_{1:h} \sim \pi^\star(\cdot|x)}\left[\exp\left(\beta^{-1}|\Delta(x, y_{1:h})|\right)\right] \leq 1 + \varepsilon_V. \tag{11}$$

While an average-case assumption, this is still stronger than typical statistical guarantees for regression (e.g., Wainwright (2019)). One may naturally ask whether a mean-squared error bound suffices for Theorem 4.2; we show that it does not.[15] This result demonstrates a distinction between the $H = 1$ setting (Huang et al., 2025b) and our setting.

To state the formal result, we introduce the following oracle framework for KL-regularized optimization, in which the algorithm has sample access to the base model and query access to the approximate value function (but does not have access to the true outcome-level reward). Note that Theorem 4.2 applies in this setting. See Huang et al. (2025b); Foster et al. (2025) for similar models.

**Definition D.1** (Oracle framework for KL-regularized optimization). *Let $\pi_{\text{ref}}$ be a base model, let $r^\star$ be a reward function, and let $\beta > 0$. We consider algorithms that have access to the following oracles:*

- *Value function oracle: Given a prompt $x \in \mathcal{X}$ and a partial completion $y_{1:h} \in \mathcal{A}^h$, returns $\widehat{Q}(x, y_{1:h}) = \beta \log \widehat{V}(x, y_{1:h})$.*

- *Conditional density oracle: Given a prompt $x \in \mathcal{X}$ and a partial completion $y_{1:h-1} \in \mathcal{A}^{h-1}$, returns $\pi_{\text{ref}}(y_h \cdot \mid x, y_{1:h-1})$ for all $y_h \in \mathcal{A}$.*

We show that if Eq. (11) is relaxed to an $\ell_p$ error bound for any constant $p$, then there is no algorithm in the above framework that achieves an analogous "approximate coverage" guarantee to Theorem 4.2.

**Theorem D.1.** *Let $p \geq 2$ and let $H \in \mathbb{N}$ be given. Set $\mathcal{X} := \{\perp\}$, $\mathcal{A} := \{0, 1\}$ and $\mathcal{Y} = \mathcal{A}^H$. For any algorithm in the above oracle framework with $\beta := \frac{1}{2(p+2)H \log H}$, there exists a base model $\pi_{\text{ref}} : \mathcal{X} \to \Delta(\mathcal{Y})$, reward function $r^\star : \mathcal{X} \times \mathcal{Y} \to [0, 1]$, and approximate value function $\widehat{Q}$ satisfying*

$$\mathbb{E}_{\pi_{\text{ref}}}\left[\left(\widehat{Q}(x, y_{1:h}) - Q^\star_\beta(x, y_{1:h})\right)^p\right] \leq \varepsilon_Q^p, \quad \text{and} \quad \mathbb{E}_{\pi^\star_\beta}\left[\left(\widehat{Q}(x, y_{1:h}) - Q^\star_\beta(x, y_{1:h})\right)^p\right] \leq \varepsilon_Q^p, \tag{12}$$

*where $\varepsilon_Q = O(\beta)$, and yet the output distribution $\widehat{\pi}$ of the algorithm satisfies*

$$\mathbb{P}^{\pi^\star_\beta}\left[\frac{\pi^\star_\beta(y)}{\widehat{\pi}(y)} > 2^{H-3}\right] \geq \frac{1}{4}.$$

In other words, there can be a constant-fraction of the mass of $\pi^\star_\beta$ where $\widehat{\pi}$ exponentially undercovers the responses. This result can be contrasted with Theorem 4.2 and Assumption 4.2.

Note that this result is agnostic to the query complexity of the sampling algorithm in the above oracle framework; there is simply not enough information in $\widehat{Q}$ to achieve approximate coverage. However, the result does not capture algorithms that have access to the true outcome-level rewards. In that case, there is (existentially) enough information to sample from the true distribution—but since the above result shows that the process verifier $\widehat{Q}$ can be completely uninformative, we expect that there is a query-complexity lower bound in this stronger setting.

---

[15]It is also not evident whether there is a statistical regression procedure that achieves an MSE bound on $\widehat{Q}$ (rather than $\widehat{V}$), but this result shows such a bound would still be insufficient.

**Remark D.1.** *Consider the case $p = 2$. To interpret the result, we note the value function $\widehat{Q}$ has low mean-squared error (MSE) under two policies: $\pi^\star$ and $\pi_{\mathsf{ref}}$. Low MSE under $\pi^\star$ is the MSE analogue of [Assumption 4.2](#), which the above result shows is information-theoretically insufficient to achieve a comparable coverage guarantee to [Theorem 4.2](#); this is true even when $H = 1$ (if the action space is large).*

*The mean-squared error condition with respect to $\pi_{\mathsf{ref}}$ is more subtle. In the case of $H = 1$, [Huang et al. (2025b, Theorem 4.1)](#) show that using an algorithm based on pessimism, one can achieve runtime that scales with $\mathcal{C}_{\mathsf{act}}(x)$ using only the assumption of low MSE under $\pi_{\mathsf{ref}}$. Our lower bound demonstrates that when $H$ is large, MSE is no longer (even information-theoretically) sufficient to achieve such a guarantee—essentially since* distribution shift *between $\pi_{\mathsf{ref}}$ and $\pi^\star$ arises in multi-step settings.*

**Proof of [Theorem D.1](#).** Set $\varepsilon := 1/H^{p+2}$. Fix $\pi_{\mathsf{ref}} := \mathrm{Ber}(1 - \varepsilon) \otimes \mathrm{Ber}(1/2)^{\otimes(H-1)}$ and $\beta := 1/(2H \log(1/\varepsilon))$. We define a family of instances where the reward function $r^\star$ and optimal policy $\pi_\beta^\star$ are indexed by a sequence $g = g_{2:H} \in \mathcal{A}^H$. Instance $I(g)$ is defined by reward function

$$r^\star(y_{1:H}) := \frac{1}{2} + \beta \log \frac{\pi_\beta^\star(y_{1:H})}{\pi_{\mathsf{ref}}(y_{1:H})},$$

where $\pi_\beta^\star \in \Delta(\mathcal{Y})$ is defined as the law of the following process: sample $y_1 \sim \mathrm{Ber}(1/2)$. If $y_1 = 1$, then sample $y_{2:H} \sim \mathrm{Ber}(1/2)^{\otimes(H-1)}$. Otherwise, independently sample $y_2, \ldots, y_H$ so that $\mathbb{P}[y_i = g_i] = 1 - \varepsilon$. By definition of $r^\star$, we have that $\pi_\beta^\star(y_{1:H}) \propto \pi_{\mathsf{ref}}(y_{1:H}) \exp(\beta^{-1} r^\star(y_{1:H}))$, so $r^\star$ and $\pi^\star$ define a valid instance of the KL-regularized optimization problem. Moreover, it is straightforward to check that $\pi_\beta^\star(y_{1:H})/\pi_{\mathsf{ref}}(y_{1:H}) \in [\varepsilon^{-H}, \varepsilon^H]$ for all $y_{1:H}$, and therefore $r^\star(y_{1:H}) \in [0, 1]$ by choice of $\beta$.

We have

$$Q_\beta^\star(y_{1:h}) = \beta \log \mathbb{E}^{\pi_{\mathsf{ref}}}[\exp(\beta^{-1} r^\star(y_{1:H})) \mid y_{1:h}] = \frac{1}{2} + \beta \log \frac{\pi_\beta^\star(y_{1:h})}{\pi_{\mathsf{ref}}(y_{1:h})}.$$

Define the approximation $\widehat{Q}$ by

$$\widehat{Q}(y_{1:h}) := \begin{cases} Q_\beta^\star(y_{1:h}) & \text{if } y_1 = 1 \\ Q_\beta^\star(0, g_{2:h}) & \text{otherwise} \end{cases}.$$

Since $r^\star(y_{1:H}) \in [0, 1]$ for all $y_{1:H}$, we have $Q_\beta^\star(y_{1:h}) \in [0, 1]$ for all $y_{1:h}$, and therefore similarly $\widehat{Q}(y_{1:h}) \in [0, 1]$ for all $y_{1:h}$. It follows that

$$\mathbb{E}^{\pi_{\mathsf{ref}}}\left[\left(\widehat{Q}(y_{1:h}) - Q_\beta^\star(y_{1:h})\right)^p\right] \leq \mathbb{P}^{\pi_{\mathsf{ref}}}[y_1 = 0]$$
$$= \varepsilon.$$

Similarly,

$$\mathbb{E}^{\pi_\beta^\star}\left[\left(\widehat{Q}(y_{1:h}) - Q_\beta^\star(y_{1:h})\right)^p\right] \leq \mathbb{P}^{\pi_\beta^\star}[y_1 = 0 \wedge y_{2:h} \neq g_{2:h}]$$
$$\leq \varepsilon H.$$

Note that $(\varepsilon H)^{1/p} \leq O(1/(2H \log(1/\varepsilon))) = O(\beta)$ so long as $\varepsilon \leq 1/H^{p+2}$. Thus, for each instance $I(g)$, the approximate value function satisfies the required accuracy guarantees. However, we can observe that $\widehat{Q}$ does not depend on $g$. Thus, the law $\widehat{\pi}$ of the sampling algorithm's output on instance $I(g)$ does not depend on $g$. For each $g$, in instance $I(g)$ it holds that $\pi_\beta^\star(0, g_{2:H}) = \frac{1}{2}(1 - \varepsilon)^{H-1} \geq \frac{1}{4}$. However, there is some $g_{2:H}$ such that $\widehat{\pi}(0, g_{2:H}) \leq 2^{1-H}$. Thus, in instance $I(g)$, it holds that

$$\mathbb{P}^{\pi_\beta^\star}\left[\frac{\pi_\beta^\star(y)}{\widehat{\pi}(y)} > 2^{H-3}\right] \geq \frac{1}{4}$$

as claimed. $\qquad\square$

# E  FURTHER EXPERIMENTAL DETAILS AND OMITTED FIGURES

**Basic setting.**  Our experimental tasks are instantiations of the following basic setup. Let $\mathcal{X}$ denote a prompt space, and let $\mathcal{Y} := \mathcal{A}^H$ denote a response space. Let $\pi_{\mathsf{ref}} : \mathcal{X} \to \Delta(\mathcal{Y})$ be a generative model from which we can draw autoregressive samples (and compute next-token conditional probabilities). Let $r^\star : \mathcal{X} \times \mathcal{Y} \to \{0, 1\}$ be a binary reward function that we can query (we will make it available to the generation algorithms in tasks where we wish to measure *distributional* accuracy, but not in tasks where we wish to measure pure reward maximization). Given a prompt $x \in \mathcal{X}$, our primary goal is to sample from the distribution $\pi^\star(\cdot \mid x) \in \Delta(\mathcal{Y})$ defined as the conditional distribution of $\pi_{\mathsf{ref}}(\cdot \mid x)$ on reward-1 responses:

$$\pi^\star(y \mid x) \propto \pi_{\mathsf{ref}}(y \mid x) \cdot r^\star(x, y).$$

In different tasks, we will measure performance at this goal in different ways. As discussed in Section 1, it can be checked that $\pi^\star$ satisfies

$$\pi^\star(y_h \mid x, y_{1:h-1}) \propto \pi_{\mathsf{ref}}(y_h \mid x, y_{1:h-1}) \cdot V^\star_{\mathtt{tilt}}(x, y_{1:h})$$

where

$$V^\star_{\mathtt{tilt}}(x, y_{1:h}) = \mathbb{E}^{\pi_{\mathsf{ref}}}[r^\star(x, y_{1:H}) \mid y_{1:h}].$$

**Value function estimation by Monte Carlo rollouts.**  For all but the last of our tasks, we will *train* an approximation of $V^\star_{\mathtt{tilt}}$ via regression onto Monte Carlo rollouts. In particular, expanding on the discussion in Section 1.2, the above expression for $V^\star_{\mathtt{tilt}}$ suggests the following natural way to train a value function, which approximates the common approach in prior work (Yang & Klein, 2021; Wang et al., 2025):[16]

1. Draw samples $(x^{(i)}, y^{(i)}_{1:H})^N_{i=1}$, where $x^{(i)} \sim \rho$ is from some prompt distribution and $y^{(i)}_{1:H} \sim \pi_{\mathsf{ref}}(\cdot \mid x^{(i)})$.

2. For each $i \in [N]$, evaluate the reward $r^{(i)} := r^\star(x^{(i)}, y^{(i)}_{1:H})$.

3. For each $h \in [H]$, solve the regression problem

$$\widehat{V}_h \leftarrow \arg\min_{V \in \mathcal{V}} \sum_{i=1}^N (V_h(x^{(i)}, y^{(i)}_{1:h}) - r^{(i)})^2.$$

In specific experiments, we may deviate slightly from this formula, and we will describe such details in the corresponding sections.

**Organization of section.**  In Appendix E.1, we discuss in detail how we implement VGB and baselines, focusing particularly on the differences from the theoretical versions of the algorithms (made for efficiency and avoidance of unnecessary hyperparameters). In Appendices E.2 to E.6 we provide details on the task setup, value function training setup, and evaluation protocol for each of the ABC task, parity task, Dyck grammar task, Python test case generation task, and letter avoidance task respectively. In Appendix E.7 we provide additional exploratory results.

## E.1  IMPLEMENTATION DETAILS FOR SAMPLING ALGORITHMS

**Implementation details for VGB.**  Recall that the provable guarantees for VGB require repeatedly running the Markov Chain for a fixed number of steps, until the resulting generation is a leaf node of the autoregressive tree (Algorithm 1). For our practical implementation of VGB, we instead return the *first leaf reached during the random walk.*[17] We also remove self-loop transitions, as these now have no effect. For tasks with small alphabet size (ABC, Parity, and Dyck grammar) these modifications are the only changes made for the implementation; see Algorithm 3.

For tasks with large alphabet size (Python test case generation and letter avoidance), we make one additional change. Enumerating the alphabet at each step would be computationally infeasible. Instead, we implement the transitions using an implicit approximation of rejection sampling (that

---

[16]One difference is that these works typically do not train a separate value function for each position $h \in [H]$; we revisit this detail in the Python test case generation task (Appendix E.5).

[17]We enforce a fixed generation length in all of our experiments (so all leaves of the tree are at the same height $H$), but we expect that our methods can be adapted to settings where generation lengths vary. We treat EOS as a normal token.

avoids needing to estimate a normalization constant, unlike Algorithm 5). At iteration $t$, we sample $K = 32$ candidate tokens from $\pi_{\mathsf{ref}}(\cdot \mid x, y_{1:h}^{(t)})$. Append each of these to $y_{1:h}^{(t)}$ and let the resulting $K$ sequences be denoted by $z[1], \ldots, z[K] \in \mathcal{A}^{h+1}$. Also define $z[0] := y_{1:h-1}^{(t)}$. We define a probability distribution $\widehat{p}^{(t)}$ over $z[0], \ldots, z[K]$ by

$$\widehat{p}^{(t)}(z[0]) \propto K \cdot \widehat{V}(x, y_{1:h}^{(t)})$$

and, for $1 \le i \le K$,

$$\widehat{p}^{(t)}(z[i]) \propto \widehat{V}(x, z[i]).$$

We then sample $y^{(t+1)} \sim \widehat{p}^{(t)}$. It can be checked that as $K \to \infty$, this distribution approximates the ideal distribution over the entire neighborhood of $y_{1:h}^{(t)}$.

**Implementation details for `VGB-Momentum`.** We implement a version of VGB with momentum, analogous to the modification made by Hayes & Sinclair (2010) to the original Sinclair-Jerrum chain. This means that there are two copies of each node of the generation tree, and each transition of the original undirected Markov Chain is replaced by a directed cycle; the crossing flows are then cancelled out to decrease diffusive behavior. See Hayes & Sinclair (2010) for the details of the modification, which carries over readily to our setting. The only practical modification we make is to stop the Markov Chain as soon as it reaches a leaf (as with VGB). We only implement this algorithm for settings with small alphabets, but one can see that it would readily extend to large alphabets by the technique described above.

**Implementation details for `ActionLevelRS`.** When the true rewards are used for leaf nodes of the generation tree, `ActionLevelRS` may get "stuck", i.e. all continuation probabilities may be 0. In this case, we simply restart the algorithm at the root node. See Algorithm 4 for the implementation of `ActionLevelRS` for small alphabets. The implementation for large alphabets is the same as described above for VGB (again with $K = 32$) except that the transition distribution puts zero mass on the backtracking candidate $z[0] = y_{1:h-1}^{(t)}$. We do not tune the choice of $K$ for either algorithm.

**Implementation details for `OutcomeLevelRS`.** This algorithm is particularly simple in the constrained sampling setting: it simply repeatedly draws responses from $\pi_{\mathsf{ref}}$ until we find a response with reward 1.

**Implementation details for Block Best-of-$B$ and Block Rejection Sampling.** In both algorithms, we generate via an iterative procedure: sample $B$ blocks of length $L$, from $\pi_{\mathsf{ref}}$ conditioned on the current partial generation. Then choose one of the resulting continuations by either (a) choosing the continuation with the largest estimated value (Block Best-of-$B$), or (b) sampling the continuations proportional to their values (Block Rejection Sampling).

**Efficiency metric: step count.** Besides various error, diversity, and accuracy metrics, we will often measure the efficiency of different algorithms. Our main efficiency metric is *step count*. For VGB, this is the number of transitions of the Markov chain until it reaches a leaf node; for `ActionLevelRS`, this is exactly the horizon $H$ (i.e. length of responses) unless it gets "stuck" and either restarts or (in the large-$|\mathcal{A}|$ setting of Appendix E.6) has to redraw and evaluate new candidate tokens. For Block Best-of-$B$ and Block Rejection Sampling, this is $BH$ where $B$ is the number of candidate blocks. For `OutcomeLevelRS`, this is $nH$ where $n$ is the number of repetitions required to find a reward-1 response.[18]

### E.2   ABC TASK

**Task setup.** Let $H \in \mathbb{N}$ be a fixed horizon length, and define alphabet $\mathcal{A} := \{\mathfrak{a}, \mathfrak{b}, \mathfrak{c}\}$. Define $\mathcal{X} := \{\bot\}$ to be the trivial prompt space, and define the response space to be $\mathcal{Y} := \mathcal{A}^H$. Define $\pi_{\mathsf{ref}} \in \Delta(\mathcal{Y})$ to be uniform over $\mathcal{Y}$. Define reward function $r^\star(y) := \mathbb{I}[y_h \in \{\mathfrak{a}, \mathfrak{b}\} \forall h \in [H]]$. Thus, $\pi^\star$ is uniform over $\{\mathfrak{a}, \mathfrak{b}\}^H$. This task, following Example 3.1, is designed to measure the effect of benign training errors on the sampling algorithms; note that there should be no computational obstacles to learning the true value function.

---

[18]We do not argue that step count is the only efficiency metric one might care about; indeed, in practice there are numerous other factors that heavily influence wall-clock time, including but not limited to the number of language model queries per step, number of value function queries per step, key-value caching, and ease of parallelization; some of these are the focus of entire bodies of research in guided generation (Lipkin et al., 2025). These considerations are largely orthogonal to the focus of our work, for which step count is a simple theoretically-aligned metric. We do nevertheless demonstrate that VGB can outperform `ActionLevelRS` in strict wall-clock time, albeit in the synthetic setting of the Parity task (Appendix E.3).

---

**Algorithm 3** Practical implementation of VGB for small alphabets

---

1: **input:** Reference model $\pi_{\mathsf{ref}}$; approximate value function $\widehat{V}$, prompt $x \in \mathcal{X}$, horizon $H \in \mathbb{N}$.
   `// If outcome-level reward is available, set` $\widehat{V}(x, y_{1:H}) = r^\star(x, y_{1:H})$.
2: Initialize $y^{(0)} := \varnothing$ and $t = 0$.
3: **while** $|y^{(t)}| < H$ **do**
4:     Set $h := |y^{(t)}|$.
5:     Define $p^{(t)}$ as the distribution over neighborhood $\mathcal{N}(y_{1:h}^{(t)})$ of $y_{1:h}^{(t)}$ where

$$
p^{(t)}(y_{1:h-1}^{(t)}) \propto \begin{cases} \widehat{V}(x, y_{1:h}^{(t)}) & \text{if } h > 0, \\ 0 & \text{if } h = 0, \end{cases}
$$

and for each $y_{h+1} \in \mathcal{A}$,

$$
p^{(t)}(y_{1:h}^{(t)}, y_{h+1}) \propto \begin{cases} \pi_{\mathsf{ref}}(y_{h+1} \mid x, y_{1:h}^{(t)}) \, \widehat{V}(x, y_{1:h}^{(t)}, y_{h+1}) & \text{if } h < H, \\ 0 & \text{if } h = H. \end{cases}
$$

6:     Sample $y^{(t+1)} \sim p^{(t)}$, and increment $t$.
7: **return** $y^{(t)}$.

---

**Algorithm 4** Practical implementation of `ActionLevelRS` for small alphabets

---

1: **input:** Reference model $\pi_{\mathsf{ref}}$; approximate value function $\widehat{V}$, prompt $x \in \mathcal{X}$, horizon $H \in \mathbb{N}$.
   `// If outcome-level reward is available, set` $\widehat{V}(x, y_{1:H}) = r^\star(x, y_{1:H})$.
2: Initialize $y^{(0)} := \varnothing$ and $t = 0$.
3: **while** $|y^{(t)}| < H$ **do**
4:     **if** $\sum_{y_{h+1} \in \mathcal{A}} \pi_{\mathsf{ref}}(y_{h+1} \mid x, y_{1:h}^{(t)}) \widehat{V}(x, y_{1:h}^{(t)}, y_{h+1}) = 0$ **then**
5:         Set $y^{(t+1)} \leftarrow \varnothing$ and increment $t$.
6:     Set $h := |y^{(t)}|$.
7:     Define $p^{(t)}$ as the distribution over neighborhood $\mathcal{N}(y_{1:h}^{(t)})$ of $y_{1:h}^{(t)}$ where for each $y_{h+1} \in \mathcal{A}$,

$$
p^{(t)}(y_{1:h}^{(t)}, y_{h+1}) \propto \begin{cases} \pi_{\mathsf{ref}}(y_{h+1} \mid x, y_{1:h}^{(t)}) \, \widehat{V}(x, y_{1:h}^{(t)}, y_{h+1}) & \text{if } h < H, \\ 0 & \text{if } h = H. \end{cases}
$$

8:     Sample $y^{(t+1)} \sim p^{(t)}$, and increment $t$.
9: **return** $y^{(t)}$.

---

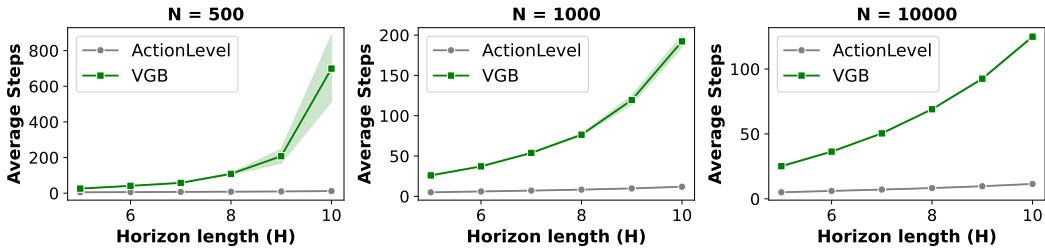

Figure 4: Average step counts for ABC task ([Appendix E.2](#)). Each experiment is repeated 10 times and we plot the mean and standard error.

**Value function training setup.** Let $N \in \mathbb{N}$ be the number of training samples. We train a separate value network for each generation length $h \in [H - 1]$ (at length $H$ we use the true rewards). Each network takes as input a one-hot encoding of the generation and has one hidden layer of width 128; the internal activation function is a ReLU and the final activation function is a sigmoid. Each network is trained with 100 steps of Adam, and learning rate 0.01, on the binary cross-entropy loss from the $N$ Monte Carlo roll-outs.

**Evaluation protocol and results.** For fixed $H$ and $N$, we train the value functions as described above, and then draw $10 \cdot 2^H$ samples from VGB ([Algorithm 3](#)) and ActionLevelRS ([Algorithm 4](#)). For each sampling algorithm, we compute the KL divergence of the empirical distribution to $\pi^\star$. We repeat this experiment for 10 seeds, and all combinations of $N \in \{500, 1000, 10000\}$ and $H \in \{5, 6, 7, 8, 9, 10\}$. We observe that VGB robustly achieves lower KL-divergence than ActionLevelRS, with particularly pronounced benefits for larger $H$ ([Figure 2](#)). We also record the average steps used by each algorithm ([Figure 4](#)), and observe that the step count of VGB appears to grow roughly quadratically in $H$ for large $N$, as predicted by the theory, but may scale considerably worse for small $N$.

### E.3 PARITY TASK

**Task setup.** Let $K, M \in \mathbb{N}$. Let $H := MK$ be the horizon length, and define alphabet $\mathcal{A} := \{0, 1\}$. Define $\mathcal{X} := \{\bot\}$ to be the trivial prompt space, and define the response space to be $\mathcal{Y} := \mathcal{A}^H$. Define $\pi_{\mathsf{ref}} \in \Delta(\mathcal{Y})$ by

$$\pi_{\mathsf{ref}}(y_h \mid y_{1:h-1}) := \begin{cases} \mathbb{I}[y_h \equiv y_{h+1-K} + \cdots + y_{h-1} \bmod 2] & \text{if } h \in \{K, 2K, \ldots, MK\} \\ 1/2 & \text{otherwise} \end{cases}.$$

Define reward function $r^\star(y) := \mathbb{I}[y_{tK} = 0 \, \forall t \in [M]]$.

**Theoretical intuition for task.** Intuitively, this task is designed so that a decision has to be made at each position $h \equiv K - 1 \bmod K$, but the correctness of this decision is only made obvious at position $h + 1$ (a la [Example 3.2](#)). This is motivated by the common empirical observation that in language model generations there is often natural "segmentation" where a complete segment is easier to score than an incomplete segment ([Li et al., 2024](#)).

More precisely, with this setup, we observe that the true value function is

$$V^\star_{\mathsf{tilt}}(y_{1:h}) = 2^{\lceil h/K \rceil - M} \mathbb{I}[y_{tK} = 0 \, \forall 1 \le t < \lceil h/K \rceil] \begin{cases} \mathbb{I}[y_h = 0] & \text{if } h \equiv K \bmod K \\ \mathbb{I}\left[\sum_{i=h+2-K}^{h} y_i \equiv 0 \bmod 2\right] & \text{if } h \equiv K - 1 \bmod K \\ 1/2 & \text{otherwise} \end{cases}.$$

For any position $h \equiv K - 1 \bmod K$, learning the value function requires fitting a parity of $K - 1$ variables (as well as an AND clause)—a task that is notoriously challenging for gradient descent ([Barak et al., 2022](#)). In contrast, for any position $h \equiv K \bmod K$, learning the value function only requires learning an AND clause. Thus, we hypothesize that there is a broad training regime in which a value function trained via gradient descent will approximately have the following form:

$$\widehat{V}(y_{1:h}) \approx 2^{\lceil h/K \rceil - M} \mathbb{I}[y_{tK} = 0 \, \forall 1 \le t < \lceil h/K \rceil] \begin{cases} \mathbb{I}[y_h = 0] & \text{if } h \equiv K \bmod K \\ 1/2 & \text{if } h \equiv K - 1 \bmod K \\ 1/2 & \text{otherwise} \end{cases}. \quad (13)$$

If this is the case, then `ActionLevelRS` will make a mistake at each position $h \equiv K - 1 \mod K$ with probability $1/2$, and will therefore be *exponentially unlikely* to find *any* reward-1 response—let alone sample from the distribution $\pi^\star$ over reward-1 responses—whereas VGB may be able to automatically use the values at positions $h \equiv K \mod K$ to correct earlier errors (this is not a priori obvious from the theory, since $\widehat{V}(y_{1:h})/V^\star_{\text{tilt}}(y_{1:h})$ can be infinite, so Assumptions 4.1 and 4.2 are not technically satisfied, but a formal theoretical guarantee can likely be proven under Eq. (13)). This intuition is borne out by the experiment—where we train $\widehat{V}$ as discussed below.

**Value function training setup.** We use the same network parametrization as in the ABC task, with a separate network for each generation length (and true rewards at length $H$). We train each network with Adam for a single epoch, with $10^4$ batches of size 128 and learning rate 0.001, on the MSE loss from the batch of Monte Carlo roll-outs. We checkpoint the value functions every 1000 batches.

**Evaluation protocol and results.** We set $K = 4$ and vary $M \in \{8, 9, 10\}$. For each $N \in \{1000, 2000, \ldots, 9000\}$, after training the value functions for $N$ batches, we draw 100 samples each from VGB and `ActionLevelRS`, and we record how long each algorithm takes on average (in wall clock time), and how many steps it used. If the algorithm takes longer than 1 minute to generate a single sample, we call that sample a timeout, and we track the number of timeouts. We repeat the experiment for $K = 5$ (this time with $N \in \{5000, 6000, \ldots, 14000\}$ for computational reasons). Results are shown in Figs. 5 and 6.

We see that after early checkpoints, VGB robustly improves upon `ActionLevelRS` in all metrics, with increased benefits for larger $K$ and $M$ (and therefore larger horizon $H$). To illustrate the source of this improvement, for $K = 5$ and $M = 8$, we visualize the error in the trained value function at each position $h \in [H]$ (Figure 7). As theoretically predicted, training is substantially more efficient at positions $h \equiv 0 \mod K$ than at positions $h \equiv -1 \mod K$. VGB improves upon `ActionLevelRS` in the regime after the value function has been learned at positions $h \equiv 0 \mod K$ but before the value function has been learned at positions $h \equiv -1 \mod K$.

### E.4 DYCK GRAMMAR TASK

**Task setup.** The basic setup for this task follows Botta et al. (2025). Define $\mathcal{A} := \{(, ), [, ], B, E, P, S\}$. Define prompt space $\mathcal{X} := \mathcal{A}^{17}$ and response space $\mathcal{Y} := \mathcal{A}^{17}$. We define a reward function $r^\star : \mathcal{X} \times \mathcal{Y} \to \{0, 1\}$ where a pair $(x, y) \in \mathcal{X} \times \mathcal{Y}$ has reward 1 if and only if it has the form $xy = Bs_{1:32}E$ where $s_{1:32} \in \mathcal{A}^{32}$ lies in the Dyck grammar for $\{(, ), [, ]\}$.

We train a transformer-based language model $\pi_{\text{ref}}$ on a distribution of strings in $\mathcal{X} \times \mathcal{Y}$ that have reward 1. This distribution is designed so that square brackets appear with probability 0.8 whereas round parentheses appear with probability 0.2. Following Botta et al. (2025), tokens "P" and "S" are extraneous tokens that do not appear in the training data and are not present in reward-1 strings. [19]

While $\pi_{\text{ref}}$ typically achieves high reward on in-distribution prompts, as we will see subsequently, there are out-of-distribution prompts on which it achieves low average reward. These are the prompts on which we will train value functions and use value-guided samplers.

**Value function training setup.** We train a separate value network for each generation length $h \in [H]$. The network parametrization is the same as in previous tasks except the hidden layer has width 64 (this was not tuned, but simply chosen to mitigate overparametrization since the input layer has much larger width than in previous tasks). Each network is trained for 40 epochs on $10,000$ Monte Carlo roll-outs, using the Adam optimizer with learning rate 0.003, weight decay 0.1, batch size 32, and the MSE loss. We checkpoint the value functions at the end of every epoch.

**Accuracy/diversity tradeoff: protocol and results.** First, we fix the length-17 prefix

$$x := B((()((((((((()(((,$$

which was randomly chosen subject to the constraint that $\pi_{\text{ref}}$ has low success at producing reward-1 completions, i.e. $\mathbb{E}_{y \sim \pi_{\text{ref}}(\cdot|x)}[r^\star(x, y)]$ is small. We train value functions, via the procedure described above, on Monte Carlo rollouts from this specific prompt. For $e \in \{1, 2, 3, 5, 10, 40\}$, we draw 3000 completions of this prompt from VGB, using the learned value function checkpointed at $e$ epochs of training. For this first experiment, we do not use the true rewards, so the algorithm may produce

---

[19]This is consistent with practical LLMs: the vocabulary typically includes a few unused token IDs. Our results show that VGB is robust to unseen, irrelevant tokens in the vocabulary.

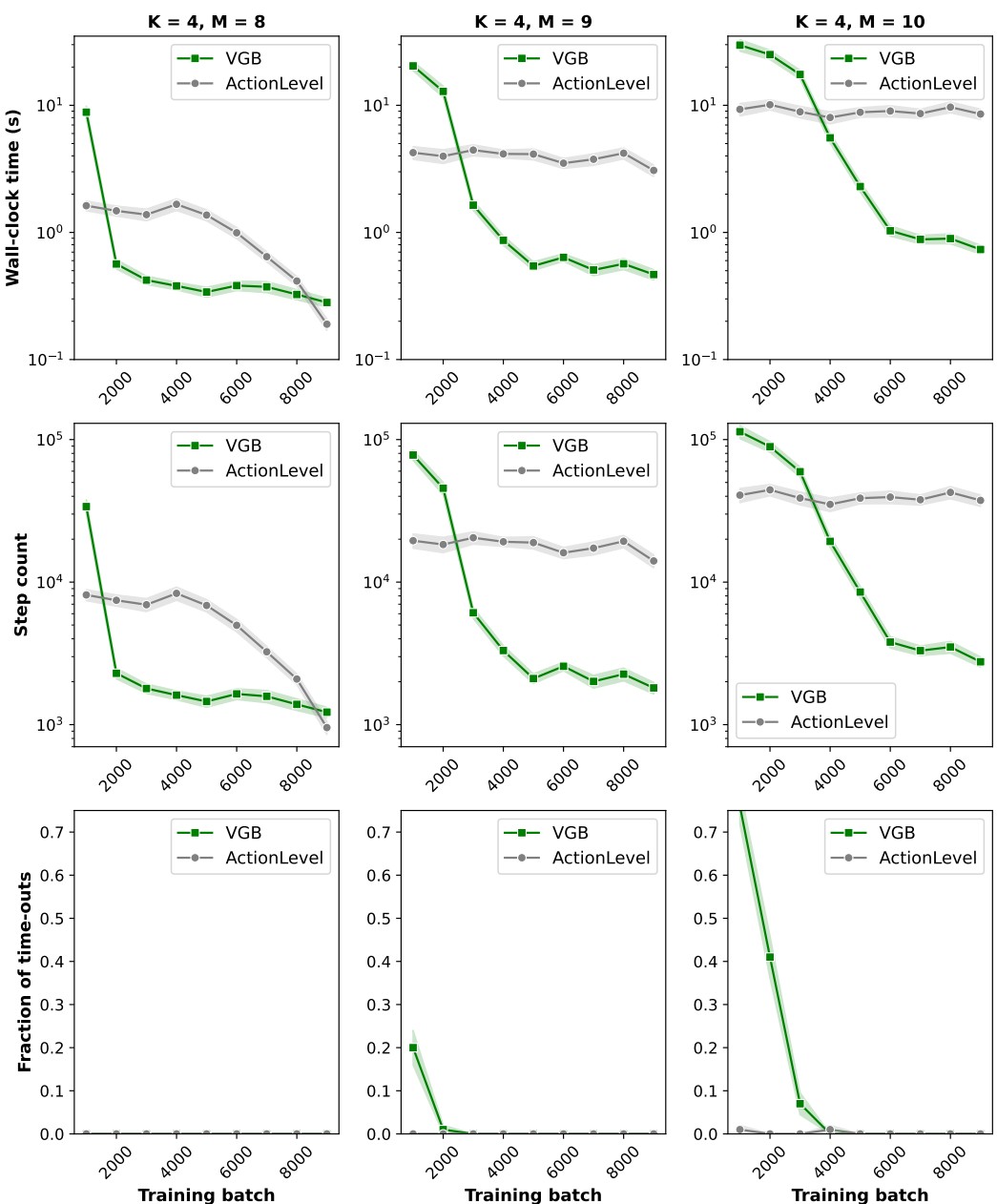

Figure 5: Efficiency of VGB and ActionLevelRS at finding a reward-1 response in Parity task (Appendix E.3) with $K = 4$ and $M \in \{8, 9, 10\}$ (so that $H \in \{32, 36, 40\}$), across training checkpoints $N \in \{1000, 2000, \ldots, 9000\}$ of the value function. We cap the time to draw a single response at 60 seconds. **Top row**: wall-clock time (in seconds), conditioned on not timing out; **middle row**: step count, conditioned on not timing out; **bottom row**: fraction of time-outs. For each $M$ and each training checkpoint, we report mean and standard error over 100 samples drawn from each algorithm. We plot wall-clock time and step count on a log scale to make evident that the multiplicative benefit of VGB over ActionLevelRS increases with $M$.

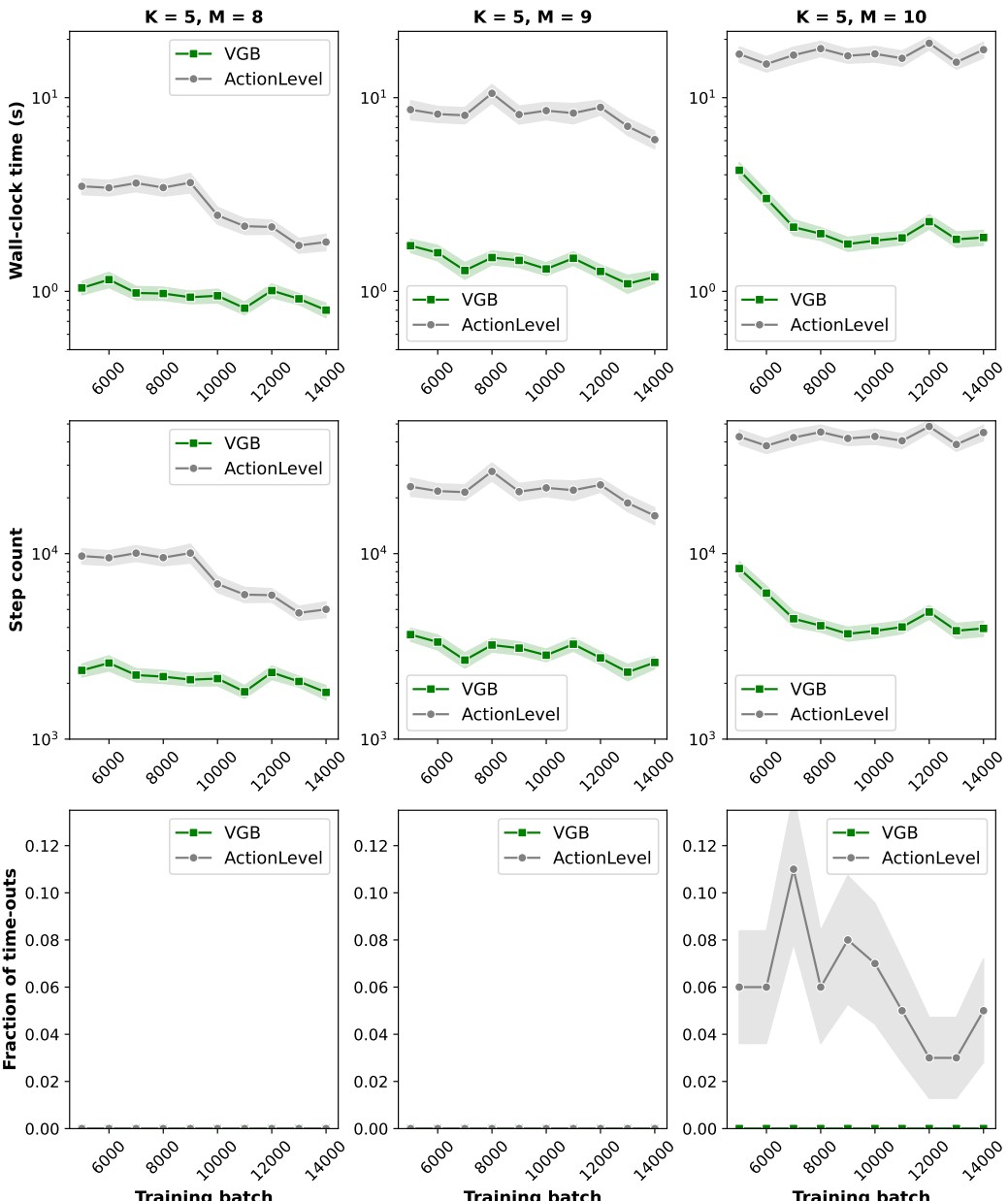

Figure 6: Efficiency of `VGB` and `ActionLevelRS` at finding a reward-1 response in Parity task (Appendix E.3) with $K = 5$ and $M \in \{8, 9, 10\}$ (so that $H \in \{40, 45, 50\}$), across training checkpoints $N \in \{5000, 6000, \ldots, 14000\}$ of the value function. We cap the time to draw a single response at 60 seconds. **Top row**: wall-clock time (in seconds), conditioned on not timing out; **middle row**: step count, conditioned on not timing out; **bottom row**: fraction of time-outs. For each $M$ and each training checkpoint, we report mean and standard error over 100 samples drawn from each algorithm. We plot wall-clock time and step count on a log scale to make evident that the multiplicative benefit of `VGB` over `ActionLevelRS` increases with $M$.

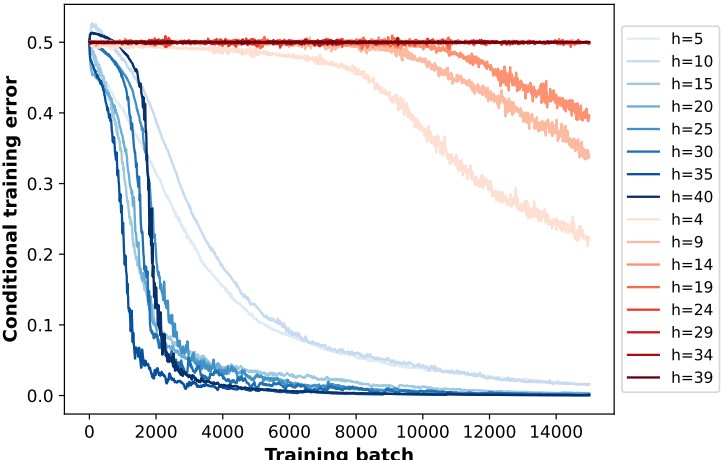

Figure 7: Visualization of training error for value function at each position $h \in [H]$ in Parity task (Appendix E.3) with $K = 5$ and $M = 8$, across training checkpoints of the value function. For each $h \in [H]$, we compute the probability that `ActionLevelRS` makes a mistake at position $h$ of generation, on average over 100 roll-ins $y_{1:h-1} \sim \pi^\star$. Since `ActionLevelRS` would never make a mistake with the true value function $V_{\texttt{tilt}}^\star$, this quantifies the effective training error at each position (while producing a more informative visualization than directly measuring MSE, as this metric is appropriately normalized across $h$). For clarity, we only plot positions $h \equiv -1, 0 \mod K$.

incorrect completions. We do the same for `VGB-Momentum`, `ActionLevelRS`, the base model $\pi_{\texttt{ref}}$, Block Best-of-$N$, and Block Rejection Sampling. For the latter two algorithms, we sweep over block lengths in $\{1, 2, 4, 8, 16\}$ and # of candidate blocks in $\{2, 4, 8, 16, 32\}$.

The results comparing VGB against the baselines are summarized in Figs. 8 and 9. We see that after the first few epochs, VGB achieves an accuracy/diversity tradeoff outside the Pareto frontier of Block Best-of-$N$ and Block Rejection Sampling. While we do not plot `ActionLevelRS` on these plots, we remark that it is comparable to Block Rejection Sampling with block length 1 and 32 candidate blocks. See Table 1 for a breakdown of the results for all baselines, including `ActionLevelRS` and $\pi_{\texttt{ref}}$ itself, when using the value function trained for 40 epochs. We observe from this table that the result is "query-equalized", i.e. we are allowing the baselines as many (in fact, slightly more) queries than used by VGB.

The results comparing `VGB-Momentum` against the baselines are summarized in Figs. 10 and 11. In these figures we only compare against baselines with # of candidate blocks in $\{2, 4, 8\}$, since these are the baselines with competitive query efficiency to `VGB-Momentum`.

**Distributional comparison: protocol and results.** In this experiment, using the same prompt as before, and varying the training epochs over $\{1, 2, 5, 10, 40\}$, we draw 1000 samples each from VGB, `VGB-Momentum`, and `ActionLevelRS`, this time using the true rewards at the last position so that each algorithm only produces reward-1 generations (as discussed previously, this requires restarting `ActionLevelRS` if it gets stuck). For each algorithm and epoch, we compute the empirical histogram of where open parentheses/brackets are located in the completion. To estimate the ground truth histogram, we draw 10000 samples from $\pi^\star$ (i.e. from $\pi_{\texttt{ref}}$ conditioned on reward 1). We then compute the $\ell_1$ error of each histogram compared to the estimated ground truth. For statistical power, we repeat the experiment for 11 other out-of-distribution prompts. As shown in Figure 12, for all epochs $e \geq 5$, VGB achieves robustly lower distributional error than `ActionLevelRS`; however, `VGB-Momentum` is comparable to `ActionLevelRS`. While `OutcomeLevelRS` (i.e. sequence-level rejection sampling from $\pi_{\texttt{ref}}$) would yield exactly the correct distribution, we see that this naive method is less query-efficient than VGB. See Table 2 for a prompt-by-prompt breakdown of the results at training epoch 40.

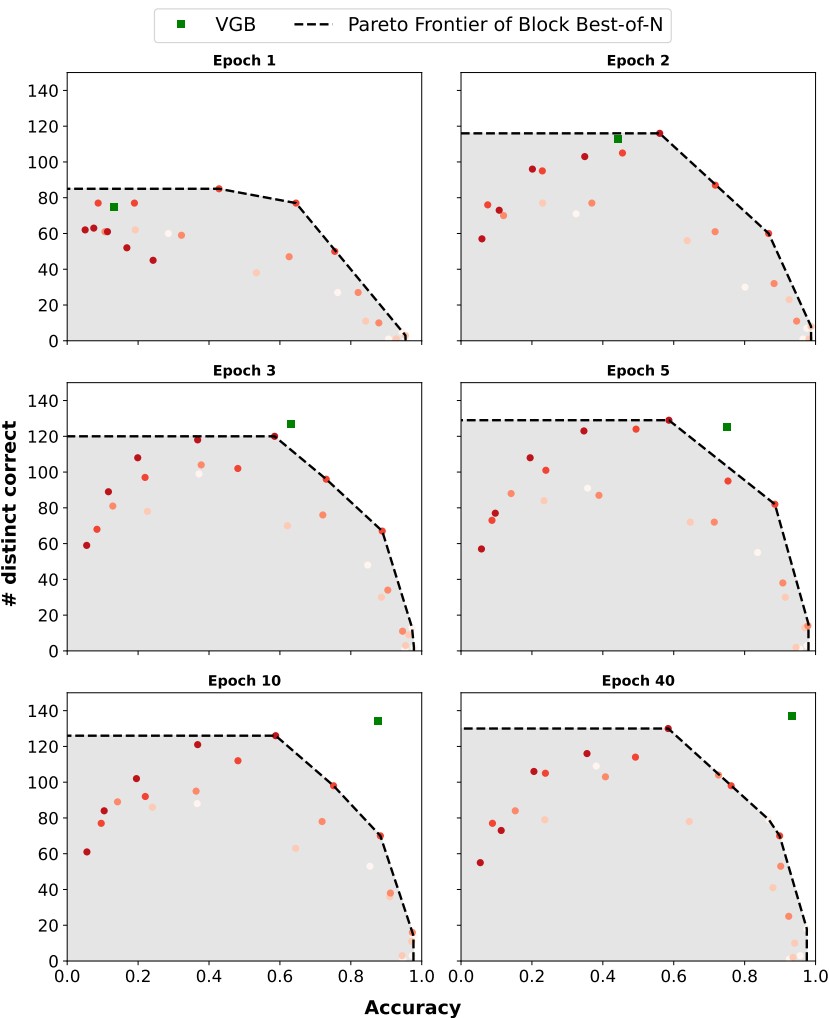

Figure 8: Accuracy (**x**) and diversity (**y**) of VGB vs. Block Best-of-N on Dyck grammar task (Appendix E.4) with pre-trained base model and value function trained for varied number of epochs. Each circle indicates performance of the baseline with specific hyperparameters (block length $\in \{1, 2, 4, 8, 16\}$ and # of candidates $\in \{2, 4, 8, 16, 32\}$); darker red indicates larger block length.

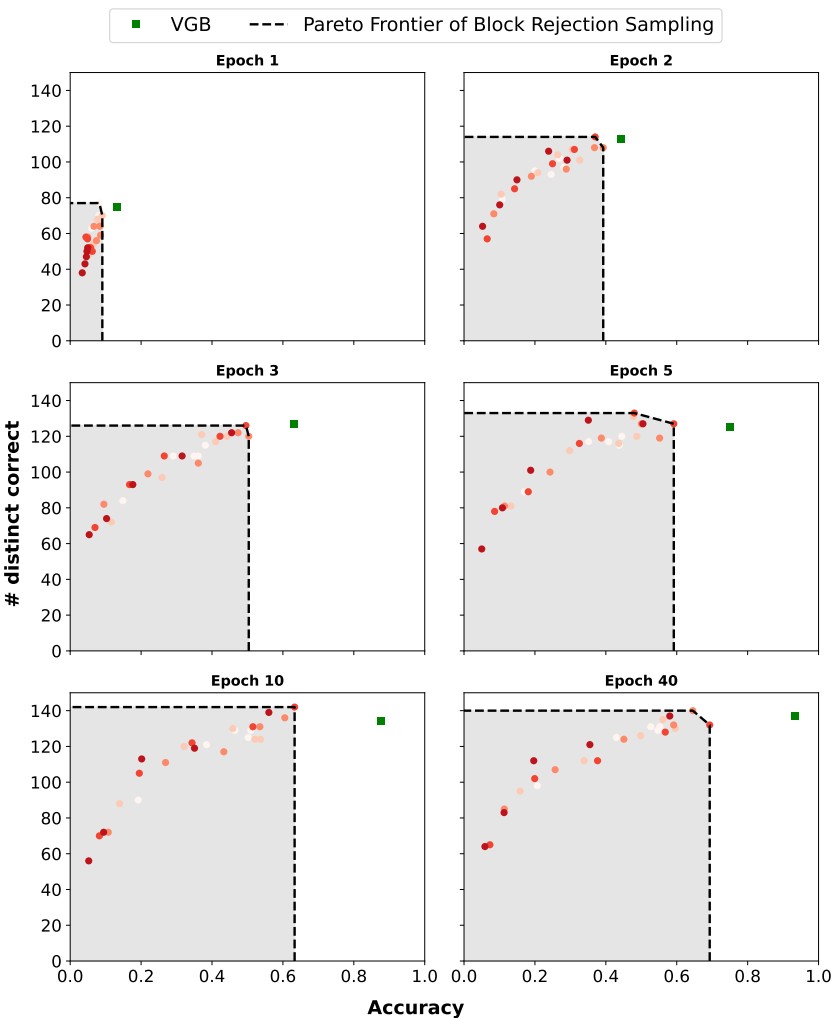

Figure 9: Accuracy (**x**) and diversity (**y**) of VGB vs. Block Rejection Sampling on Dyck grammar task (Appendix E.4) with pre-trained base model and value function trained for varied number of epochs. Each circle indicates performance of the baseline with specific hyperparameters (block length $\in \{1, 2, 4, 8, 16\}$ and # of candidates $\in \{2, 4, 8, 16, 32\}$); darker red indicates larger block length.

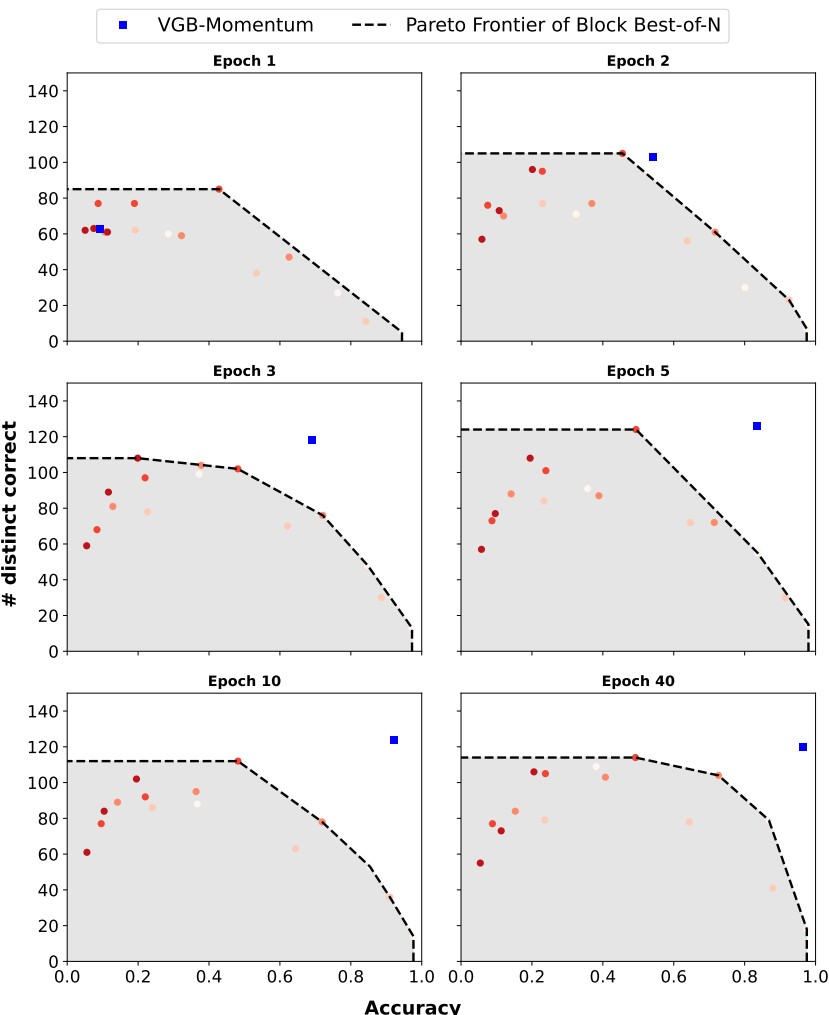

Figure 10: Accuracy (**x**) and diversity (**y**) of VGB-Momentum vs. Block Best-of-N on Dyck grammar task (Appendix E.4) with pre-trained base model and value function trained for varied number of epochs. Each circle indicates performance of the baseline with specific hyperparameters (block length $\in \{1, 2, 4, 8, 16\}$ and # of candidates $\in \{2, 4, 8\}$, to equalize query complexity); darker red indicates larger block length.

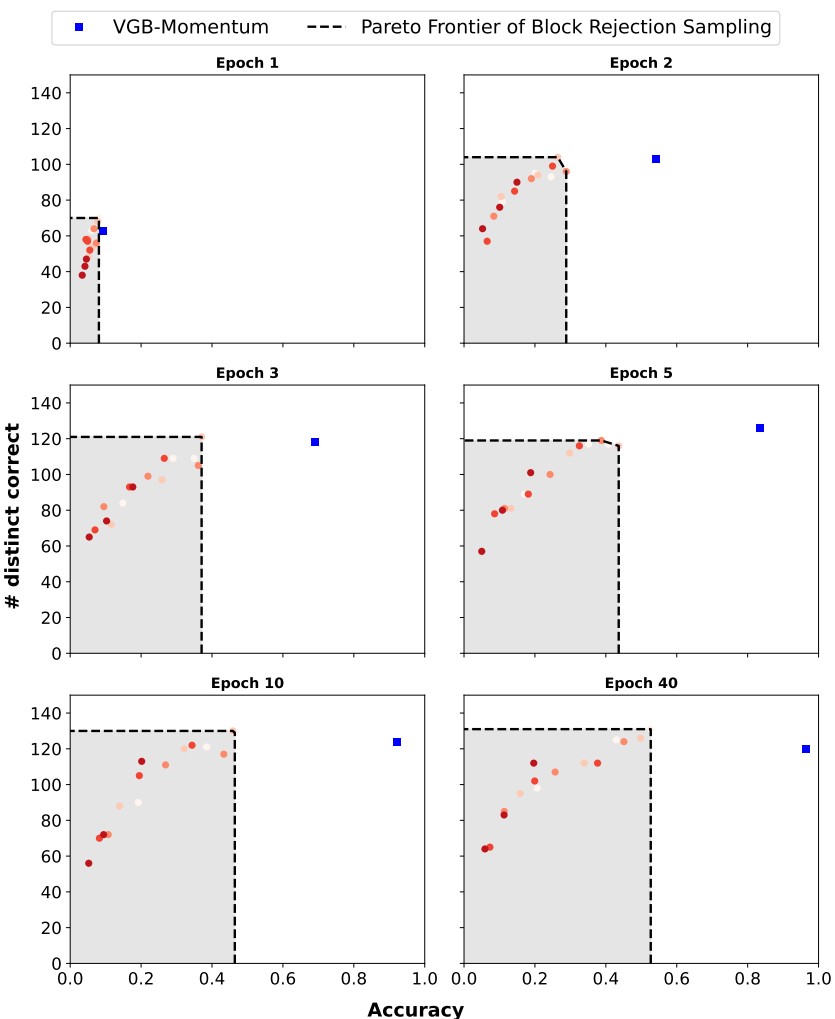

Figure 11: Accuracy (**x**) and diversity (**y**) of VGB-Momentum vs. Block Rejection Sampling on Dyck grammar task (Appendix E.4) with pre-trained base model and value function trained for varied number of epochs. Each circle indicates performance of the baseline with specific hyperparameters (block length $\in \{1, 2, 4, 8, 16\}$ and # of candidates $\in \{2, 4, 8\}$, to equalize query complexity); darker red indicates larger block length.

| Algorithm | # Candidates | Block Len. | Avg. Steps | Accuracy | # Distinct Correct |
|---|---|---|---|---|---|
| BlockLevelBoN | 8 | 1 | 136.000 | 0.975 | 18 |
| VGB-Momentum | | | 77.617 | 0.964 | 120 |
| BlockLevelBoN | 16 | 1 | 272.000 | 0.957 | 3 |
| BlockLevelBoN | 16 | 2 | 272.000 | 0.941 | 10 |
| BlockLevelBoN | 32 | 2 | 544.000 | 0.936 | 2 |
| VGB | | | 393.471 | 0.932 | 137 |
| BlockLevelBoN | 32 | 1 | 544.000 | 0.926 | 1 |
| BlockLevelBoN | 32 | 4 | 544.000 | 0.924 | 25 |
| BlockLevelBoN | 16 | 4 | 272.000 | 0.902 | 53 |
| BlockLevelBoN | 32 | 8 | 544.000 | 0.899 | 70 |
| BlockLevelBoN | 8 | 2 | 136.000 | 0.880 | 41 |
| BlockLevelBoN | 4 | 1 | 68.000 | 0.868 | 79 |
| BlockLevelBoN | 16 | 8 | 272.000 | 0.762 | 98 |
| BlockLevelBoN | 8 | 4 | 136.000 | 0.726 | 104 |
| BlockLevelRS | 32 | 8 | 544.000 | 0.693 | 132 |
| BlockLevelRS | 32 | 4 | 544.000 | 0.645 | 140 |
| BlockLevelBoN | 4 | 2 | 68.000 | 0.644 | 78 |
| BlockLevelRS | 32 | 2 | 544.000 | 0.596 | 130 |
| BlockLevelRS | 16 | 4 | 272.000 | 0.591 | 132 |
| BlockLevelBoN | 32 | 16 | 544.000 | 0.584 | 130 |
| BlockLevelRS | 32 | 16 | 544.000 | 0.580 | 137 |
| ActionLevelRS | | | 17.000 | 0.580 | 132 |
| BlockLevelRS | 16 | 8 | 272.000 | 0.568 | 128 |
| BlockLevelRS | 16 | 2 | 272.000 | 0.561 | 135 |
| BlockLevelRS | 16 | 1 | 272.000 | 0.552 | 131 |
| BlockLevelRS | 32 | 1 | 544.000 | 0.546 | 129 |
| BlockLevelRS | 8 | 1 | 136.000 | 0.527 | 131 |
| BlockLevelRS | 8 | 2 | 136.000 | 0.498 | 126 |
| BlockLevelBoN | 8 | 8 | 136.000 | 0.492 | 114 |
| BlockLevelRS | 8 | 4 | 136.000 | 0.451 | 124 |
| BlockLevelRS | 4 | 1 | 68.000 | 0.430 | 125 |
| BlockLevelBoN | 4 | 4 | 68.000 | 0.407 | 103 |
| BlockLevelBoN | 2 | 1 | 34.000 | 0.381 | 109 |
| BlockLevelRS | 8 | 8 | 136.000 | 0.377 | 112 |
| BlockLevelBoN | 16 | 16 | 272.000 | 0.355 | 116 |
| BlockLevelRS | 16 | 16 | 272.000 | 0.355 | 121 |
| BlockLevelRS | 4 | 2 | 68.000 | 0.339 | 112 |
| BlockLevelRS | 4 | 4 | 68.000 | 0.257 | 107 |
| BlockLevelBoN | 4 | 8 | 68.000 | 0.238 | 105 |
| BlockLevelBoN | 2 | 2 | 34.000 | 0.236 | 79 |
| BlockLevelRS | 2 | 1 | 34.000 | 0.207 | 98 |
| BlockLevelBoN | 8 | 16 | 136.000 | 0.206 | 106 |
| BlockLevelRS | 4 | 8 | 68.000 | 0.199 | 102 |
| BlockLevelRS | 8 | 16 | 136.000 | 0.197 | 112 |
| BlockLevelRS | 2 | 2 | 34.000 | 0.158 | 95 |
| BlockLevelBoN | 2 | 4 | 34.000 | 0.153 | 84 |
| BlockLevelRS | 2 | 4 | 34.000 | 0.114 | 85 |
| BlockLevelBoN | 4 | 16 | 68.000 | 0.113 | 73 |
| BlockLevelRS | 4 | 16 | 68.000 | 0.113 | 83 |
| BlockLevelBoN | 2 | 8 | 34.000 | 0.089 | 77 |
| BlockLevelRS | 2 | 8 | 34.000 | 0.073 | 65 |
| BlockLevelRS | 2 | 16 | 34.000 | 0.059 | 64 |
| BlockLevelBoN | 2 | 16 | 34.000 | 0.054 | 55 |
| $\pi_{\text{ref}}$ | | | 17.000 | 0.028 | 43 |

Table 1: Detailed results for accuracy/diversity tradeoff experiment in Dyck grammar task (Appendix E.4) with value function trained for $40$ epochs. BlockLevelBoN denotes Block Best-of-$N$ and BlockLevelRS denotes Block Rejection Sampling.

| | Average Steps (↓) | | Distributional Error (↓) | |
| Prefix | $\pi_{\texttt{ref}}$ | VGB | ActionLevelRS | VGB |
|---|---|---|---|---|
| ((((((((((([][]( | **276.303** | 314.998 | 0.485 | **0.327** |
| (((((([]((()((((([ | 403.232 | **330.082** | 0.536 | **0.179** |
| ()()((((((((((((( | **426.003** | 437.812 | 0.344 | **0.187** |
| (()(((((([]((((( | 442.204 | **396.146** | 0.416 | **0.283** |
| ()(()((((((((((( | 505.481 | **445.886** | 0.347 | **0.149** |
| (((()((((((()((( | 608.796 | **410.510** | 0.475 | **0.387** |
| (()(()(((((([([ | 639.817 | **422.092** | 0.364 | **0.272** |
| ((((()(()((((([ | 670.543 | **462.700** | 0.435 | **0.259** |
| ((([(((((([(()()( | 706.200 | **508.182** | 0.409 | **0.263** |
| ()((((((((((()((( | 739.038 | **446.574** | 0.265 | **0.176** |
| ((((()[([[(((()() | 852.620 | **378.564** | 0.239 | **0.193** |
| (((((()[(((()(((([( | 981.361 | **440.922** | 0.324 | **0.300** |

Table 2: Prefix-by-prefix breakdown of distributional comparison experiment for Dyck grammar task (Appendix E.4) at epoch 40, sorted by the average steps used by $\pi_{\texttt{ref}}$ to produce a correct generation. On each prefix, VGB achieves lower distributional error (compared to $\pi_{\texttt{ref}}$) than ActionLevelRS, and on 10 of 12 prefixes it uses fewer steps on average than $\pi_{\texttt{ref}}$. See Figure 12 for the aggregated results for all epochs.

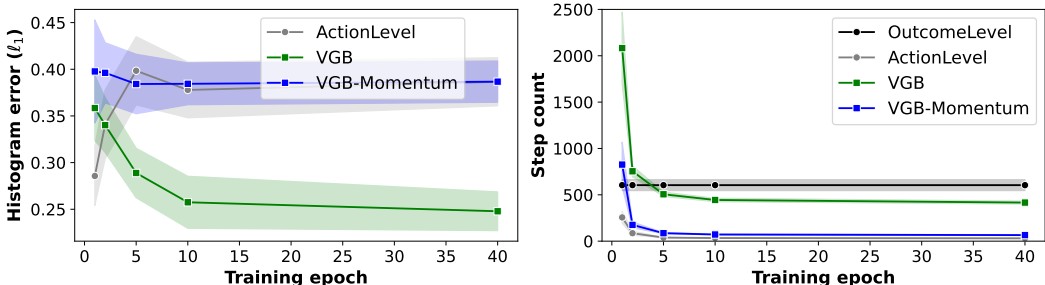

Figure 12: Distributional comparison between VGB, VGB-Momentum, and baselines in Dyck grammar task (Appendix E.4), across training epochs for value function, and using true rewards at final position. **Left:** Error of histogram of open-bracket positions (not distinguishing between square and round bracket types), using OutcomeLevelRS to estimate the ground truth. **Right:** Average step count. The experiment was independently repeated for 12 OOD prefixes (chosen randomly subject to the constraint that $\pi_{\texttt{ref}}$ has accuracy in $[0.01, 0.1]$), and we report mean and standard error—see Table 2 for the prefix-by-prefix results.

### E.5 PYTHON TEST CASE GENERATION TASK

**Task setup.** Let $\pi_{\texttt{ref}}$ be the pre-trained language model Qwen2.5-0.5B, which has a token space $\mathcal{A}$ of size roughly $150,000$. We set $H = 32$ and define the response space to be $\mathcal{A}^H$, so responses are fixed-length (however, in this task the prompts may have varying lengths). Following Botta et al. (2025), we define a distribution over prompts as follows. Pick a random three-letter function name (we draw from a pool of 10 possible names). The prompt first gives an in-context example of test case generation: (1) the Python implementation of the addition function $f(a, b) := a + b$, (2) a text prompt to produce a test case for $f$, and (3) a valid random test case for $f$. The prompt then gives the Python implementation of the list append function, but with the random function name, and asks for a test case for this function. We define a reward function that checks if the response is a valid Python test case (in the desired format).[20] See Figure 13 for an example prompt and a correct response.

---

[20]More precisely, the reward function parses each line of the response and returns reward 1 if there is exactly one valid response. Of course, there are many other reward functions that may be natural (e.g. at least one valid response) but we did not explore these possibilities. We used the parser developed by Botta et al. (2025) for this purpose.

```
1   def f(a, b):
2       return a + b
3
4   List 1 test case of the above function f, in one line:
5   assert f(7, 7) == 14
6
7   def ovs(l, item):
8       assert type(l) is list
9       l.append(item)
10      return l
11
12
13  List 1 test case of the above function ovs, in one line:
```

```
1   assert ovs([3,2], "asd") == [3,2,"asd"]
```

Figure 13: Example prompt and reward-1 completion for Python test case generation task (Appendix E.5). See Botta et al. (2025) for additional details on prompt distribution.

**Value function training setup.** Since this task has large alphabet size, training value networks on one-hot encodings of generations is infeasible. Instead, we train a single value network that can be applied to generations of any length $h \in [H]$. We generate a dataset by drawing $50,000$ prompts from the distribution described above; for each prompt $x$, we sample a response $y$ from $\pi_{\text{ref}}$ and evaluate the reward. We then pick a random index $h \sim \text{Unif}(\{1, 2, \ldots, |y|\})$, and compute $\frac{1}{h} \sum_{i=1}^{h} \phi(x, y_{1:i})$, where $\phi(x, y_{1:i})$ is the final-token hidden state at the penultimate layer of Qwen2.5-0.5B.[21]

Since the hidden states of Qwen2.5-0.5B have dimension $896$, the value network has input dimension $896$ as well. It is fully-connected with a single hidden layer with $8$ neurons; the first activation function is a ReLU and the second activation function is a sigmoid. We train the value network on the dataset described above for $100$ epochs using the Adam optimizer with learning rate $10^{-4}$, weight decay $10^{-4}$, and batch size $100$.

**Evaluation protocol and results.** We draw $1000$ fresh prompts and sample a completion for each prompt from VGB and ActionLevelRS, using the true rewards at the last layer. We also sample a completion for each prompt from $\pi^\star$. We compare the empirical distributions produced by VGB and ActionLevelRS to that of $\pi^\star$ using three metrics: (1) the histogram of *lengths* of the final Python list in each generated test case, (2) the fraction of *strings* in the final lists in the test cases (the most common object type is an integer, and the second most common is a string; the remaining types have negligible occurrences, so this essentially tracks the histogram of object types), and (3) the fraction of *distinct* final lists in the generated test cases. For (1) and (2) we compute the error of each algorithm compared to $\pi^\star$; (3) is a measure of diversity, and from examination we saw that $\pi^\star$ had greater diversity than both algorithms. We repeat the experiment for 10 independent seeds (the dataset generation and value function training are independent as well), and in Table 3 we report the mean and standard error across seeds for each metric.

### E.6  LETTER AVOIDANCE TASK

**Task setup.** Let $\pi_{\text{ref}}$ be the pre-trained language model Qwen2.5-0.5B, as in the preceding task. We set $H = 32$ and define the response space to be $\mathcal{A}^H$, where $\mathcal{A}$ is the token space for the language model. We fix a prompt, which asks the model to generate a sentence without using the letter "e" (Figure 14), and we define a reward function $r^\star$ that checks (1) the letter "e" was not used (in the response), and (2) the EOS token for the language model was not used. We find that $\pi_{\text{ref}}$ only produces 6 reward-1 responses out of $10,000$.

---

[21]It has been previously observed by Botta et al. (2025) that the final layer does not necessarily lead to the best performance. We did not tune the layer index, since our goal is to compare different guided sampling algorithms, not to optimize the end-to-end pipeline.

| Algorithm | Length hist. error ($\downarrow$) | Type hist. error ($\downarrow$) | Frac. distinct generations ($\uparrow$) |
|---|---|---|---|
| VGB | **69.7500** (7.9782) | 0.0172 (0.0018) | **0.3741** (0.0033) |
| ActionLevelRS | 92.2000 (6.6375) | **0.0157** (0.0031) | 0.3700 (0.0036) |

Table 3: Distributional comparison of VGB and ActionLevelRS for Python test case generation task (Appendix E.5). For each metric, we evaluated based on 1000 reward-1 generations from each algorithm. We repeated the experiment with 20 independent seeds and report the mean and standard error (in parentheses). While ActionLevelRS outperforms VGB on the histogram of types, note that the discrepancy is within the standard error. We hypothesize that VGB achieves a better distribution over test case lengths since there is likely substantial heterogeneity in the difficulty of generating different lengths, leading ActionLevelRS to have a biased distribution.

```
1  Generate a one-sentence story without using the letter 'e':
```

```
1  Vicky is having a party and wants to buy two hamburgars. A sub
      sandwich costs $1.00 and a ravioli costs $3
```

Figure 14: Prompt for letter avoidance task (Appendix E.6), and an example generation from naive locally constrained decoding (i.e., ActionLevelRS).

**Value function.** In this task, we do not train a value function. Instead, for $\alpha \in \{0.1, 0.2, 0.3, 0.4, 0.5\}$, we define

$$\widehat{V}_\alpha(x, y) := \alpha^{H-|y|} r^\star(x, y).$$

Note that with ActionLevelRS, all of these value functions are equivalent, and induce a standard algorithm for constrained generation called *locally constrained decoding* (Scholak et al., 2021; Lipkin et al., 2025): at each generation step $(x, y_{1:h})$, the next token $y_{h+1}$ is drawn from $\pi_{\mathsf{ref}}(\cdot \mid x, y_{1:h})$ conditioned on the event $r^\star(x, y_{1:h+1}) = 1$. However, these value functions are not equivalent for VGB, as they essentially vary the probability that VGB will backtrack.

**Evaluation protocol and results.** With the prompt shown in Figure 14 and for each $\alpha \in \{0.1, 0.2, 0.3, 0.4, 0.5\}$, we draw 1000 responses from VGB using $\widehat{V}_\alpha$. We also draw 1000 responses from ActionLevelRS using $\widehat{V}_{0.1}$ (as we discussed above, the choice of $\alpha$ is irrelevant for ActionLevelRS). Recall that in our implementation of ActionLevelRS with large alphabets, we draw $K = 32$ candidate tokens at each generation step, and sample proportional to their estimated values. Since in this setting the estimated value can be exactly 0, this can be ill-posed. We implement two natural heuristics for dealing with this: the first is to simply draw $K = 32$ more candidate tokens, which increments the step count (we simply refer to this algorithm as ActionLevelRS, since it is roughly comparable to sending $K \to \infty$); the second is to restart the response from the beginning (we refer to this algorithm as ActionLevelRS-Reset). We remark that the first method can get "stuck" if the base model, conditioned on some prefix, puts almost all mass on tokens containing the letter "e". This is why the step count of ActionLevelRS has higher mean and standard error than ActionLevelRS-Reset (Figure 16).

We evaluate VGB against ActionLevelRS and ActionLevelRS-Reset using two metrics. The first metric is winrate according to GPT-4o-mini: given 1000 completions from two algorithms, we pair up the completions and ask GPT-4o-mini to judge which is more coherent, using the prompt template in Figure 17. We present the two completions in a random order to avoid (positive or negative) primacy bias. The second metric is the average horizon-normalized log-probability of the completions (conditioned on the prompt) as computed by Qwen2.5-1.5B. See Figs. 3 and 15 for the comparisons with ActionLevelRS and ActionLevelRS-Reset respectively. For completeness, we also plot the average step count of each algorithm; see Figure 16.

To quantify how the benefits scale in $H$, we repeat the experiment for $H \in \{8, 16\}$ (this time only evaluating against ActionLevelRS for computational reasons). As shown in Figure 18, we broadly see that VGB has larger benefits for larger $H$, particularly in the (horizon-normalized) log-

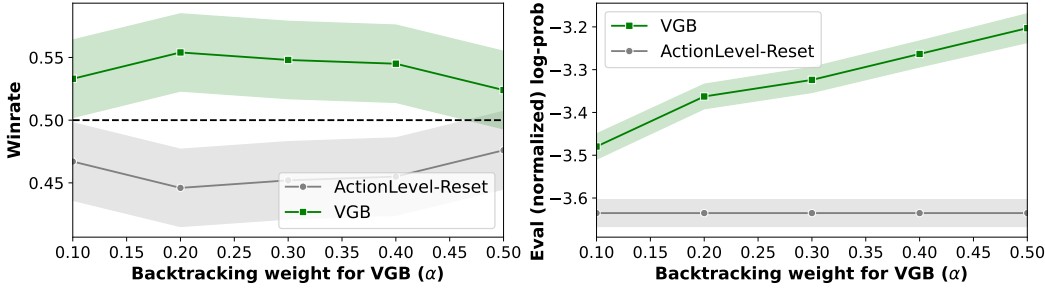

Figure 15: Comparison of VGB against `ActionLevelRS-Reset` for letter avoidance task (Appendix E.6), with $H = 32$ and varied backtracking weight $\alpha$ for VGB. **Left:** Winrate of VGB against `ActionLevelRS-Reset` under pairwise comparison of $1000$ responses from each algorithm, using `GPT-4o-mini` to judge writing quality. We report the $95\%$ binomial confidence intervals. **Right:** Horizon-normalized log-probabilities of generated responses, evaluated by `Qwen-2.5-1.5B`; for each algorithm, we report the mean and standard error over $1000$ responses.

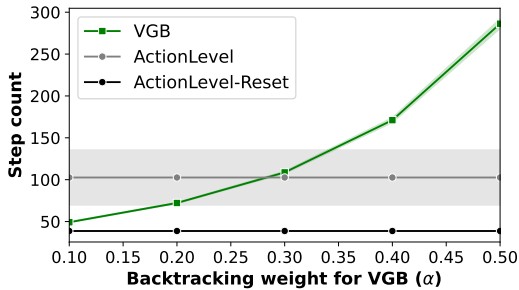

Figure 16: Step count comparison for VGB against `ActionLevelRS` and `ActionLevelRS-Reset` for letter avoidance task (Appendix E.6), with $H = 32$ and varied backtracking weight $\alpha$ for VGB.

probabilities—the winrate increases from $H = 8$ to $H = 16$ but seems roughly comparable between $H = 16$ and $H = 32$.

### E.7 Additional Exploratory Experimental Results

**Value function parametrization matters.** In the Dyck grammar task, we saw that the single-line change from `ActionLevelRS` to VGB—adding a probability of backtracking—robustly improves *accuracy* across training epochs. In this case, the value function was trained on the concatenation of one-hot encodings of the partial generation. In the code generation task, this parametrization is infeasible due to the large token space. Instead, the value function is trained on the mean-pooling of the hidden states (in the base model) of each token in the partial generation. Again, VGB has robustly higher accuracy than `ActionLevelRS`. But if the value function is trained on the hidden state of just the *last* token in the partial generation, then this effect disappears (Figure 19). While in principle the last hidden state should encode all relevant information about the partial generation, this finding suggests that in practice it may heavily attend to recent tokens —thus, some benefits of backtracking-based algorithms can be contingent on exactly what features they are trained on.

**Momentum improves the query complexity of VGB.** A classical method for accelerating Markov chains like VGB is to add *momentum*, which decreases the likelihood that the chain will "switch directions" (from going down the tree to going up, and vice versa) while preserving the stationary distribution. Hayes and Sinclair (Hayes & Sinclair, 2010) theoretically investigated the benefits of momentum specifically for the Sinclair-Jerrum walk, and showed that it provably improves runtime if the approximate counts are $(1 \pm o(1))$-accurate (specifically, the accelerated chain mixes to $\pi^\star$ in time $\varepsilon_V H^2$ under Assumption 4.1). We implement this Markov chain (with the practical tweak of stopping it as soon as it reaches a leaf), and find that it improves the query complexity of VGB by nearly an order of magnitude, while achieving comparable (or even higher) accuracy (Table 1).

```
1  You are a writing quality judge. You will be given a pair of
       text fragments. Say ONLY 'A' if the first fragment is more
        coherent, or ONLY 'B' if the second fragment is more
       coherent.
2  Fragment A: ______________
3
4  Fragment B: ______________
```

Figure 17: Template of prompt fed to `GPT-4o-mini` for winrate computation in letter avoidance task (Appendix E.6).

However, it does not enjoy the same distributional benefits as `VGB` (Figure 12), suggesting a potentially fundamental statistical-computational tradeoff.

**Backtracking can help throughout generation.** One might wonder whether e.g. backtracking only helps at the last position, where the value function is likely most accurate (or even perfect, if the true rewards are being used). In Figure 20, we demonstrate that for the Dyck grammar task, for a prefix where `ActionLevelRS` achieves average reward roughly $0.5$, it often makes mistakes fairly early in the generation sequence: the most common error indices are $\{4, 5, 6, 8\}$ out of 17. Moreover, Figure 20 shows that *these errors can be detected early*: conditioned on a particular index for the first error in a generation, we see that the average estimated value at subsequent indices rapidly drops off (compared to the average estimated value for sequences where `ActionLevelRS` makes no error). Thus, it is (at least at a population level) possible for backtracking to be useful before reaching the leaves of the generation tree.

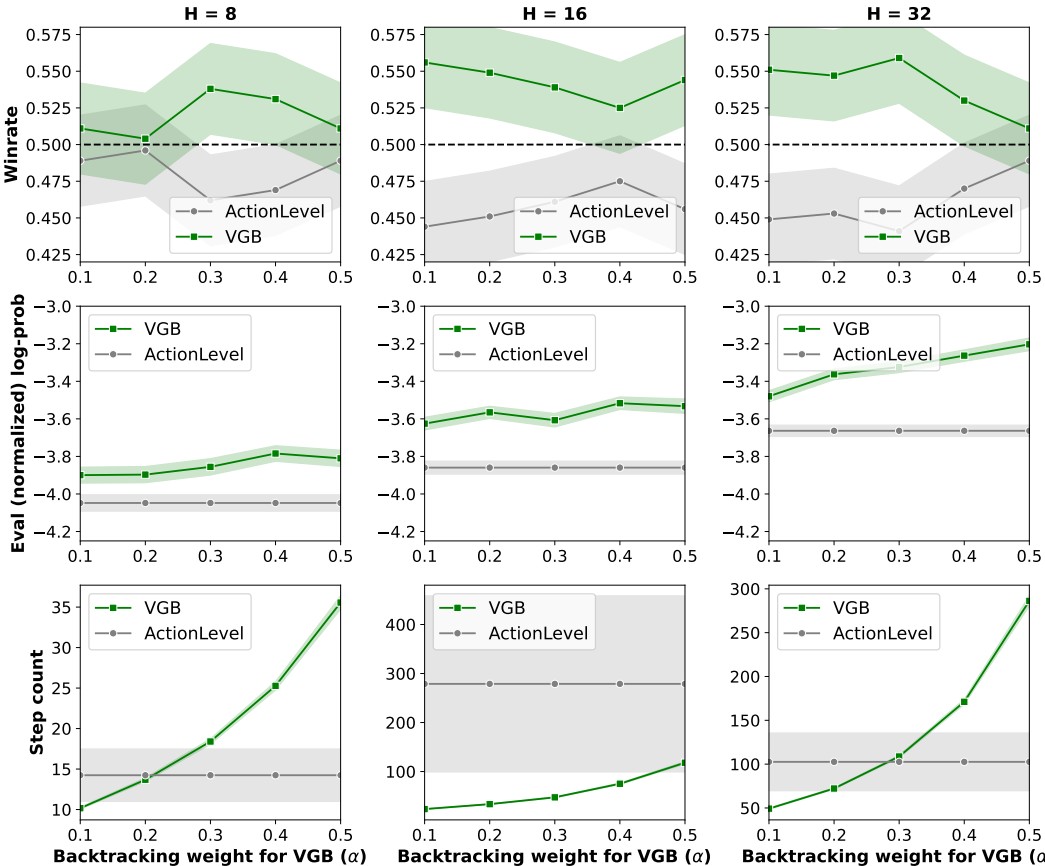

Figure 18: Comparison of VGB against ActionLevelRS for letter avoidance task (Appendix E.6) with varied horizon $H \in \{8, 16, 32\}$. **Top row:** Winrate of VGB against ActionLevelRS under pairwise comparison of 1000 responses from each algorithm, using GPT-4o-mini to judge writing quality. We report the 95% binomial confidence intervals. **Middle row:** Average horizon-normalized log-probabilities of generated responses, evaluated by Qwen-2.5-1.5B; for each algorithm, we report the mean and standard error over 1000 responses. **Bottom row:** Step count comparison; we note that the average step count for ActionLevelRS is higher at $H = 16$ than $H = 32$ due to an outlier generation that required 148251 steps.

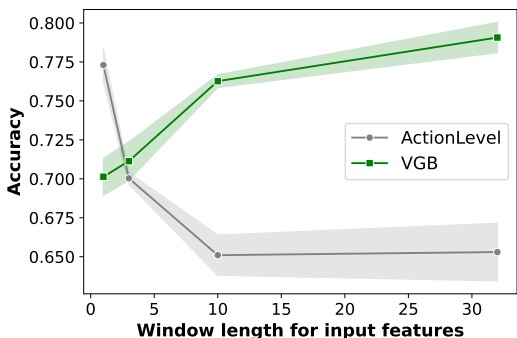

Figure 19: Window length of mean-pooled input features for value function (**x**) against accuracy of `VGB` and `ActionLevelRS` (**y**) for Python test case generation task (Appendix E.5). We repeat the experiment for 3 independent seeds and report the mean and standard error. All algorithms generate at most 32 new tokens per prompt, so window length 32 corresponds to mean-pooling over all tokens generated so far (the method used for the main experiment in Appendix E.5). We never pool over the tokens in the prompt.

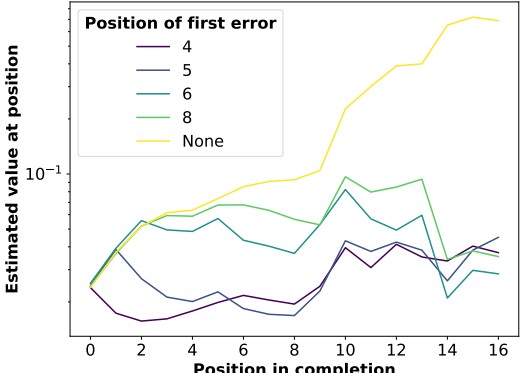

Figure 20: Average estimated values (**y**) in the Dyck grammar task (Appendix E.4), by position of first error (**hue**) and position in completion (**x**). We pick a single prefix and value function trained for 20 epochs. We generate 1000 completions using `ActionLevelRS` and record the estimated value at each step of each generation, as well as the first position in the generation at which a mistake has been made (i.e. it becomes impossible to complete to a valid string); if the generation is correct, we record this as "None". For each of the most common error positions $\{4, 5, 6, 8, \text{None}\}$, we plot the average estimated value at each position of the completion. We see that errors early in the completion can be witnessed (at least, at a population level) well before the final position.

# Part II

# Proofs and Supporting Results

## F    PRELIMINARIES

In this section, we collect standard results that will be used in our proofs.

### F.1    PROBABILITY THEORY

**Definition F.1** (Probability divergences). *Let $p, q$ be distributions on a common domain $\mathcal{X}$, with $p \ll q$. The $\chi^2$-divergence from $p$ to $q$ is defined as*

$$D_{\chi^2}(p \parallel q) := \mathop{\mathbb{E}}_{\omega \sim q} \left[ \left( \frac{p(\omega)}{q(\omega)} \right)^2 \right] - 1.$$

*The KL-divergence from $p$ to $q$ is defined as*

$$D_{\mathsf{KL}}(p \parallel q) := \mathop{\mathbb{E}}_{\omega \sim p} \left[ \log \frac{p(\omega)}{q(\omega)} \right].$$

*The total variation distance between $p$ and $q$ is defined as*

$$D_{\mathsf{TV}}(p, q) = \sup_A |p(A) - q(A)| = \frac{1}{2} \mathop{\mathbb{E}}_{\omega \sim q} \left| \frac{p(\omega)}{q(\omega)} - 1 \right|.$$

**Proposition F.1** (Divergence inequalities (Sason & Verdú, 2016)). *For any two distributions $p, q$ on a common domain, with $p \ll q$, we have*

$$2 \cdot D_{\mathsf{TV}}(p, q)^2 \leq D_{\mathsf{KL}}(p \parallel q) \leq D_{\chi^2}(p \parallel q).$$

**Lemma F.1.** *Let $n \in \mathbb{N}$ and $\Delta \in (0, 1/2)$. Then*

$$D_{\mathsf{TV}}(\mathrm{Bin}(n, 1/2), \mathrm{Bin}(n, 1/2 + \Delta)) = \Omega(\Delta \sqrt{n}).$$

**Proof.** Let $p := \mathrm{Bin}(n, 1/2)$ and $q := \mathrm{Bin}(n, 1/2 + \Delta)$. Then

$$
\begin{aligned}
D_{\mathsf{TV}}(p, q) &\gtrsim \mathop{\mathbb{E}}_{X \sim p} \left| 1 - \frac{q(X)}{p(X)} \right| \\
&= \mathop{\mathbb{E}}_{X \sim p} \left| 1 - (1 + \Delta)^X (1 - \Delta)^{n-X} \right| \\
&= \mathop{\mathbb{E}}_{X \sim p} \left| 1 - (1 - \Delta^2)^X (1 - \Delta)^{n-2X} \right| \\
&\geq \mathop{\mathbb{E}}_{X \sim p} \left[ 1 - (1 - \Delta)^{n-2X} \right] \mathbb{I}[X \leq n/2] \\
&\geq \mathop{\mathbb{E}}_{X \sim p} \left[ 1 - e^{-(n-2X)\Delta} \right] \mathbb{I}[X \leq n/2] \\
&\gtrsim \mathop{\mathbb{E}}_{X \sim p} \min((n - 2X)\Delta, 1) \mathbb{I}[X \leq n/2]
\end{aligned}
$$

using the fact that $1 - e^{-t} \geq \frac{1}{2} \min(t, 1)$ for all $t \geq 0$. It follows that

$$D_{\mathsf{TV}}(p, q) \gtrsim \mathop{\mathbb{E}}_{X \sim p} \min(\Delta \sqrt{n}, 1) \mathbb{I}[X \leq n/2 - \sqrt{n}] \gtrsim \min(\Delta \sqrt{n}, 1)$$

where the final inequality is by a standard anti-concentration bound for the central Binomial distribution (Matoušek & Vondrák, 2001, Proposition 7.3.2). $\qquad\square$

**Lemma F.2.** *Let $p, q$ be distributions on a finite domain $\mathcal{X}$ and let $\mathcal{E} \subseteq \mathcal{X}$ be an event. Then*

$$D_{\mathsf{TV}}(p|_{\mathcal{E}}, q|_{\mathcal{E}}) \leq \frac{D_{\mathsf{TV}}(p, q)}{q(\mathcal{E})}$$

*and*

$$D_{\chi^2}(p|_{\mathcal{E}} \parallel q|_{\mathcal{E}}) \leq \frac{D_{\chi^2}(p \parallel q)}{q(\mathcal{E}) - 2\sqrt{q(\mathcal{E}) D_{\chi^2}(p \parallel q)}}.$$

**Proof.** We have

$$
\begin{aligned}
2D_{\mathsf{TV}}(p|_{\mathcal{E}}, q|_{\mathcal{E}}) &= \sum_{x \in \mathcal{E}} \left| \frac{p(x)}{p(\mathcal{E})} - \frac{q(x)}{q(\mathcal{E})} \right| \\
&\leq \frac{1}{q(\mathcal{E})} \sum_{x \in \mathcal{E}} |p(x) - q(x)| + \left| \frac{1}{p(\mathcal{E})} - \frac{1}{q(\mathcal{E})} \right| \sum_{x \in \mathcal{E}} p(x) \\
&= \frac{1}{q(\mathcal{E})} \sum_{x \in \mathcal{E}} |p(x) - q(x)| + \frac{|p(\mathcal{E}) - q(\mathcal{E})|}{q(\mathcal{E})} \\
&= \frac{1}{q(\mathcal{E})} \sum_{x \in \mathcal{E}} |p(x) - q(x)| + \frac{|p(\overline{\mathcal{E}}) - q(\overline{\mathcal{E}})|}{q(\mathcal{E})} \\
&\leq \frac{1}{q(\mathcal{E})} \sum_{x \in \mathcal{E}} |p(x) - q(x)| + \frac{1}{q(\mathcal{E})} \sum_{x \notin \mathcal{E}} |p(x) - q(x)| \\
&= \frac{2}{q(\mathcal{E})} D_{\mathsf{TV}}(p, q)
\end{aligned}
$$

which proves the first claim. For the second claim, we note that

$$
\begin{aligned}
D_{\chi^2}(p|_{\mathcal{E}} \parallel q|_{\mathcal{E}}) &= \frac{1}{q(\mathcal{E})} \sum_{x \in \mathcal{E}} q(x) \left( 1 - \frac{q(\mathcal{E})}{p(\mathcal{E})} \frac{p(x)}{q(x)} \right)^2 \\
&= \frac{q(\mathcal{E})}{p(\mathcal{E})^2} \sum_{x \in \mathcal{E}} q(x) \left( 1 - \frac{p(x)}{q(x)} \right)^2 - \left( 1 - \frac{q(\mathcal{E})}{p(\mathcal{E})} \right)^2 \\
&\leq \frac{q(\mathcal{E})}{p(\mathcal{E})^2} D_{\chi^2}(p \parallel q).
\end{aligned}
$$

Now observe that by the data processing inequality, $\frac{(p(\mathcal{E}) - q(\mathcal{E}))^2}{q(\mathcal{E})} \leq D_{\chi^2}(p \parallel q)$, and therefore $|p(\mathcal{E}) - q(\mathcal{E})| \leq \sqrt{q(\mathcal{E}) D_{\chi^2}(p \parallel q)}$. It follows that $p(\mathcal{E}) \geq q(\mathcal{E}) - \sqrt{q(\mathcal{E}) D_{\chi^2}(p \parallel q)}$, and so

$$
D_{\chi^2}(p|_{\mathcal{E}} \parallel q|_{\mathcal{E}}) \leq \frac{q(\mathcal{E}) D_{\chi^2}(p \parallel q)}{(q(\mathcal{E}) - \sqrt{q(\mathcal{E}) D_{\chi^2}(p \parallel q)})^2} \leq \frac{D_{\chi^2}(p \parallel q)}{q(\mathcal{E}) - 2\sqrt{q(\mathcal{E}) D_{\chi^2}(p \parallel q)}}
$$

where the first inequality requires $D_{\chi^2}(p \parallel q) \leq q(\mathcal{E})$ (the final bound is vacuous if this is false, so it holds unconditionally). $\qquad\square$

**Lemma F.3.** *Let $T \in \mathbb{N}$ and $\varepsilon_{1:T}, \delta > 0$. and let $Z_1, \ldots, Z_T$ be nonnegative random variables with $\Pr[Z_i < \delta] \leq \varepsilon_i$ for each $i \in [T]$. Then*

$$
\Pr\left[ \frac{1}{T} \sum_{i=1}^{T} Z_i < \frac{\delta}{2} \right] \leq \frac{2}{T} \sum_{i=1}^{T} \varepsilon_i.
$$

**Proof.** Define the random variable $N := \#\{i : Z_i < \delta\}$. Then $\mathbb{E}[N] \leq \sum_{i=1}^{T} \varepsilon_i$, so $\Pr[N > T/2] \leq \frac{2}{T} \sum_{i=1}^{T} \varepsilon_i$. Moreover, if $N \leq T/2$ then $\frac{1}{T} \sum_{i=1}^{T} Z_i \geq \delta/2$ as needed. $\qquad\square$

### F.1.1 MARKOV CHAINS

We present several basic definitions and results from the theory of Markov chains. See Levin et al. (2009) or other standard references for additional background.

**Definition F.2.** *A Markov chain $\mathbb{P}$ on state space $\mathcal{X}$ is* reversible *with stationary distribution $\mu$ if, for all $u, v \in \mathcal{X}$, it holds that*

$$
\mathbb{P}(v \mid u)\mu(u) = \mathbb{P}(u \mid v)\mu(v).
$$

**Definition F.3.** *A Markov chain $\mathbb{P}$ on state space $\mathcal{X}$ is* irreducible *if, for all $u, v \in \mathcal{X}$, there exists $t \geq 0$ such that $\mathbb{P}^t(u \mid v) > 0$. The Markov chain is* aperiodic *if, for every $u \in \mathcal{X}$, the GCD of $\{t \geq 1 : \mathbb{P}(u \mid u) > 0\}$ is 1. The Markov chain is* ergodic *if it is both irreducible and aperiodic.*

Conductance is a standard measure of the bottlenecks in a Markov chain, and is formally defined as follows.

**Definition F.4** (Conductance). *The conductance of a Markov chain $\mathbb{P}$ with stationary distribution $\mu$ is defined as*

$$\Phi := \min_{S \subset \mathcal{T}: 0 < \mu(S) \leq 1/2} \Phi_S$$

*where*

$$\Phi_S := \frac{\sum_{u \in S, v \notin S} \mu(u)\mathbb{P}(v \mid u)}{\sum_{u \in S} \mu(u)}.$$

We recall the following fundamental result that relates the conductance with the mixing time of the Markov chain. This result can be derived from Cheeger's inequality, which relates the conductance of a Markov chain to its spectral gap; see e.g. (Montenegro & Tetali, 2005, Chapter 3) or Theorem 2.1 and the proof of Theorem 1.2 in Liu (2023).

**Theorem F.1** (Conductance implies mixing in $\chi^2$). *For any ergodic reversible Markov chain $\mathbb{P}$ with stationary distribution $\mu$ and conductance $\Phi$, for any initial state $z$ and any $t \in \mathbb{N}$, it holds that*

$$D_{\chi^2}\big(\mathbb{P}^t(\cdot \mid z) \,\|\, \mu\big) \lesssim e^{-\Phi^2 t/2} \frac{1}{\mu(z)}. \tag{14}$$

## F.2 KL-Regularized Reinforcement Learning

In the KL-regularized setting introduced in Section 2, the optimal KL-regularized value function $Q_\beta^\star$ (e.g., Rafailov et al. (2024); Xie et al. (2025)) is often defined via the following inductive procedure. For step $H$, we set $Q_\beta^\star(x, y_{1:H}) := r^\star(x, y_{1:H})$. Then, for $h = H - 1, \ldots, 1$:

$$Q_\beta^\star(x, y_{1:h}) = \sup_{p \in \Delta(\mathcal{A})} \left( \mathbb{E}_{y_{h+1} \sim p}\big[Q_\beta^\star(x, y_{1:h+1})\big] - \beta \cdot D_{\mathsf{KL}}(p \,\|\, \pi_{\mathsf{ref}}(y_{h+1} = \cdot \mid x, y_{1:h})) \right) \tag{15}$$

$$= \beta \log\left( \mathbb{E}_{y_{h+1} \sim \pi_{\mathsf{ref}}(\cdot \mid x, y_{1:h})} \big[\exp\big(\beta^{-1} Q_\beta^\star(x, y_{1:h+1})\big)\big] \right). \tag{16}$$

Eq. (16) can be viewed as a soft Bellman update; as $\beta \to 0$, this update recovers standard dynamic programming (over those actions for which $\pi_{\mathsf{ref}}(y_{h+1} = \cdot \mid x, y_{1:h}) > 0$). The optimal policy satisfies $\pi_\beta^\star(y_h \mid x, y_{1:h-1}) \propto \pi_{\mathsf{ref}}(y_h \mid x, y_{1:h-1}) \cdot \exp\big(\beta^{-1} Q_\beta^\star(x, y_{1:h})\big)$, and we further define $A_\beta^\star(x, y_{1:h}) := Q_\beta^\star(x, y_{1:h}) - Q_\beta^\star(x, y_{1:h-1})$ as the optimal *advantage function*. It can be checked that with the above definition, $Q_\beta^\star$ also satisfies

$$Q_\beta^\star(x, y_{1:h}) = \beta \log\left( \mathbb{E}_{y_{h+1:H} \sim \pi_{\mathsf{ref}}(\cdot \mid x, y_{1:h})} \big[\exp\big(\beta^{-1} r^\star(x, y_{1:H})\big)\big] \right). \tag{17}$$

The following standard fact relates KL-regularized regret to KL-divergence to the optimal policy.

**Lemma F.4** (See (Foster et al., 2025, Lemma F.4)). *Fix any $x \in \mathcal{X}$ and $\widehat{\pi} : \mathcal{X} \to \Delta(\mathcal{Y})$. It holds that*

$$J_\beta(\pi^\star; x) - J_\beta(\widehat{\pi}; x) = \beta \cdot D_{\mathsf{KL}}(\widehat{\pi}(\cdot \mid x) \,\|\, \pi^\star(\cdot \mid x)).$$

## F.3 Value Functions

The following lemma states the key property of the true value function $V_{\mathsf{tilt}}^\star$, which relates it to the next-action conditional probabilities of $\pi^\star$. This identity has been observed previously by, e.g., Yang & Klein (2021). We will use this fact throughout the paper.

**Lemma F.5.** *For any $x \in \mathcal{X}$, $h \in [H]$, and $y_{1:h} \in \mathcal{A}^h$, it holds that*

$$\pi^\star(y_h \mid x, y_{1:h-1}) = \pi_{\mathsf{ref}}(y_h \mid x, y_{1:h-1}) \frac{V_{\mathsf{tilt}}^\star(x, y_{1:h})}{V_{\mathsf{tilt}}^\star(x, y_{1:h-1})}$$

**Proof.** We have

$$
\begin{aligned}
\pi^\star(y_{1:h} \mid x) &= \sum_{y_{h+1:H} \in \mathcal{A}^{H-h}} \pi^\star(y_{1:H} \mid x) \\
&= \frac{1}{V^\star_{\mathsf{tilt}}(x)} \sum_{y_{h+1:H} \in \mathcal{A}^{H-h}} \pi_{\mathsf{ref}}(y_{1:H} \mid x)\tau(x, y_{1:H}) \\
&= \frac{\pi_{\mathsf{ref}}(y_{1:h} \mid x)}{V^\star_{\mathsf{tilt}}(x)} \mathbb{E}^{\pi_{\mathsf{ref}}}[\tau(x, y_{1:H}) \mid y_{1:h}] \\
&= \frac{\pi_{\mathsf{ref}}(y_{1:h} \mid x)}{V^\star_{\mathsf{tilt}}(x)} V^\star_{\mathsf{tilt}}(x, y_{1:h}).
\end{aligned}
$$

It follows that

$$
\pi^\star(y_h \mid x, y_{1:h-1}) = \frac{\pi^\star(y_{1:h} \mid x)}{\pi^\star(y_{1:h-1} \mid x)} = \pi_{\mathsf{ref}}(y_h \mid x, y_{1:h-1}) \frac{V^\star_{\mathsf{tilt}}(x, y_{1:h})}{V^\star_{\mathsf{tilt}}(x, y_{1:h-1})}
$$

as claimed. $\qquad \square$

### F.4  Approximate Rejection Sampling

We present a guarantee due to Foster et al. (2025) for (a slight variant of) rejection sampling, which is used to implement OutcomeLevelRS, and is used as a subroutine in ActionLevelRS and the full (large-$|\mathcal{A}|$) version of VGB.

The only difference between this algorithm and the standard rejection sampling algorithm is that it automatically estimates the normalization constant for the target distribution so that the rejection threshold can be set purely as a function of the density ratio between the target and base.

The basic setting is that we have sample access to a base measure $\mu \in \Delta(\mathcal{Z})$ and query access to a tilt function $g : \mathcal{Z} \to \mathbb{R}_{\geq 0}$, and we would like to sample from the distribution with density proportional to $\mu(z)g(z)$. The algorithm pseudocode is given in Algorithm 5 and the guarantee is given in Proposition F.2. We remark that Foster et al. (2025) consider the case where $g(z) > 0$ (specifically, they write $g(z) = \exp(\beta^{-1}f(z))$ for a function $f : \mathcal{Z} \to \mathbb{R}$ and temperature parameter $\beta > 0$, motivated by KL-regularized optimization), but the algorithm and guarantee extend unchanged to the setting where $g(z) \geq 0$.

---

**Algorithm 5** RejectionSampling; variant of SoftmaxRejectionSampling (Foster et al., 2025)

**input:** Function $g : \mathcal{Z} \to \mathbb{R}$, base measure $\mu \in \Delta(\mathcal{Z})$, rejection threshold $M > 0$, failure probability $\delta \in (0, 1)$.

1: Let $N := 4M \log(4\delta^{-1})$.
   /* Estimate normalization constant */
2: Sample $z_1, \dots, z_N \sim \mu$ i.i.d.
3: Set $\widehat{Z} := \frac{1}{N} \sum_{i=1}^N g(z_i)$.
   /* Rejection sampling */
4: **for** iteration $i = 1, 2, \dots, N$ **do**
5: $\quad$ Sample $z \sim \mu(\cdot)$ and $\xi \sim \mathsf{Ber}(\min(g(z)/\widehat{Z}M, 1))$.
6: $\quad$ If $\xi = 1$, **return** $z$.
7: **return** $z \sim \mu(\cdot)$. $\qquad\qquad$ // Failure event; occurs with low probability.

---

**Proposition F.2** (Guarantee for Algorithm 5; see Foster et al. (2025)). *Let $g : \mathcal{Z} \to \mathbb{R}$ be given, and define*

$$
\pi(z) \propto \mu(z) \cdot g(z), \quad \text{and} \quad C_\infty := \left\| \frac{\pi(\cdot)}{\mu(\cdot)} \right\|_\infty. \tag{18}
$$

*Fix $\delta \in (0, 1)$, and fix any $M \geq 4C_\infty$. Then there is an event $\mathcal{E}$ that occurs with probability at least $1 - \delta$, such that the output $z \in \mathcal{Z}$ of Algorithm 5 with inputs $(g, \mu, M, \delta)$ satisfies*

$$
\mathbb{P}[z = \cdot \mid \mathcal{E}] = \pi(\cdot).
$$

*As a consequence, the unconditional law of $z$ satisfies $D_{\mathsf{TV}}(\widehat{\pi}, \pi) \leq \delta$. Moreover, if $g(z) \in [1, G]$ for all $z \in \mathcal{Z}$, then $D_{\mathsf{KL}}(\widehat{\pi} \,\|\, \pi) \leq 4\delta \log(4MG \log(4/\delta))$. The total number of sampling queries $y \sim \mu$ and function evaluations $g(\cdot)$ used by the algorithm is at most $8M \log(4\delta^{-1}) + 1$.*

**Proof.** See (Foster et al., 2025, Lemma E.1). $\qquad\square$

---

**Algorithm 6** `OutcomeLevelRS`: Outcome-Level Rejection Sampling

---

**Input:** base model $\pi_{\mathsf{ref}}$; tilt function $\tau$, prompt $x \in \mathcal{X}$, threshold $M > 0$, error $\varepsilon > 0$.
1: Sample $y_{1:H} \sim \texttt{RejectionSampling}(\tau(x,\cdot), \pi_{\mathsf{ref}}(\cdot \mid x), M, \varepsilon)$.        ▷ Algorithm 5
2: **return** $y_{1:H}$.

---

**Algorithm 7** `ActionLevelRS`: Action-Level Rejection Sampling

---

**Input:** base model $\pi_{\mathsf{ref}}$; appx. value function $\widehat{V}$, prompt $x \in \mathcal{X}$.
1: **Additional input for large-$|\mathcal{A}|$ regime:** threshold $M > 0$, error $\varepsilon > 0$.
2: **for** $1 \leq h \leq H$ **do**
    *Small-$|\mathcal{A}|$ regime:* explicitly compute $\mu_h \in \Delta(\mathcal{A})$ defined by

$$\mu_h(y_h) \propto \pi_{\mathsf{ref}}(y_h \mid x, y_{1:h-1}) \cdot \widehat{V}(x, y_{1:h}),$$

    and sample $y_h \sim \mu_h$.
    *Large-$|\mathcal{A}|$ regime:* sample                    ▷ Algorithm 5

$$y_h \sim \texttt{RejectionSampling}(\widehat{V}(x, (y_{1:h-1}, \cdot)), \pi_{\mathsf{ref}}(\cdot \mid x, y_{1:h-1}), M, \varepsilon/H).$$

3: **return** $y_{1:H}$.

---

## G    Omitted Results and Proofs from Section 2

In Appendix G.1, we give formal pseudocode and analyses for the baseline alignment algorithms `OutcomeLevelRS` (Proposition G.1) and `ActionLevelRS` (Propositions G.2 and G.3), which were informally introduced in Section 2. In Appendix G.2, we show that if the approximate value function satisfies Bellman consistency, then `ActionLevelRS` avoids error compounding (Proposition G.4). In Appendix G.3, we show that any algorithm that does not have access to a process verifier must incur a similar time complexity to `OutcomeLevelRS` (Proposition G.5).

### G.1    Pseudocode and Analyses for Baseline Sampling Algorithms

We formally describe and analyze the baseline sampling algorithms `OutcomeLevelRS` and `ActionLevelRS`, in the general test-time alignment setting where $\pi_{\mathsf{ref}} : \mathcal{X} \to \mathcal{Y}$ is the base model, $\tau : \mathcal{X} \times \mathcal{Y} \to \mathbb{R}_{\geq 0}$ is the tilt function, and given $x \in \mathcal{X}$ we would like to sample from the model $\pi^\star$ defined by $\pi^\star(y \mid x) \propto \pi_{\mathsf{ref}}(y \mid x)\tau(x,y)$. We also give specialized analyses for the KL-regularized setting where $\tau(x,y) = \exp(\beta^{-1} r^\star(x,y))$.

**Formal pseudocode for baseline algorithms.** See Algorithms 6 and 7 for pseudocode of the versions of `OutcomeLevelRS` and `ActionLevelRS` that we theoretically analyze below. We remark that the versions that we implement in experiments differ slightly—see Appendix E.1.

**Analyses of baseline algorithms.** Recall that we defined the *sequence-level coverage coefficient* and *action-level coverage coefficient* for a prompt $x$ as

$$\mathcal{C}_{\mathsf{seq}}(x) := \max_{y \in \mathcal{Y}} \frac{\pi^\star(y \mid x)}{\pi_{\mathsf{ref}}(y \mid x)}, \quad \text{and} \quad \mathcal{C}_{\mathsf{act}}(x) := \max_{h \in [H]} \max_{y_{1:h} \in \mathcal{A}} \frac{\pi^\star(y_h \mid x, y_{1:h-1})}{\pi_{\mathsf{ref}}(y_h \mid x, y_{1:h-1})}.$$

Proposition G.1 provides a guarantee for `OutcomeLevelRS`. Proposition G.2 provides a guarantee for `ActionLevelRS` in the large-$|\mathcal{A}|$ regime, and Proposition G.3 provides a guarantee for `ActionLevelRS` in the small-$|\mathcal{A}|$ regime.

**Proposition G.1** (`OutcomeLevelRS`)**.** *Given a prompt $x$ and parameter $\varepsilon > 0$, let $\widehat{\pi}(\cdot \mid x)$ be the output distribution of* `OutcomeLevelRS`$(\pi_{\mathsf{ref}}, \tau, x, 4\mathcal{C}_{\mathsf{seq}}(x), \varepsilon)$*. Then*

$$D_{\mathsf{TV}}(\widehat{\pi}(\cdot \mid x), \pi^\star(\cdot \mid x)) \leq \varepsilon. \tag{19}$$

*Moreover, in the special case where $\tau(x, y_{1:H}) = \exp(\beta^{-1} r^\star(x, y_{1:H}))$ for some $r^\star : \mathcal{X} \times \mathcal{Y} \to [0, R_{\mathsf{max}}]$ and $\beta > 0$, it holds that*

$$J_\beta(\pi^\star; x) - J_\beta(\widehat{\pi}; x) \lesssim (R_{\mathsf{max}} + \beta \log(16\mathcal{C}_{\mathsf{seq}}(x) \log(4/\varepsilon))) \cdot \varepsilon. \tag{20}$$

*The algorithm uses $\widetilde{O}\big(\mathcal{C}_{\mathsf{seq}}(x) \cdot \log(\varepsilon^{-1})\big)$ reward evaluations and generations $y \sim \pi_{\mathsf{ref}}(\cdot \mid x)$.*

To interpret the second guarantee, which bounds the KL-regularized regret of the law of `OutcomeLevelRS`, note that a natural parameter regime is $R_{\max} = 1$ (e.g. binary rewards) and $\beta \leq 1$. In this case, it holds that $J_\beta(\pi^\star; x) - J_\beta(\widehat{\pi}; x) \leq O(\varepsilon \log(\mathcal{C}_{\mathsf{seq}}(x)))$. Note that $\mathcal{C}_{\mathsf{seq}}(x)$ can in practice be exponentially large in $H$; however, since the time complexity of `OutcomeLevelRS` only depends logarithmically on $\varepsilon$, in this regime one can achieve $J_\beta(\pi^\star; x) - J_\beta(\widehat{\pi}; x) \leq \varepsilon$ while paying only a factor of $\log(H/\varepsilon)$ in the time complexity. In any case, the time complexity is dominated by $\mathcal{C}_{\mathsf{seq}}(x)$.

**Proof of Proposition G.1.** The first inequality Eq. (19) and the efficiency guarantee both follow immediately from Proposition F.2, using the choice of $M$ and $\delta$. The second inequality Eq. (20) follows from Proposition F.2 together with the choice of $M$ and the algebraic equality $J_\beta(\pi^\star; x) - J_\beta(\widehat{\pi}; x) = \beta D_{\mathsf{KL}}(\widehat{\pi} \parallel \pi^\star)$, which is shown in (Foster et al., 2025, Lemma F.4). $\qquad\square$

**Proposition G.2** (`ActionLevelRS`; large-$|\mathcal{A}|$ regime)**.** *Let $\widehat{V}$ be an approximate value function with*

$$\max\left\{ \frac{\widehat{V}(x, y_{1:h})}{V^\star_{\mathsf{tilt}}(x, y_{1:h})}, \frac{V^\star_{\mathsf{tilt}}(x, y_{1:h})}{\widehat{V}(x, y_{1:h})} \right\} \leq \kappa$$

*for all $h \in [H]$. The output distribution $\widehat{\pi}(\cdot \mid x)$ of* `ActionLevelRS`$(\pi_{\mathsf{ref}}, \widehat{V}, x, 4\mathcal{C}_{\mathsf{act}}(x)\kappa^2, \varepsilon)$ *in the large-$|\mathcal{A}|$ regime satisfies*

$$D_{\mathsf{TV}}(\widehat{\pi}(\cdot \mid x), \pi^\star(\cdot \mid x)) \leq \varepsilon + H\sqrt{2\log(\kappa)}. \tag{21}$$

*Moreover, in the special case where $\tau(x, y_{1:H}) = \exp(\beta^{-1} r^\star(x, y_{1:H}))$ for some $r^\star : \mathcal{X} \times \mathcal{Y} \to [0, R_{\max}]$ and $\beta > 0$, it holds that*

$$J_\beta(\pi^\star; x) - J_\beta(\widehat{\pi}; x) \lesssim (R_{\max} + \beta \log(16\mathcal{C}_{\mathsf{act}}(x) \log(4H/\varepsilon))) \cdot \varepsilon + 2H\beta \log(\kappa). \tag{22}$$

*In either case, the algorithm uses $\widetilde{O}(\mathcal{C}_{\mathsf{act}}(x)\kappa^2 \cdot H \log(H\varepsilon^{-1}))$ value function evaluations and generations $y_h \sim \pi_{\mathsf{ref}}(\cdot \mid x, y_{1:h-1})$.*

To interpret Eq. (22), we remark that if $\widehat{Q}(x, y_{1:h}) = \beta \log \widehat{V}(x, y_{1:h})$ satisfies $|\widehat{Q}(x, y_{1:h}) - Q^\star_\beta(x, y_{1:h})| \leq \varepsilon_Q$ for all $x, y_{1:h}$, then the condition of Proposition G.2 is satisfied with $\kappa := \exp(\varepsilon_Q/\beta)$, and if $R_{\max}, \beta, \mathcal{C}_{\mathsf{act}}(x) \leq O(1)$ then Eq. (22) reduces to

$$J_\beta(\pi^\star; x) - J_\beta(\widehat{\pi}; x) \lesssim \widetilde{O}(\varepsilon + \varepsilon_Q \cdot H).$$

While the term $\varepsilon$ can be efficiently reduced by increasing computation, the term $\varepsilon_Q \cdot H$ is inherent for this algorithm, as we will see later.

**Proof.** We have (by the fact that TV-distance is a metric and satisfies the data processing inequality) that

$$D_{\mathsf{TV}}(\widehat{\pi}(\cdot \mid x), \pi^\star(\cdot \mid x)) \leq \sum_{h=1}^{H} \mathbb{E}_{y_{1:h-1} \sim \widehat{\pi}(\cdot \mid x)} \left[ D_{\mathsf{TV}}(\widehat{\pi}_h(\cdot \mid x, y_{1:h-1}), \pi^\star_h(\cdot \mid x, y_{1:h-1})) \right] \tag{23}$$

where $\widehat{\pi}_h(\cdot \mid x, y_{1:h-1})$ is the marginal distribution on $y_h$ under $\widehat{\pi}(\cdot \mid x, y_{1:h-1})$, and similarly for $\pi^\star$. Fix any $h \in [H]$ and condition on $y_{1:h-1}$. We invoke Proposition F.2 with base measure $\mu(y_h) := \pi_{\mathsf{ref}}(y_h \mid x, y_{1:h-1})$ and tilt function $g(y_h) := \widehat{V}(x, y_{1:h})$. Define $\pi \in \Delta(\mathcal{A})$ by $\pi(y_h) \propto \mu(y_h)\widehat{V}(x, y_{1:h})$. It holds that

$$\begin{aligned}
\left\| \frac{\pi}{\mu} \right\|_\infty &= \max_{y_h \in \mathcal{A}} \frac{\widehat{V}(x, y_{1:h})}{\sum_{y'_h \in \mathcal{A}} \mu(y'_h)\widehat{V}(x, y_{1:h-1}, y'_h)} \\
&\leq \kappa^2 \cdot \max_{y_h \in \mathcal{A}} \frac{V^\star_{\mathsf{tilt}}(x, y_{1:h})}{\sum_{y'_h \in \mathcal{A}} \mu(y'_h)V^\star_{\mathsf{tilt}}(x, y_{1:h-1}, y'_h)} \\
&= \kappa^2 \cdot \max_{y_h \in \mathcal{A}} \frac{V^\star_{\mathsf{tilt}}(x, y_{1:h})}{\sum_{y'_h \in \mathcal{A}} \pi^\star(y'_h \mid x, y_{1:h-1})V^\star_{\mathsf{tilt}}(x, y_{1:h-1})} \\
&= \kappa^2 \cdot \max_{y_h \in \mathcal{A}} \frac{\pi^\star(y_h \mid x, y_{1:h-1})}{\pi_{\mathsf{ref}}(y_h \mid x, y_{1:h-1})} \\
&\leq \kappa^2 \cdot \mathcal{C}_{\mathsf{act}}(x)
\end{aligned}$$

where the inequality is by the assumption on $\widehat{V}$, the second and third equalities are by Lemma F.5, and the final inequality is by definition of $\mathcal{C}_{\mathsf{act}}(x)$. Thus, $M \geq 4 \left\| \frac{\pi}{\mu} \right\|_\infty$, so Proposition F.2 gives that $D_{\mathsf{TV}}(\widehat{\pi}_h(\cdot \mid x, y_{1:h-1}), \pi) \leq \delta = \varepsilon/H$. Moreover, we have

$$D_{\mathsf{TV}}(\pi, \pi_h^\star(\cdot \mid x, y_{1:h-1}))$$

$$\leq \sqrt{D_{\mathsf{KL}}(\pi \parallel \pi_h^\star(\cdot \mid x, y_{1:h-1}))}$$

$$\leq \sqrt{\max_{y_h \in \mathcal{A}} \log \frac{\pi(y_h)}{\pi^\star(y_h \mid x, y_{1:h-1})}}$$

$$= \sqrt{\max_{y_h \in \mathcal{A}} \log \frac{\pi_{\mathsf{ref}}(y_h \mid x, y_{1:h-1})\widehat{V}(x, y_{1:h})}{\pi_{\mathsf{ref}}(y_h \mid x, y_{1:h-1})V_{\mathsf{tilt}}^\star(x, y_{1:h})} \cdot \frac{\sum_{y_h' \in \mathcal{A}} \pi_{\mathsf{ref}}(y_h' \mid x, y_{1:h-1})V_{\mathsf{tilt}}^\star(x, y_{1:h-1}, y_h')}{\sum_{y_h' \in \mathcal{A}} \pi_{\mathsf{ref}}(y_h' \mid x, y_{1:h-1})\widehat{V}(x, y_{1:h-1}, y_h')}}$$

$$\leq \sqrt{2 \log \kappa}$$

by Pinsker's inequality and the assumption on $\widehat{V}$. It follows that

$$D_{\mathsf{TV}}(\widehat{\pi}_h(\cdot \mid x, y_{1:h-1}), \pi_h^\star(\cdot \mid x, y_{1:h-1})) \leq \frac{\varepsilon}{H} + \sqrt{2 \log \kappa}.$$

Substituting into Eq. (23) completes the proof of Eq. (21).

The proof of Eq. (22) proceeds via an analogous argument but starting with the chain rule for KL-divergence:

$$D_{\mathsf{KL}}(\widehat{\pi}(\cdot \mid x) \parallel \pi^\star(\cdot \mid x)) = \sum_{h=1}^H \mathop{\mathbb{E}}_{y_{1:h-1} \sim \widehat{\pi}(\cdot \mid x)} \left[ D_{\mathsf{KL}}(\widehat{\pi}_h(\cdot \mid x, y_{1:h-1}) \parallel \pi_h^\star(\cdot \mid x, y_{1:h-1})) \right]. \quad (24)$$

Fix $h \in [H]$ and condition on $y_{1:h-1}$. As before, the conditions of Proposition F.2 are satisfied, but in this special case, the proposition also gives a bound on the error induced by Algorithm 5 in terms of KL-divergence, instead of just TV-distance:

$$D_{\mathsf{KL}}(\widehat{\pi}_h(\cdot \mid x, y_{1:h-1}) \parallel \pi) \leq 4 \frac{\varepsilon}{H}(R_{\mathsf{max}}\beta^{-1} + \log(16\mathcal{C}_{\mathsf{act}}(x)\kappa^2 \log(4H/\varepsilon))).$$

As before, we have

$$\max_{y_h \in \mathcal{A}} \log \frac{\pi(y_h)}{\pi^\star(y_h \mid x, y_{1:h-1})} \leq 2 \log \kappa.$$

It follows that

$$D_{\mathsf{KL}}(\widehat{\pi}_h(\cdot \mid x, y_{1:h-1}) \parallel \pi_h^\star(\cdot \mid x, y_{1:h-1}))$$

$$= D_{\mathsf{KL}}(\widehat{\pi}_h(\cdot \mid x, y_{1:h-1}) \parallel \pi) + \mathop{\mathbb{E}}_{y_h \sim \widehat{\pi}_h(\cdot \mid x, y_{1:h-1})} \log \frac{\pi(y_h)}{\pi_h^\star(y_h \mid x, y_{1:h-1})}$$

$$\leq 4 \frac{\varepsilon}{H}(R_{\mathsf{max}}\beta^{-1} + \log(16\mathcal{C}_{\mathsf{act}}(x)\kappa^2 \log(4H/\varepsilon))) + 2 \log(\kappa).$$

Substituting into Eq. (24) and applying the fact that $J_\beta(\pi^\star; x) - J_\beta(\widehat{\pi}; x) = \beta D_{\mathsf{KL}}(\widehat{\pi}(\cdot \mid x) \parallel \pi^\star(\cdot \mid x))$ (see (Foster et al., 2025, Lemma F.4)) gives Eq. (22). □

In the small-$|\mathcal{A}|$ regime, we avoid the error incurred by approximate rejection sampling, but retain the main error term incurred by inaccuracy of the approximate value function (and incur time complexity scaling with $|\mathcal{A}|$). We omit the proof as it is a strict subset of the argument used to prove Proposition G.2.

**Proposition G.3** (ActionLevelRS; small-$|\mathcal{A}|$ regime). *Let $\widehat{V}$ be an approximate value function with*

$$\max \left\{ \frac{\widehat{V}(x, y_{1:h})}{V_{\mathsf{tilt}}^\star(x, y_{1:h})}, \frac{V_{\mathsf{tilt}}^\star(x, y_{1:h})}{\widehat{V}(x, y_{1:h})} \right\} \leq \kappa$$

*for all $h \in [H]$. The output distribution $\widehat{\pi}(\cdot \mid x)$ of* ActionLevelRS$(\pi_{\mathsf{ref}}, \widehat{V}, x)$ *in the small-$|\mathcal{A}|$ regime satisfies*

$$D_{\mathsf{TV}}(\widehat{\pi}(\cdot \mid x), \pi^{\star}(\cdot \mid x)) \le H\sqrt{2\log(\kappa)}. \tag{25}$$

*Moreover, in the special case where $\tau(x, y_{1:H}) = \exp(\beta^{-1} r^{\star}(x, y_{1:H}))$ for some $r^{\star} : \mathcal{X} \times \mathcal{Y} \to [0, R_{\mathsf{max}}]$ and $\beta > 0$, it holds that*

$$J_\beta(\pi^{\star}; x) - J_\beta(\widehat{\pi}; x) \le 2H\beta\log(\kappa). \tag{26}$$

*In either case, the algorithm uses $O(H|\mathcal{A}|)$ evaluations of $\widehat{V}$ and conditional density queries to $\pi_{\mathsf{ref}}$.*

### G.2 Action-Level Sampling for Consistent Value Functions

In order to better appreciate the advantage of VGB, it helps to consider the case when the approximate value function $\widehat{V}$ used by ActionLevelRS is in fact the (exact) value function for some policy $\widetilde{\pi}$. This condition is equivalent to a *Bellman consistency* condition, which is the formulation we use below, and it can arise if $\widehat{V}$ is an *implicit value function*: that is, if it is implicitly defined via

$$\widehat{V}(x, y_{1:h}) := \widehat{V}(x, y_{1:h-1}) \cdot \frac{\widetilde{\pi}(y_h \mid x, y_{1:h-1})}{\pi_{\mathsf{ref}}(y_h \mid x, y_{1:h-1})}$$

for an explicitly-learned policy $\widetilde{\pi}$ (this is essentially a rephrasing of the basic equality from Lemma F.5).

In this case, ActionLevelRS is simply sampling from the distribution $\widetilde{\pi}$ (up to the error due to rejection sampling), and the closeness of $\widetilde{\pi}$ to $\pi^{\star}$ depends only on the multiplicative ratio between $\widehat{V}$ and $V_{\mathsf{tilt}}^{\star}$, and not on $H$.

There is empirical evidence suggesting that implicit value functions may be more effective at fine-grained action-level guidance than explicit value functions (Liu et al., 2024c). The following result, contrasted with Proposition G.2 and the failures discussed in Section 3, provides a theoretical explanation for this observation. Moreover, it provides another perspective on the advantage of VGB: it can achieve comparable results for *explicit* value functions as ActionLevelRS achieves for implicit value functions.

**Proposition G.4** (ActionLevelRS: Improved guarantees for consistent value functions). *In the setting of Proposition G.2, suppose that the approximate value function $\widehat{V}$ additionally satisfies Bellman consistency: for all $x \in \mathcal{X}$, $h \in [H]$, and $y_{1:h-1} \in \mathcal{A}^{h-1}$,*

$$\widehat{V}(x, y_{1:h-1}) = \sum_{y_h \in \mathcal{A}} \pi_{\mathsf{ref}}(y_h \mid x, y_{1:h-1}) \widehat{V}(x, y_{1:h}).$$

*Then the output distribution $\widehat{\pi}(\cdot \mid x)$ of* ActionLevelRS *satisfies*

$$D_{\mathsf{TV}}(\widehat{\pi}(\cdot \mid x), \pi^{\star}(\cdot \mid x)) \le \varepsilon + \sqrt{2\log(\kappa)}.$$

**Proof of Proposition G.4.** From Bellman consistency, we have that

$$\widehat{V}(x, y_{1:h}) = \mathop{\mathbb{E}}_{y_{h+1:H} \sim \pi_{\mathsf{ref}}(\cdot \mid x, y_{1:h})}[\widehat{V}(x, y_{1:H})],$$

so $\widehat{V}$ is the value function for the tilted distribution $\widetilde{\pi} : \mathcal{X} \to \Delta(\mathcal{Y})$ defined by $\widetilde{\pi}(y_{1:H} \mid x) \propto \pi_{\mathsf{ref}}(y_{1:H} \mid x)\widehat{V}(x, y_{1:H})$. Then Lemma F.5 applied to $\widetilde{\pi}$ gives

$$\widetilde{\pi}(y_h \mid x, y_{1:h-1}) \propto \pi_{\mathsf{ref}}(y_h \mid x, y_{1:h-1})\widehat{V}(x, y_{1:h}).$$

We now analyze ActionLevelRS using this fact. Fix any $h \in [H]$ and condition on $y_{1:h-1}$. We invoke Proposition F.2 with base measure $\mu(y_h) := \pi_{\mathsf{ref}}(y_h \mid x, y_{1:h-1})$ and tilt function $g(y_h) := \widehat{V}(x, y_{1:h})$. We define $\pi \in \Delta(\mathcal{A})$ by $\pi(y_h) \propto \mu(y_h)\widehat{V}(x, y_{1:h})$. As in the proof of Proposition G.2, we get that $D_{\mathsf{TV}}(\widehat{\pi}_h(\cdot \mid x, y_{1:h-1}), \pi) \le \varepsilon/H$. But now we observe that $\pi$ coincides with $\widetilde{\pi}_h(\cdot \mid x, y_{1:h-1})$, i.e. the conditional distribution of $y_h$ under $\widetilde{\pi}(\cdot \mid x, y_{1:h-1})$. Therefore

$$D_{\mathsf{TV}}(\widehat{\pi}(\cdot \mid x), \widetilde{\pi}(\cdot \mid x)) \le \sum_{h=1}^{H} \mathop{\mathbb{E}}_{y_{1:h} \sim \widehat{\pi}(\cdot \mid x)}[D_{\mathsf{TV}}(\widehat{\pi}_h(\cdot \mid x, y_{1:h-1}), \widetilde{\pi}_h(\cdot \mid x, y_{1:h-1}))] \le \varepsilon.$$

We now observe that

$$D_{\mathsf{TV}}(\widetilde{\pi}(\cdot \mid x), \pi^\star(\cdot \mid x)) \le \sqrt{D_{\mathsf{KL}}(\widetilde{\pi}(\cdot \mid x) \,\|\, \pi^\star(\cdot \mid x))}$$

$$\le \sqrt{\max_{y_{1:H} \in \mathcal{Y}} \log \frac{\widetilde{\pi}(y_{1:H} \mid x)}{\pi^\star(y_{1:H} \mid x)}}$$

$$\le \sqrt{\max_{y_{1:H} \in \mathcal{Y}} \log \frac{\widehat{V}(x, y_{1:H})}{V^\star_{\mathtt{tilt}}(x, y_{1:H})} \cdot \frac{\sum_{y'_{1:H} \in \mathcal{Y}} V^\star_{\mathtt{tilt}}(x, y'_{1:H})}{\sum_{y'_{1:H} \in \mathcal{Y}} \widehat{V}(x, y'_{1:H})}}$$

$$\le \sqrt{2 \log \kappa}$$

by the assumption that the multiplicative ratio between $\widehat{V}$ and $V^\star_{\mathtt{tilt}}$ is bounded by $\kappa$ (see the statement of Proposition G.2). The triangle inequality for TV-distance completes the proof. □

### G.3 LOWER BOUND FOR OUTCOME-LEVEL ALGORITHMS

The following result shows that time complexity dependence on the sequence-level coverage coefficient $\mathcal{C}_{\mathsf{seq}}(x)$ is unavoidable when only outcome-level rewards are available; the proof is standard but included for completeness.

**Proposition G.5** (Lower bound for outcome-level algorithms). *Consider the KL-regularized optimization setting, where the algorithm only has access to $\pi_{\mathsf{ref}}$, $\beta$, and $r^\star$ (but no value function). Let $\mathcal{X} := \{\bot\}$ and $\mathcal{Y} := \{0,1\}^H$, and set $\pi_{\mathsf{ref}} := \mathsf{Unif}(\mathcal{Y})$. For any $C \in [1, 2^H]$, set $\beta := 1/\log(C)$. Any algorithm that satisfies $J_\beta(\pi^\star; x) - J_\beta(\widehat{\pi}; x) \le \frac{1}{8\log(C)}$ for all instances with $\mathcal{C}_{\mathsf{seq}}(x) \le C$ must use at least $N = \Omega(C)$ reward evaluations. Similarly, any algorithm that satisfies $D_{\mathsf{TV}}(\pi^\star, \widehat{\pi}) \le 1/4$ for all instances with $\mathcal{C}_{\mathsf{seq}}(x) \le C$ must use at least $N = \Omega(C)$ reward evaluations.*[22]

**Proof of Proposition G.5.** Without loss of generality, assume that $C$ is a power of 2. Fix $x \in \mathcal{X}$ and $\mathcal{Y} = \{0,1\}^H$; we ignore the dependence on $x$ henceforth. Let $\pi_{\mathsf{ref}} = \mathsf{Unif}(\mathcal{Y})$. We construct a *random* instance of the alignment problem as follows. Set $\beta := 1/\log(C)$ and let $S \subset \mathcal{Y}$ be a uniformly random set of size $10 \cdot 2^H / C$. Define $r^\star_S(y) := \mathbb{I}[y \in S]$, and let $\pi^\star_S$ be the desired distribution, i.e. $\pi^\star_S(y) \propto \pi_{\mathsf{ref}}(y) \exp(\beta^{-1} r^\star_S(y))$. Then

$$Z := \sum_{y \in \mathcal{Y}} \pi_{\mathsf{ref}}(y) \exp(\beta^{-1} r^\star_S(y)) = \frac{10}{C} \cdot C + \left(1 - \frac{10}{C}\right) \cdot 1 \in [10, 11].$$

Therefore for any $y \in \mathcal{Y}$,

$$\frac{\pi^\star_S(y)}{\pi_{\mathsf{ref}}(y)} := \frac{\exp(\beta^{-1} r^\star(y))}{Z} \le C,$$

and hence this instance satisfies the desired bound $\mathcal{C}_{\mathsf{seq}}(x) \le C$ on the sequence-level coverage coefficient. Let $\widehat{\pi}_S \in \Delta(\mathcal{Y})$ be the law of the algorithm for this instance.

Consider also the "null" instance in which $r^\star(y) := 0$ for all $y \in \mathcal{Y}$. Let $\widehat{\pi}_\circ \in \Delta(\mathcal{Y})$ be the law of the algorithm for this instance. The desired distribution for this instance is $\pi^\star_\circ := \pi_{\mathsf{ref}}$.

Suppose that the algorithm uses at most $N \le C/40$ reward evaluations. Then on average over the choice of $S$, the probability that the algorithm observes any reward-1 generation is at most $1/4$. Thus, there is some specific $S$ such that the probability that the algorithm observes a reward-1 generation is at most $1/4$. For this $S$, it holds that $D_{\mathsf{TV}}(\widehat{\pi}_S, \widehat{\pi}_\circ) \le 1/4$.

We now consider the two cases of the lemma's assumption: either the algorithm is accurate in TV-distance, or in KL-regularized regret. We show both are impossible under the preceding assumption that $N \le C/40$.

1. If $D_{\mathsf{TV}}(\widehat{\pi}_S, \pi^\star_S) \le 1/4$ and $D_{\mathsf{TV}}(\widehat{\pi}_\circ, \pi_{\mathsf{ref}}) \le 1/4$, then $D_{\mathsf{TV}}(\pi^\star_S, \pi_{\mathsf{ref}}) \le 3/4$. But $\pi_{\mathsf{ref}}(\mathcal{Y} \setminus S) = 1 - 10/C \ge 0.9$, whereas $\pi^\star_S(\mathcal{Y} \setminus S) \le 1/Z \le 0.1$. Thus $D_{\mathsf{TV}}(\pi^\star_S, \pi_{\mathsf{ref}}) \ge 0.8$, which is a contradiction.

---

[22]Formally, this result holds in the "sample-and-evaluate" model of computation (Huang et al., 2025a;b), where $\pi_{\mathsf{ref}}$ and $r^\star$ are initially unknown to the algorithm, and can only be accessed through black-box reward evaluation queries $r^\star(x, y)$ and generation queries $y \sim \pi_{\mathsf{ref}}(\cdot \mid x)$.

2. Similarly, suppose that $J_\beta(\pi_S^\star; S) - J_\beta(\widehat{\pi}_S; S) \leq 1/(8\log C)$ and $J_\beta(\pi_\circ^\star; \circ) - J_\beta(\widehat{\pi}_\circ; \circ) \leq 1/(8\log C)$, where $J_\beta(\cdot; S)$ is the KL-regularized regret for the instance parametrized by $S$, and $J_\beta(\cdot; \circ)$ is the KL-regularized regret for the null instance. It follows from (Foster et al., 2025, Lemma F.4) and choice of $\beta$ that $D_{\mathsf{KL}}(\widehat{\pi}_S \,\|\, \pi_S^\star) \leq 1/8$ and $D_{\mathsf{KL}}(\widehat{\pi}_\circ \,\|\, \pi_\circ^\star) \leq 1/8$. From Pinsker's inequality, we get $D_{\mathsf{TV}}(\widehat{\pi}_S, \pi_S^\star) \leq 1/4$ and $D_{\mathsf{TV}}(\widehat{\pi}_\circ, \pi_\circ^\star) \leq 1/4$. The proof concludes as in (1).

This completes the proof. $\qquad\square$

# H OMITTED RESULTS AND PROOFS FROM SECTION 3

In Appendix H.1 we provide computations omitted from Examples 3.1 and 3.2, formally proving that `ActionLevelRS` incurs $\Omega(1)$ TV-error in Example 3.1 and $\Omega(1)$ KL-regularized regret in Example 3.2. We also demonstrate that `OutcomeLevelRS` requires exponential time in these examples, and that `ActionLevelRS` achieves exponentially poor coverage over $\pi^\star$ (in contrast with the guarantee of Theorem 4.2 for VGB).

In Appendix H.2 we demonstrate that Example 3.1 can be adapted to the KL-regularized setting.

## H.1 DETAILS FOR EXAMPLES 3.1 AND 3.2

**Details for Example 3.1.** The calculation of $V_{\texttt{tilt}}^\star$ is straightforward from the definition

$$V_{\texttt{tilt}}^\star(y_{1:h}) = \mathbb{E}^{\pi_{\text{ref}}}[\tau(y_{1:H}) \mid y_{1:h}].$$

Moreover, by construction it is clear that $\widehat{V}(y_{1:h})/V_{\texttt{tilt}}^\star(y_{1:h}) \in [1/(1+\varepsilon_V), 1+\varepsilon_V]$ for all $y_{1:h}$. Note that $\pi^\star = \mathsf{Unif}(\{\mathfrak{a}, \mathfrak{b}\}^H)$ whereas the output distribution $\widehat{\pi}$ of `ActionLevelRS` is a product distribution that puts mass $(1+\varepsilon_V)/(2+\varepsilon_V)$ on $\mathfrak{a}$ for each coordinate.

First, we lower bound $D_{\mathsf{TV}}(\widehat{\pi}, \pi^\star)$. By the data processing inequality and Lemma F.1, we have

$$D_{\mathsf{TV}}(\widehat{\pi}, \pi^\star) \geq D_{\mathsf{TV}}(\mathrm{Bin}(H, 1/2), \mathrm{Bin}(H, (1+\varepsilon_V)/(2+\varepsilon_V))) \geq \Omega(\min(\varepsilon_V \sqrt{H}, 1))$$

as claimed.

Second, we observe that $\pi_{\text{ref}}(y_{1:H}) = 3^{-H}$ for all $y_{1:H} \in \mathcal{Y}$, whereas $\pi^\star(\mathfrak{a}, \ldots, \mathfrak{a}) = 2^{-H}$. Thus, $\mathcal{C}_{\texttt{seq}} \geq (3/2)^H$, so `OutcomeLevelRS` requires $\exp(\Omega(H))$ time in this example.

Third, in the regime $\varepsilon_V = \Theta(1)$, we show that $\widehat{\pi}$ does not approximately cover $\pi^\star$ (in the sense of Eq. (7) in Theorem 4.2). Suppose without loss of generality that $H$ is even, and let $\mathcal{E} \subset \mathcal{Y}$ be the set of $y_{1:H}$ that contain at most $H/2$ occurrences of $\mathfrak{a}$. Defining $\Delta := \varepsilon_V/(4+2\varepsilon_V)$, we have for any $y_{1:H} \in \mathcal{E}$ that

$$\frac{\widehat{\pi}(y_{1:H})}{\pi^\star(y_{1:H})} = (1+\Delta)^{\#\{h:y_h=\mathfrak{a}\}}(1-\Delta)^{H-\#\{h:y_h=\mathfrak{a}\}} \leq (1-\Delta^2)^{H/2} = \exp(-\Omega(H))$$

since $\Delta = \Omega(\varepsilon_V) = \Omega(1)$. It follows that for some constant $c = c(\varepsilon_V)$,

$$\Pr_{y_{1:H} \sim \pi^\star}\left[\frac{\pi^\star(y_{1:H})}{\widehat{\pi}(y_{1:H})} > \exp(cH)\right] \geq \pi^\star(\mathcal{E}) = \frac{1}{2},$$

i.e. $\widehat{\pi}$ has exponentially poor coverage of a constant fraction of the mass of $\pi^\star$.  $\square$

**Details for Example 3.2.** We calculate that the optimal regularized reward is $J_\beta(\pi^\star) = Q_\beta^\star(\varnothing) = \log((1+e)/2)$. For the distribution $\widehat{\pi}$ produced by `ActionLevelRS`, it is clear that $\widehat{\pi} = \pi_{\text{ref}} = \mathsf{Unif}(\{0,1\}^H)$, and hence

$$J_\beta(\pi^\star) - J_\beta(\widehat{\pi}) = J_\beta(\pi^\star) - \mathbb{E}_{\pi_{\text{ref}}}[r^\star(y_{1:H})] = \log((1+e)/2) - \frac{1}{2} \geq \frac{1}{10} \qquad (27)$$

as claimed. Next, we observe that $\pi^\star = \mathrm{Ber}(1/2 + \Delta)^{\otimes H}$ for some $\Delta = \Omega(1)$, whereas $\widehat{\pi} = \pi_{\text{ref}} = \mathrm{Ber}(1/2)^{\otimes H}$. Thus, the fact that `OutcomeLevelRS` requires $\exp(\Omega(H))$ time follows by an analogous argument as in Example 3.1, as does the fact that `ActionLevelRS` fails to approximately cover $\pi^\star$.  $\square$

## H.2 KL-REGULARIZED ABC EXAMPLE

**Example H.1** (Failure of `ActionLevelRS` with approximate $Q_\beta^\star$). *Consider the following construction for $\varepsilon_Q \in [0,1]$ and $H \in \mathbb{N}$. We have a single prompt $\mathcal{X} = \{\bot\}$ (henceforth we will omit the prompt from the notation) and action space $\mathcal{A} = \{\mathfrak{a}, \mathfrak{b}, \mathfrak{c}\}$. We define $\pi_{\text{ref}} = \mathsf{Unif}(\{\mathfrak{a}, \mathfrak{b}, \mathfrak{c}\}^H)$ and $r^\star(y_{1:H}) = \mathbb{I}\{\mathfrak{c} \notin y_{1:H}\}$. Then it can be checked that*

$$Q_\beta^\star(y_{1:h}) = \beta \log\left(1 + (2/3)^{H-h}(e^{\beta^{-1}} - 1)\mathbb{I}[\mathfrak{c} \notin y_{1:h}]\right).$$

*Let us define*

$$\widehat{Q}(y_{1:h}) := \begin{cases} Q_\beta^\star(y_{1:h}) + \varepsilon_Q & \text{if } y_h = \mathfrak{a} \\ Q_\beta^\star(y_{1:h}) & \text{otherwise} \end{cases}.$$

*Then* `ActionLevelRS` *with approximate value function* $\widehat{V}(y_{1:h}) := \exp(\beta^{-1}\widehat{Q}(y_{1:h}))$ *satisfies* $J_\beta(\pi^\star) - J_\beta(\widehat{\pi}) \geq \Omega(\min(\varepsilon_Q, \beta) \cdot H).$[23] $\triangleleft$

**Details for Example H.1.** We first derive the expression for $Q_\beta^\star$. For any $y_{1:h} \in \mathcal{A}^h$, if $\mathfrak{c} \in y_{1:h}$, then

$$Q_\beta^\star(y_{1:h}) = \beta \log \mathbb{E}^{\pi_{\text{ref}}}[\exp(\beta^{-1} r^\star(y_{1:H})) \mid y_{1:h}] = \beta \log 1$$

since $r^\star(y_{1:H})$ will be almost surely 0. On the other hand, if $\mathfrak{c} \notin y_{1:h}$, then

$$\begin{aligned} Q_\beta^\star(y_{1:h}) &= \beta \log \mathbb{E}^{\pi_{\text{ref}}}[\exp(\beta^{-1} r^\star(y_{1:H})) \mid y_{1:h}] \\ &= \beta \log \left( (2/3)^{H-h} \cdot \exp(\beta^{-1}) + (1 - (2/3)^{H-h}) \cdot 1 \right) \\ &= \beta \log \left( 1 + (2/3)^{H-h}(\exp(\beta^{-1}) - 1) \right). \end{aligned}$$

Thus, it holds for all $y_{1:h} \in \mathcal{A}^h$ that

$$Q_\beta^\star(y_{1:h}) = \beta \log \left( 1 + (2/3)^{H-h}(\exp(\beta^{-1}) - 1)\mathbb{I}[\mathfrak{c} \notin y_{1:h}] \right)$$

as claimed. Next, we observe that by construction, $|\widehat{Q}(y_{1:h}) - Q^\star(y_{1:h})| \leq \varepsilon_Q$ for all $h \in [H]$ and $y_{1:h} \in \mathcal{A}^h$. It remains to verify the lower bound on KL-regularized regret. By (Foster et al., 2025, Lemma F.4) and the chain rule for KL-divergence, we have

$$J_\beta(\pi^\star) - J_\beta(\widehat{\pi}) = \beta \sum_{h=1}^H \mathbb{E}^{\pi_{\text{ref}}} \left[ D_{\mathsf{KL}}(\widehat{\pi}_h(\cdot \mid y_{1:h-1}) \,\|\, \pi_h^\star(\cdot \mid y_{1:h-1})) \right]. \tag{28}$$

Fix $h \in [H]$ and condition on $y_{1:h-1}$. We consider two cases.

1. Suppose that $\mathfrak{c} \in y_{1:h-1}$. Then $\exp(\beta^{-1}Q_\beta^\star(y_{1:h})) = 1$ for each $y_h \in \mathcal{A}$, so $\pi_h^\star(\cdot \mid y_{1:h-1})$ is uniform over $\mathcal{A}$. However, $\exp(\beta^{-1}\widehat{Q}(y_{1:h})) = \exp(\varepsilon_Q\beta^{-1}\mathbb{I}[y_h = \mathfrak{a}])$, so $\widehat{\pi}_h(\mathfrak{a} \mid x, y_{1:h-1}) = e^{\varepsilon_Q/\beta}/(2 + e^{\varepsilon_Q/\beta}) \geq 1/3$. It follows that

$$D_{\mathsf{KL}}(\widehat{\pi}_h(\cdot \mid y_{1:h-1}) \,\|\, \pi_h^\star(\cdot \mid y_{1:h-1})) \geq \frac{1}{3} \cdot \log \frac{e^{\varepsilon_Q/\beta}/(2 + e^{\varepsilon_Q/\beta})}{1/3} \geq \Omega(\min(\varepsilon_Q/\beta, 1)). \tag{29}$$

2. Suppose that $\mathfrak{c} \notin y_{1:h-1}$. Then $\exp(\beta^{-1}Q_\beta^\star(y_{1:h})) = 1 + (2/3)^{H-h}(e^{\beta^{-1}} - 1)\mathbb{I}[y_h \neq \mathfrak{c}]$. On the other hand, $\exp(\beta^{-1}\widehat{Q}(y_{1:h})) = \exp(\beta^{-1}Q_\beta^\star(y_{1:h})) \cdot \exp(\varepsilon_Q\beta^{-1}\mathbb{I}[y_h = \mathfrak{a}])$. Therefore

$$\begin{aligned} &D_{\mathsf{KL}}(\widehat{\pi}_h(\cdot \mid y_{1:h-1}) \,\|\, \pi_h^\star(\cdot \mid y_{1:h-1})) \\ &\geq \frac{1}{3} \log \frac{e^{\varepsilon_Q/\beta}/(e^{\beta^{-1}Q_\beta^\star(y_{1:h-1},\mathfrak{a})+\varepsilon/\beta} + e^{\beta^{-1}Q_\beta^\star(y_{1:h-1},\mathfrak{b})} + e^{\beta^{-1}Q_\beta^\star(y_{1:h-1},\mathfrak{c})})}{1/(e^{\beta^{-1}Q_\beta^\star(y_{1:h-1},\mathfrak{a})} + e^{\beta^{-1}Q_\beta^\star(y_{1:h-1},\mathfrak{b})} + e^{\beta^{-1}Q_\beta^\star(y_{1:h-1},\mathfrak{c})})} \\ &\geq \frac{1}{3} \log \left( 1 + \frac{e^{\varepsilon_Q/\beta} - 1}{e^{\varepsilon_Q/\beta} + 2} \right) \\ &\geq \Omega(\min(\varepsilon_Q/\beta, 1)) \end{aligned}$$

where the penultimate inequality uses the fact that $Q_\beta^\star(y_{1:h-1}, \mathfrak{a}) = Q_\beta^\star(y_{1:h-1}, \mathfrak{b}) \geq Q_\beta^\star(y_{1:h-1}, \mathfrak{c})$.

Substituting into Eq. (28) gives that $J_\beta(\pi^\star) - J_\beta(\widehat{\pi}) \geq \Omega(\min(\varepsilon_Q, \beta)H)$ as claimed. $\square$

---

[23]The sequence-level coverage coefficient is also exponential in $H$, $\mathcal{C}_{\text{seq}} \geq 2^{\Omega(H)}$, and hence `OutcomeLevelRS` requires an exponential computational budget.

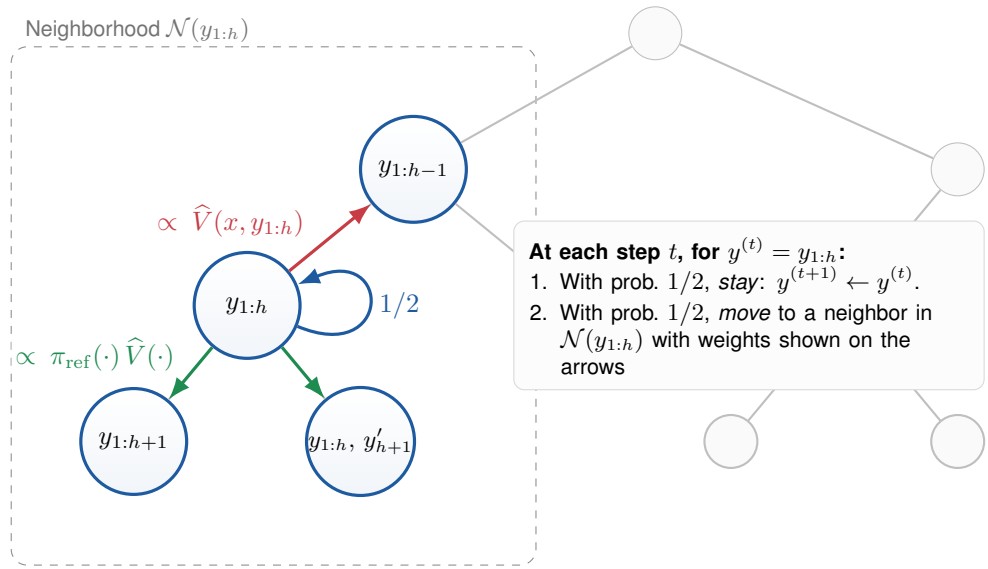

Figure 21: Illustration of execution of VGB at each step $t$.

## I  OMITTED RESULTS AND PROOFS FROM SECTION 4

In this section we formally prove Theorems 4.1 and 4.2. Specifically, this section is organized as follows. In Appendix I.1 we formally discuss how VGB is implemented in the large-$|\mathcal{A}|$ regime (Algorithm 8). In Appendix I.2 we show that Theorem 4.1 follows from a more general guarantee for the small-$|\mathcal{A}|$ regime, Theorem I.1. We also give a regret guarantee for VGB in the KL-regularized setting (Corollary I.1). In Appendix I.3 we prove Theorem I.1. In Appendix I.4 we prove Theorem 4.2.

We also include a high-level illustration of VGB (Figure 21).

**Notation.**  Recall that we let $\mathcal{T}$ denote the autoregressive tree of generations, which has node set $\bigcup_{h=0}^{H} \mathcal{A}^h$. We also let $\mathcal{N}(y_{1:h})$ denote the neighborhood of node $y_{1:h}$ in $\mathcal{T}$. In the proofs in this section, we will typically omit dependence on the prompt $x$, therefore writing e.g. $\pi_{\mathsf{ref}}(y_{1:H})$ instead of $\pi_{\mathsf{ref}}(y_{1:H} \mid x)$. Moreover, we will consider both distributions over the set $\mathcal{A}^H$ (leaves of the autoregressive tree) as well as distributions over the entire tree $\mathcal{T}$. To avoid confusion, we will notate the former as $\pi_{\mathsf{ref}}, \pi^\star, \widetilde{\pi}$, etc. whereas we will use other characters for the latter. Thus, $\pi_{\mathsf{ref}}(y_{1:h})$ refers to the marginal density of $\pi_{\mathsf{ref}} \in \Delta(\mathcal{A}^H)$ on $y_{1:h}$, whereas $\nu(y_{1:h})$ refers to the single-point density of $\nu \in \Delta(\mathcal{T})$ at $y_{1:h}$.

### I.1  DETAILED IMPLEMENTATION OF VGB FOR UNIFORM-ERROR RESULTS

In this section we present Algorithm 8, which gives more detailed pseudocode for VGB (compared to Algorithm 1). In particular, Algorithm 8 shows how to efficiently implement the transitions of the random walk in VGB via rejection sampling in the large-$|\mathcal{A}|$ case. For any node $y_{1:h}$ in the autoregressive tree $\mathcal{T}$, it is convenient to define the following distribution $q_{\mathsf{ref}}(\cdot \mid y_{1:h})$ over the neighborhood $\mathcal{N}(y_{1:h})$.

**Definition I.1** (Unweighted neighborhood distribution). *Let $y_{1:h} \in \mathcal{T}$ (so that $0 \le h \le H$). We define $q_{\mathsf{ref}}(\cdot \mid y_{1:h}) \in \Delta(\mathcal{N}(y_{1:h}))$ as follows:*

- *If $h = 0$, then $q_{\mathsf{ref}}(y_{1:h+1} \mid y_{1:h}) := \pi_{\mathsf{ref}}(y_{h+1} \mid y_{1:h})$ for all $y_{h+1} \in \mathcal{A}$.*

- *If $h = H$, then $q_{\mathsf{ref}}(y_{1:h-1} \mid y_{1:h}) := 1$.*

- *If $0 < h < H$, then $q_{\mathsf{ref}}(y_{1:h-1} \mid y_{1:h}) := \frac{1}{2}$ and $q_{\mathsf{ref}}(y_{1:h+1} \mid y_{1:h}) := \frac{1}{2}\pi_{\mathsf{ref}}(y_{h+1} \mid y_{1:h})$ for all $y_{h+1} \in \mathcal{A}$.*

In other words, $q_{\mathsf{ref}}$ represents a transition kernel on $\mathcal{T}$ that transitions to the parent node with probability $1/2$ (if possible), and transitions to a random child node with probaiblity $1/2$ (if possible,

---

**Algorithm 8** Detailed Implementation of VGB for Uniform-Error Results

---

1: **Input:** Reference model $\pi_{\mathrm{ref}}$, estimated value function $\widehat{V}$, prompt $x \in \mathcal{X}$, horizon $H \in \mathbb{N}$, step count $T \in \mathbb{N}$, sampling regime ("large-$|\mathcal{A}|$" or "small-$|\mathcal{A}|$").
2: **Additional input for large-$|\mathcal{A}|$ regime:** rejection threshold $M > 0$, rejection error $\delta_{\mathrm{rej}} \in (0, 1)$.
3: Initialize $y^{(0)} := \varnothing$.
4: **for** $0 \leq t < T$ **do**
5:     Set $h := |y^{(t)}|$.
6:     Set $q_{\mathrm{ref}}^{(t)} := q_{\mathrm{ref}}(\cdot \mid y^{(t)})$.                                             ▷ Definition I.1
7:     Let $g^{(t)} : \mathcal{N}(y_{1:h}^{(t)}) \to \mathbb{R}_{\geq 0}$ be defined by

$$g^{(t)}(y_{1:h-1}^{(t)}) := \widehat{V}(x, y_{1:h}^{(t)})$$

    and, for all $y_{h+1}' \in \mathcal{A}$,

$$g^{(t)}(y_{1:h}^{(t)}, y_{h+1}') := \widehat{V}(x, y_{1:h}^{(t)}, y_{h+1}')$$

8:     With probability $1/2$, set $y^{(t+1)} := y^{(t)}$, else:
  *Small-$|\mathcal{A}|$ regime:* explicitly compute $p^{(t)} \in \Delta(\mathcal{N}(y_{1:h}^{(t)}))$ defined by

$$p^{(t)}(z) \propto q_{\mathrm{ref}}^{(t)}(z) \cdot g^{(t)}(z),$$

  and sample $y^{(t+1)} \sim p^{(t)}$.
  *Large-$|\mathcal{A}|$ regime:* sample                                       ▷ Algorithm 5

$$y^{(t+1)} \leftarrow \mathtt{RejectionSampling}(g^{(t)}, q_{\mathrm{ref}}^{(t)}, M, \delta_{\mathrm{rej}}).$$

9: **return** $y^{(T)}$.

---

and under the distribution induced by $\pi_{\mathrm{ref}}$). Notably, sampling from $q_{\mathrm{ref}}(\cdot \mid y_{1:h})$ is tractable with a single conditional sampling query to $\pi_{\mathrm{ref}}$. To implement the transitions of VGB at step $t$ in the large-$|\mathcal{A}|$ case, we use $\mathtt{RejectionSampling}$ (Algorithm 5) with base distribution $q_{\mathrm{ref}}(\cdot \mid y_{1:h}^{(t)})$ and tilt function defined using $\widehat{V}$. See Algorithm 8 for details.

## I.2 PROOF OF THEOREM 4.1: UNIFORM ERROR BOUNDS

In this section, we state Theorem I.1—our general guarantee under uniform error bounds—and use it to prove Theorem 4.1. Notably, Theorem I.1 does not require exact access to the outcome-level reward $\tau$, and also implies a regret bound in the KL-regularized setting (Corollary I.1). It requires the following assumption on the multiplicative error of $\widehat{V}$, which reduces to Assumption 4.1 in the special case that $\kappa_{\mathrm{leaf}} = 1$. We assume that $\kappa_{\mathrm{leaf}} \leq \kappa$ for convenience.

**Assumption I.1** (Uniform bound on value errors; generalization of Assumption 4.1). *Let $\kappa \geq \kappa_{\mathrm{leaf}} \geq 1$. For all $x \in \mathcal{X}$, $h \in [H]$, and $y_{1:h} \in \mathcal{A}^h$, it holds that*

$$\max\left\{ \frac{\widehat{V}(x, y_{1:h})}{V_{\mathrm{tilt}}^\star(x, y_{1:h})}, \frac{V_{\mathrm{tilt}}^\star(x, y_{1:h})}{\widehat{V}(x, y_{1:h})} \right\} \leq \kappa. \tag{30}$$

*Moreover, for all $x \in \mathcal{X}$ and $y_{1:H} \in \mathcal{A}^H$, it holds that*

$$\max\left\{ \frac{\widehat{V}(x, y_{1:H})}{V_{\mathrm{tilt}}^\star(x, y_{1:H})}, \frac{V_{\mathrm{tilt}}^\star(x, y_{1:H})}{\widehat{V}(x, y_{1:H})} \right\} \leq \kappa_{\mathrm{leaf}}. \tag{31}$$

We remark that in general, $\kappa_{\mathrm{leaf}}$ can be interpreted as capturing the quality of the estimated reward model, while $\kappa$ captures the quality of the estimated value function. If the estimated reward model is perfect, we have that $\kappa_{\mathrm{leaf}} = 1$.

To state the formal guarantee of VGB under Assumption I.1, we must introduce the following model $\widetilde{\pi}$, which is defined by tilting $\pi_{\mathrm{ref}}$ by the estimated values at the leaves.

**Definition I.2** (Ideal information-theoretic sampling distribution). *Let* $\widetilde{\pi} : \mathcal{X} \to \Delta(\mathcal{Y})$ *be defined by*

$$\widetilde{\pi}(y_{1:H} \mid x) := \frac{\pi_{\mathsf{ref}}(y_{1:H} \mid x) \, \widehat{V}(x, y_{1:H})}{\sum_{y'_{1:H} \in \mathcal{A}^H} \pi_{\mathsf{ref}}(y'_{1:H} \mid x) \, \widehat{V}(x, y'_{1:H})}. \tag{32}$$

While it is not generically possible to sample from a distribution arbitrarily close to $\pi^\star$ when $\kappa_{\mathsf{leaf}} > 1$ (even information-theoretically), we will show that it *is* possible to sample from a distribution arbitrarily close to $\widetilde{\pi}$ (with time complexity of the sampling algorithm scaling logarithmically in the sampling error). Moreover, it is straightforward to see that $\widetilde{\pi}$ is close to $\pi^\star$ in the sense of coverage:

**Fact I.1.** *Under Assumption I.1, the density ratio between $\pi^\star$ and $\widetilde{\pi}$ is bounded as*

$$\max\left\{ \frac{\widetilde{\pi}(y_{1:H} \mid x)}{\pi^\star(y_{1:H} \mid x)}, \frac{\pi^\star(y_{1:H} \mid x)}{\widetilde{\pi}(y_{1:H} \mid x)} \right\} \leq \kappa_{\mathsf{leaf}}^2 \tag{33}$$

*for all $x \in \mathcal{X}$ and $y_{1:H} \in \mathcal{A}^H$.*

We will show that VGB can be used to sample from a distribution that is close to $\widetilde{\pi}$ in $\chi^2$-divergence (Definition F.1). Note that the output of VGB is a node of $\mathcal{T}$, which may or may not be a leaf node, whereas $\widetilde{\pi}$ (and $\pi^\star$) are distributions over leaf nodes. However, the following result demonstrates that VGB produces a leaf node with reasonable probability, and that *conditioned* on this event, the output distribution is close to $\widetilde{\pi}$.

**Theorem I.1** (Main accuracy guarantee under uniform errors). *There is an absolute constant $C_{4.1} > 0$ with the following property. In the general test-time alignment setting (Section 2), suppose that Assumption I.1 holds with parameters $\kappa \geq \kappa_{\mathsf{leaf}} \geq 1$. Fix $x \in \mathcal{X}$, $T \in \mathbb{N}$, and $\varepsilon \in (0, 1)$, and let $\nu \in \Delta(\mathcal{T})$ be the distribution of the output $y^{(T)}$ of Algorithm 8 with inputs $\pi_{\mathsf{ref}}$, $\widehat{V}$, $x$, $H$, and $T$, in the small-$|\mathcal{A}|$ regime.*

*Let $\mathcal{E}$ be the event that $|y^{(T)}| = H$. If $T \geq C_{4.1}\kappa^4 H^2 \log(\kappa H / \varepsilon)$, it holds that*

$$\nu(\mathcal{E}) \geq \frac{1}{8\kappa_{\mathsf{leaf}}\kappa H} \tag{34}$$

*and*

$$D_{\chi^2}(\nu|_{\mathcal{E}} \parallel \widetilde{\pi}) \leq \varepsilon. \tag{35}$$

The proof of Theorem 4.1 from Theorem I.1 is largely straightforward. The main technical detail is to extend the accuracy guarantees to the large-$|\mathcal{A}|$ regime, which we do by approximately coupling the execution of VGB in the large-$|\mathcal{A}|$ regime to its execution in the small-$|\mathcal{A}|$ regime.

**Remark I.1** (Hyperparameters for rejection sampling in large-$|\mathcal{A}|$ regime). *We remark that in the large-$|\mathcal{A}|$ regime, as shown in Algorithm 8, VGB formally requires two additional hyperparameters (which were omitted from the statement of Theorem 4.1), in order to use* RejectionSampling *(Algorithm 5) to implement transitions in the random walk: (1) a rejection threshold $M$ and (2) an error tolerance $\delta_{\mathsf{rej}}$. In the proof of Theorem 4.1 below, we assume that $\delta$, $\mathcal{C}_{\mathsf{act}}(x)$, and $\varepsilon_V$ are known, and moreover $M := 4\mathcal{C}_{\mathsf{act}}(x)(1 + \varepsilon_V)^2$ and $\delta_{\mathsf{rej}} := \delta / (16(1 + \varepsilon_V)HT)$. In the practical implementation of VGB, we use a heuristic approximation of* RejectionSampling *that only requires a single hyperparameter—see Appendix E.1.*

**Proof of Theorem 4.1.** We observe that under Assumption 4.1, Assumption I.1 is satisfied with $\kappa := 1 + \varepsilon_V$ and $\kappa_{\mathsf{leaf}} := 1$. Also, by Fact I.1, we have $\widetilde{\pi} = \pi^\star$.

First, we consider the execution of VGB in the small-$|\mathcal{A}|$ regime (as formally described in Algorithm 8). Invoking Theorem I.1 with $\varepsilon = \delta^2/4$, we get $\Pr[\mathcal{E}_{\mathsf{leaf}}] \geq 1/(8(1 + \varepsilon_V)H)$ and

$$D_{\mathsf{TV}}(\widehat{\pi}|_{\mathcal{E}_{\mathsf{leaf}}}, \pi^\star) \leq \sqrt{D_{\chi^2}(\widehat{\pi}|_{\mathcal{E}_{\mathsf{leaf}}} \parallel \pi^\star)} \leq \delta/2 \leq \delta,$$

where the first inequality uses Proposition F.1. The time complexity bound of $O(T \cdot |\mathcal{A}|)$ is evident from the algorithm description.

We next extend the analysis to the large-$|\mathcal{A}|$ regime where each transition is implemented using RejectionSampling (as described in Algorithm 8). Fix some step $t \in [T]$. To invoke the guarantee

for `RejectionSampling` (Proposition F.2), we must bound $\left\| p^{(t)}/q_{\mathsf{ref}}^{(t)} \right\|_\infty$. Suppose that $h := |y^{(t)}|$ satisfies $0 < h < H$ (the edge cases follow by analogous and simpler arguments). We have

$$\frac{p^{(t)}(y_{1:h-1}^{(t)})}{q_{\mathsf{ref}}^{(t)}(y_{1:h-1}^{(t)})} = 2p^{(t)}(y_{1:h-1}^{(t)}) \leq 2,$$

and, for any $y_h \in \mathcal{A}$,

$$
\begin{aligned}
\frac{p^{(t)}(y_{1:h}^{(t)}, y_{h+1})}{q_{\mathsf{ref}}^{(t)}(y_{1:h}^{(t)}, y_{h+1})} &= \frac{\widehat{V}(x, y_{1:h}^{(t)}, y_{h+1})}{\frac{1}{2}\widehat{V}(x, y_{1:h}^{(t)}) + \frac{1}{2}\sum_{y'_{h+1} \in \mathcal{A}} \pi_{\mathsf{ref}}(y'_{h+1} \mid x, y_{1:h}^{(t)})\widehat{V}(x, y_{1:h}^{(t)}, y'_{h+1})} \\
&\leq \frac{\kappa^2 V_{\mathsf{tilt}}^\star(x, y_{1:h}^{(t)}, y_{h+1})}{\frac{1}{2}V_{\mathsf{tilt}}^\star(x, y_{1:h}^{(t)}) + \frac{1}{2}\sum_{y'_{h+1} \in \mathcal{A}} \pi_{\mathsf{ref}}(y'_{h+1} \mid x, y_{1:h}^{(t)})V_{\mathsf{tilt}}^\star(x, y_{1:h}^{(t)}, y'_{h+1})} \\
&= \frac{\kappa^2 V_{\mathsf{tilt}}^\star(x, y_{1:h}^{(t)}, y_{h+1})}{\frac{1}{2}V_{\mathsf{tilt}}^\star(x, y_{1:h}^{(t)}) + \frac{1}{2}V_{\mathsf{tilt}}^\star(x, y_{1:h}^{(t)})} \\
&= \kappa^2 \frac{\pi^\star(y_{h+1} \mid x, y_{1:h}^{(t)})}{\pi_{\mathsf{ref}}(y_{h+1} \mid x, y_{1:h}^{(t)})} \\
&\leq \kappa^2 \mathcal{C}_{\mathsf{act}}(x)
\end{aligned}
$$

where the first inequality is by Assumption I.1, the second equality is by definition of $V_{\mathsf{tilt}}^\star$ (Eq. (3)), the third equality is by Lemma F.5, and the final inequality is by definition of $\mathcal{C}_{\mathsf{act}}(x)$. It follows from Proposition F.2 (so long as the hyperparameters in Algorithm 8 are set appropriately—see Remark I.1) that in the event that `RejectionSampling` is invoked at step $t$, the distribution $\widehat{p}^{(t)}$ of its output satisfies $D_{\mathsf{TV}}(\widehat{p}^{(t)}, p^{(t)}) \leq \delta/(16(1 + \varepsilon_V)HT)$. Let $\widehat{\pi}$ denote the distribution of the output $y^{(T)}$, and let $\widehat{\pi}_{\mathsf{small}}$ denote the distribution of the output in the small-$|\mathcal{A}|$ regime; then a standard coupling argument gives $D_{\mathsf{TV}}(\widehat{\pi}, \widehat{\pi}_{\mathsf{small}}) \leq \delta/(16(1 + \varepsilon_V)H)$, and therefore $D_{\mathsf{TV}}(\widehat{\pi}|_\mathcal{E}, \widehat{\pi}_{\mathsf{small}}|_\mathcal{E}) \leq \delta/2$ by Lemma F.2. By the above analysis, we have $D_{\mathsf{TV}}(\widehat{\pi}_{\mathsf{small}}|_\mathcal{E}, \pi^\star) \leq \delta/2$, so we get $D_{\mathsf{TV}}(\widehat{\pi}|_\mathcal{E}, \pi^\star) \leq \delta$ as needed. The time complexity bound claimed in the theorem statement is evident from the choice of $M$ (Remark I.1) and the pseudocode for `RejectionSampling` (Algorithm 5). □

**Remark I.2** (Large-$|\mathcal{A}|$ regime with average-case error bound). *A key step in the analysis of Theorem 4.1 for the large-$|\mathcal{A}|$ regime was to show that every transition can be efficiently implemented with rejection sampling. This required showing that at every step $t$, the desired sampling distribution $p^{(t)}$ has bounded density ratio with respect to a tractable proposal distribution (to invoke Proposition F.2). Since $p^{(t)}$ is defined in terms of $\widehat{V}$, this in turn requires some* uniform *assumption on $\widehat{V}$. Above, we showed that Assumption 4.1 suffices to control the density ratio in terms of the action-level coverage coefficient (and the assumption parameter $\kappa = 1 + \varepsilon_V$).*

*However, with only the average-case bound Assumption 4.2, it is unclear how to make a similar argument work. For this reason, we only state Theorem 4.2 in the small-$|\mathcal{A}|$ regime. One could easily extend it to the large-$|\mathcal{A}|$ regime by introducing an additional assumption that explicitly controls the density bounds in the above analysis. It may also be possible to avoid extra assumptions via a more delicate average-case argument that allows the rejection sampling procedure to occasionally fail. We leave this question for future work.*

In the KL-regularized setting, we get the following additional guarantee on the KL-regularized regret (Section 2). We only provide this guarantee in the small-$|\mathcal{A}|$ regime, but we believe it may be possible to extend to the large-$|\mathcal{A}|$ regime using a similar argument as in the analysis of `ActionLevelRS` (Proposition G.2).

**Corollary I.1** (Regret guarantee for VGB in KL-regularized setting). *In the KL-regularized alignment setting (Section 2) with temperature parameter $\beta > 0$ and reward function $r^\star : \mathcal{X} \times \mathcal{Y} \to [0, R_{\max}]$, suppose that Assumption 4.1 holds with parameter $\varepsilon_V > 0$. Fix prompt $x \in \mathcal{X}$ and $T \in \mathbb{N}$, and let $\nu \in \Delta(\mathcal{T})$ be the distribution of the output $y^{(T)}$ of Algorithm 8 with inputs $\pi_{\mathsf{ref}}$, $\widehat{V}$, $x$, $H$, and $T$, in the small-$|\mathcal{A}|$ regime. Let $\mathcal{E}$ be the event that $|y^{(T)}| = H$. Then, for any $\delta \in (0, 1)$, if $T \geq C_{4.1}\kappa^4 H^2 \log(\kappa H/\delta)$, it holds that*

$$J_\beta(\pi^\star; x) - J_\beta(\nu|_\mathcal{E}; x) \leq \beta\delta. \tag{36}$$

**Proof.** As above, Assumption I.1 is satisfied with $\kappa = 1 + \varepsilon_V$ and $\kappa = 1$, and $\widetilde{\pi} = \pi^\star$. Invoking Theorem I.1 with $\varepsilon_V := \delta$, we have

$$
\begin{aligned}
J_\beta(\pi^\star; x) - J_\beta(\nu|_\varepsilon; x) &= \beta \cdot D_{\mathsf{KL}}(\nu|_\varepsilon \parallel \pi^\star) \\
&\leq \beta \cdot D_{\chi^2}(\nu|_\varepsilon \parallel \pi^\star) \\
&\leq \beta\delta
\end{aligned}
$$

where the equality is by Lemma F.4 and the first inequality is by Proposition F.1. $\qquad\square$

### I.3 PROOF OF THEOREM I.1: UNIFORM ERROR BOUNDS

This proof largely follows the template originated by Sinclair & Jerrum (1989), generalized appropriately to the test-time alignment setting: we show that VGB implements a reversible Markov chain, compute its stationary distribution, and use a conductance argument to bound its mixing time. See also Bakshi et al. (2024) for a related but incomparable generalization (using a slightly different random walk).

Fix a prompt $x \in \mathcal{X}$. Since we will be working with the same prompt throughout the proof, we will drop it from the notation in the remainder of the section, and define quantities that implicitly depend on $x$. In particular, we will write $V^\star_{\texttt{tilt}}(y_{1:h}) := V^\star_{\texttt{tilt}}(x, y_{1:h})$, $\widehat{V}(y_{1:h}) := \widehat{V}(x, y_{1:h})$, and $\pi_{\mathsf{ref}}(y_{1:h}) := \pi_{\mathsf{ref}}(y_{1:h} \mid x)$ as shorthand.

Recall that $\mathcal{T}$ is the tree with node set $\mathcal{A}^0 \cup \cdots \cup \mathcal{A}^H$ and with edge set defined by setting the parent of $y_{1:h}$ to be $y_{1:h-1}$. For $w, v \in \mathcal{T}$, we will write $w \sim v$ if $w$ and $v$ are adjacent in the tree, i.e. if one is the parent of the other as defined above.

In order to prove Theorem I.1, we will use standard notions from the theory of Markov chains, such as reversibility, stationary distributions, and conductance. For a detailed introduction to Markov chains, we refer the reader to Levin et al. (2009).

**Definition I.3.** *For any $y_{1:h} \in \mathcal{T}$ with $h > 0$, define*

$$
f(y_{1:h-1}, y_{1:h}) := f(y_{1:h}, y_{1:h-1}) := \widehat{V}(y_{1:h})\pi_{\mathsf{ref}}(y_{1:h}).
$$

*Define a Markov chain $\mathbb{P}$ on $\mathcal{T}$ where, for any $v, w \in \mathcal{T}$, the probability of transitioning from $v$ to $w$ is*

$$
\mathbb{P}(w \mid v) := \begin{cases} \frac{f(v,w)}{2\sum_{w' \sim v} f(v,w')} & \text{if } w \sim v \\ 1/2 & \text{if } w = v \\ 0 & \text{otherwise} \end{cases}.
$$

Note that this Markov chain exactly corresponds to a step in Algorithm 8 (in the small-$|\mathcal{A}|$ regime).

**Definition I.4.** *Define $\mu \in \Delta(\mathcal{T})$ by*

$$
\mu(v) := \frac{1}{Z_f} \sum_{w \in \mathcal{T}: w \sim v} f(v, w)
$$

*where $Z_f := \sum_{v \in \mathcal{T}} \sum_{w \in \mathcal{T}: w \sim v} f(v, w)$ is the normalizing constant.*

The next lemma shows that the Markov chain $\mathbb{P}$ is reversible with respect to $\mu$ (Definition F.2).

**Lemma I.1.** *The Markov chain $\mathbb{P}$ is reversible with stationary distribution $\mu$.*

**Proof.** Recall that for $w, v \in \mathcal{T}$ with $w \sim v$, the transition probability is

$$
\mathbb{P}(w \mid v) = \frac{f(v, w)}{2\sum_{w' \sim v} f(v, w')}.
$$

Hence

$$
\mu(v)\mathbb{P}(w \mid v) = \left(\frac{1}{Z_f} \sum_{w' \sim v} f(v, w')\right)\left(\frac{f(v, w)}{2\sum_{w' \sim v} f(v, w')}\right) = \frac{f(v, w)}{2Z_f}.
$$

By the same calculation with $v$ and $w$ swapped,

$$
\mu(w)\mathbb{P}(v \mid w) = \frac{f(w, v)}{2Z_f} = \frac{f(v, w)}{2Z_f},
$$

using $f(v, w) = f(w, v)$. Therefore $\mu(v)\mathbb{P}(w \mid v) = \mu(w)\mathbb{P}(v \mid w)$ for all adjacent $v, w$, verifying detailed balance and hence reversibility with stationary distribution $\mu$. $\qquad\square$

We begin by showing that the stationary distribution $\mu$ has sufficient mass on the leaves of the tree and the root of the tree. The first part is a specific claim of Theorem I.1; the second part is important in bounding the mixing time of the random walk in Algorithm 8, since it starts at the root node of the tree.

**Lemma I.2.** *Under Assumption I.1, it holds that for*

$$\sum_{y_{1:H}\in\mathcal{A}^H} \mu(y_{1:H}) \geq \frac{1}{2\kappa\kappa_{\mathsf{leaf}}H} \tag{37}$$

*and*

$$\mu(\varnothing) \geq \frac{1}{2\kappa^2 H}. \tag{38}$$

**Proof.** We observe that

$$\sum_{v=y_{1:H}\in\mathcal{A}^H} \sum_{w\in\mathcal{T}:w\sim v} f(v, w) = \sum_{y_{1:H}\in\mathcal{A}^H} \pi_{\mathsf{ref}}(y_{1:H})\widehat{V}(y_{1:H})$$

$$\geq \frac{1}{\kappa_{\mathsf{leaf}}} \sum_{y_{1:H}\in\mathcal{A}^H} \pi_{\mathsf{ref}}(y_{1:H})V_{\mathsf{tilt}}^{\star}(y_{1:H})$$

$$= \frac{1}{\kappa_{\mathsf{leaf}}}V_{\mathsf{tilt}}^{\star}(\varnothing),$$

where the final equality uses Eq. (3), and

$$Z_f = \sum_{v\in\mathcal{T}} \sum_{w\in\mathcal{T}:w\sim v} f(v, w)$$

$$\leq 2\sum_{h=1}^{H}\sum_{y_{1:h}} \pi_{\mathsf{ref}}(y_{1:h})\widehat{V}(y_{1:h})$$

$$\leq 2\kappa\sum_{h=1}^{H}\sum_{y_{1:h}} \pi_{\mathsf{ref}}(y_{1:h})V_{\mathsf{tilt}}^{\star}(y_{1:h})$$

$$= 2\kappa H V_{\mathsf{tilt}}^{\star}(\varnothing)$$

again using Eq. (3). It follows from Definition I.4 that

$$\sum_{y_{1:H}\in\mathcal{A}^H} \mu(y_{1:H}) = \frac{1}{Z_f}\sum_{v=y_{1:H}\in\mathcal{A}^H}\sum_{w\in\mathcal{T}:w\sim v} f(v, w) \geq \frac{1}{2\kappa\kappa_{\mathsf{leaf}}H}$$

which proves Eq. (37). Similarly, Eq. (38) follows from the fact that

$$\sum_{a\in\mathcal{A}} f(\varnothing, a) = \sum_{a\in\mathcal{A}} \pi_{\mathsf{ref}}(a)\widehat{V}(a) \geq \frac{1}{\kappa}\sum_{a\in\mathcal{A}} \pi_{\mathsf{ref}}(a)V_{\mathsf{tilt}}^{\star}(a) = \frac{1}{\kappa}V_{\mathsf{tilt}}^{\star}(\varnothing)$$

together with the previously-derived upper bound on $Z_f$. $\qquad\square$

In order to bound the mixing time of the Markov chain $\mathbb{P}$, we will bound its conductance (Definition F.4).

**Lemma I.3.** *The conductance $\Phi$ of the Markov chain $\mathbb{P}$ satisfies*

$$\Phi \geq \frac{1}{4\kappa^2 H}. \tag{39}$$

**Proof.** Let $S \subseteq \mathcal{T}$ be any subgraph with $0 < \mu(S) \leq 1/2$. Suppose that $\varnothing \in S$. We know that $\mu(S) \leq \mu(\mathcal{T} \setminus S)$, and thus $\Phi_S \geq \Phi_{\mathcal{T} \setminus S}$. We have $\varnothing \notin \mathcal{T} \setminus S$. Hence, it suffices to prove that $\Phi_S \geq 1/(8\kappa^2 H)$ for all $S \subseteq \mathcal{T}$ with $0 < \mu(S)$ and $\varnothing \notin S$.

Fix any $S \subseteq \mathcal{T}$ with $0 < \mu(S)$ and $\varnothing \notin S$. Let $S_1, \ldots, S_k$ be the (maximal) connected components of $S$ in $\mathcal{T}$. Then

$$\Phi_S = \frac{\sum_{i=1}^{k} \sum_{u \in S_i, v \notin S} \mu(u)\mathbb{P}(v \mid u)}{\sum_{i=1}^{k} \sum_{u \in S_i} \mu(u)} = \frac{\sum_{i=1}^{k} \sum_{u \in S_i, v \notin S_i} \mu(u)\mathbb{P}(v \mid u)}{\sum_{i=1}^{k} \sum_{u \in S_i} \mu(u)} \geq \min_{i \in [k]} \Phi_{S_i}$$

where the second equality uses the fact that if $u \in S_i$ and $v \in S_j$ for $i \neq j$ then $u \not\sim v$ and so $\mathbb{P}(v \mid u) = 0$. The above inequality implies that it suffices to prove $\Phi_S \geq 1/(8\kappa^2 H)$ for all connected subgraphs $S \subseteq \mathcal{T}$ with $0 < \mu(S)$ and $\varnothing \notin S$.

Fix any connected subgraph $S \subseteq \mathcal{T}$ with $0 < \mu(S)$ and $\varnothing \notin S$. Let $v := y_{1:h}$ be the shallowest node in $S$. Then $v \neq \varnothing$, so $v$ has a parent $u := y_{1:h-1}$. Moreover, since $S$ is connected, every $v' \in S$ is a descendant of $v$. Hence,

$$\Phi_S \geq \frac{\mu(v)\mathbb{P}(u \mid v)}{\sum_{v' \in \mathcal{T}: v' \preceq v} \mu(v')}.$$

Here, we use the notation $v' \preceq v$ to denote that $v$ is an ancestor of $v'$. Using the definition $\mu(w) := \frac{1}{Z_f} \sum_{w' \in \mathcal{T}: w' \sim w} f(w, w')$ (Definition I.4) and the definition of $\mathbb{P}$ (Definition I.3), we get

$$\frac{\mu(v)\mathbb{P}(u \mid v)}{\sum_{v' \in \mathcal{T}: v' \preceq v} \mu(v')} = \frac{\dfrac{f(v, u)}{2Z_f}}{\dfrac{1}{Z_f} \sum_{v' \in \mathcal{T}: v' \preceq v} \sum_{w \in \mathcal{T}: w \sim v'} f(v', w)}$$

$$= \frac{f(v, u)}{2 \sum_{v' \in \mathcal{T}: v' \preceq v} \sum_{w \in \mathcal{T}: w \sim v'} f(v', w)}.$$

By Assumption I.1, we have

$$f(v, u) = \pi_{\mathsf{ref}}(v)\widehat{V}(v) \geq \frac{1}{\kappa} \pi_{\mathsf{ref}}(v)V_{\mathtt{tilt}}^{\star}(v).$$

To bound the denominator, note that $f(v', w) = \pi_{\mathsf{ref}}(v' \vee w)\widehat{V}(v' \vee w)$ where $v' \vee w \in \mathcal{T}$ is the deeper node among $\{v', w\}$. Moreover, for every node $u$ in the subtree rooted at $v$, there are only two ordered pairs $(v', w) \in \mathcal{T}^2$ with $v' \sim w$ and $u = v' \vee w$: namely, $(u, u')$ and $(u', u)$ where $u'$ is the parent of $u$. It follows that

$$\sum_{v' \in \mathcal{T}: v' \preceq v} \sum_{w \in \mathcal{T}: w \sim v'} f(v', w) \leq 2 \sum_{v' \in \mathcal{T}: v' \preceq v} \pi_{\mathsf{ref}}(v')\widehat{V}(v') \leq 2\kappa \sum_{v' \in \mathcal{T}: v' \preceq v} \pi_{\mathsf{ref}}(v')V_{\mathtt{tilt}}^{\star}(v').$$

Combining the preceding displays yields

$$\Phi_S \geq \frac{\pi_{\mathsf{ref}}(v)V_{\mathtt{tilt}}^{\star}(v)}{4\kappa^2 \sum_{v' \in \mathcal{T}: v' \preceq v} \pi_{\mathsf{ref}}(v')V_{\mathtt{tilt}}^{\star}(v')}.$$

Expanding the denominator using Eq. (3), and noting that the depth of the subtree rooted at $v$ is at most $H$, we get

$$\Phi_S \geq \frac{1}{4\kappa^2 H}.$$

It follows that $\Phi$ satisfies the same bound. $\qquad\square$

Let $\mathbb{P}^t(\cdot \mid z)$ denote the distribution of the Markov chain $\mathbb{P}$ after $t$ steps starting from state $z$. We recall the standard result that a lower bound on conductance implies that the $\chi^2$ divergence $D_{\chi^2}(\mathbb{P}^t(\cdot \mid z) \parallel \mu)$ decays exponentially in $t$ (Theorem F.1). This lets us conclude the proof of Theorem I.1.

**Proof of Theorem I.1.** Let $\nu$ denote the distribution of the output of Algorithm 8. Then $\nu = \mathbb{P}^T(\cdot \mid \varnothing)$. Note that the Markov chain $\mathbb{P}$ is aperiodic since it has self-loops. Since $\widehat{V}(y_{1:h}) > 0$ if and only

if $V_{\text{tilt}}^\star(y_{1:h}) > 0$, the chain has positive transition probability between any two nodes $\{y_{1:h-1}, y_{1:h}\}$ with $\pi^\star(y_{1:h}) > 0$. Moreover, $\mu(y_{1:h}) > 0$ implies that $\pi^\star(y_{1:h}) > 0$. Thus, $\mathbb{P}$ is irreducible on the support of $\mu$, and hence ergodic on the support of $\mu$ (Definition F.3). We can now invoke Theorem F.1 to get

$$D_{\chi^2}(\nu \parallel \mu) \lesssim e^{-\Phi^2 T/2} \frac{1}{\mu(\varnothing)}. \tag{40}$$

Since $1/\mu(\varnothing) \le 4\kappa^2 H$ (Lemma I.2) and $\Phi \ge 1/(4\kappa^2 H)$ (Lemma I.3), we have for any $\varepsilon' > 0$, if $T \ge 32\kappa^4 H^2 \log(\kappa^2 H/\varepsilon')$, then

$$D_{\chi^2}(\nu \parallel \mu) \lesssim \varepsilon'. \tag{41}$$

Set $\varepsilon' := \varepsilon/(8\kappa\kappa_{\text{leaf}}H)^2$; the condition on $T$ is satisfied so long as $C_{I.1}$ is a sufficiently large constant. Let $L$ be the set of leaves of $\mathcal{T}$. Since $\mu(L) \ge 1/(2\kappa\kappa_{\text{leaf}}H)$ (Lemma I.2), we have that

$$\nu(L) \ge \mu(L) - D_{\text{TV}}(\nu, \mu) \ge \mu(L) - \sqrt{\varepsilon'} \ge \frac{1}{4\kappa\kappa_{\text{leaf}}H}$$

using Proposition F.1, which proves the first claim of the theorem.

Next, by Lemma F.2, Eq. (41), and Lemma I.2, we have

$$D_{\chi^2}(\nu|_{\mathcal{E}} \parallel \mu|_{\mathcal{E}}) \le \frac{D_{\chi^2}(\nu \parallel \mu)}{\mu(\mathcal{E}) - 2\sqrt{\mu(\mathcal{E})D_{\chi^2}(\nu \parallel \mu)}} \le \frac{\varepsilon'}{\frac{1}{2\kappa\kappa_{\text{leaf}}H} - 2\sqrt{\varepsilon'}} \le \varepsilon. \tag{42}$$

Now note that for any leaf $v = y_{1:H} \in \mathcal{T}$,

$$\mu(v) \propto f(y_{1:H-1}, y_{1:H}) = \widehat{V}(y_{1:H})\pi_{\text{ref}}(y_{1:H}) \propto \widetilde{\pi}(y_{1:H})$$

by Definitions I.2 and I.4. Therefore $\mu|_{\mathcal{E}} = \widetilde{\pi}$, and so

$$D_{\chi^2}(\nu|_{\mathcal{E}} \parallel \widetilde{\pi}) \le \varepsilon \tag{43}$$

which proves the second claim of the theorem. $\qquad\square$

### I.4 Proof of Theorem 4.2: Average Error Bounds

The proof of Theorem 4.2 analyzes the same Markov chain $\mathbb{P}$ as the proof of Theorem 4.1 (modulo returning $y^{(t)}$ for a random $t \in [T]$ rather than returning $y^{(T)}$), but the analysis requires substantially different techniques. In particular, under Assumption 4.2, it is no longer true that the Markov chain mixes rapidly to the "globally" stationary distribution, since there may be branches of the tree where $\widehat{V}$ is extremely inaccurate. We instead proceed using the machinery of *local stationarity / meta-stability* (Balasubramanian et al., 2022; Liu et al., 2024b), which is a weakening of fast mixing, analogous to how, in *optimization theory*, local critical points are a weakened solution concept compared to global optima.

For a broad class of Markov chains, the sampling analogue of an approximate zero-gradient condition is a low *Dirichlet form* condition, which implies that the sampling law "locally" resembles the stationary distribution (Liu et al., 2024b). To prove Theorem 4.2, we apply this idea to $\mathbb{P}$ and use local stationarity to show that the marginal distribution of the Markov chain at a random timestep approximately covers the true stationary distribution (which, in turn, approximately covers $\pi^\star$).

As before, we will fix the prompt $x \in \mathcal{X}$ and drop it from the notation. Throughout, we will assume that Assumption 4.2 holds with parameter $\varepsilon_V$, and for notational convenience we will write $\kappa := 1 + \varepsilon_V$. Thus, for each $h \in [H]$, we have the following bounds on $\widehat{V}$ with respect to $V_{\text{tilt}}^\star$:

$$\max\left(\mathbb{E}^{\pi^\star}\left[\frac{V_{\text{tilt}}^\star(y_{1:h})}{\widehat{V}(y_{1:h})}\right], \mathbb{E}^{\pi^\star}\left[\frac{\widehat{V}(y_{1:h})}{V_{\text{tilt}}^\star(y_{1:h})}\right]\right) \le \kappa. \tag{44}$$

In this setting, it is no longer true that every cut has large conductance (with respect to the Markov chain $\mathbb{P}$). We define a set of "bad" nodes at each layer of the autoregressive tree $\mathcal{T}$, where "bad" roughly means that the subtree rooted at that node has small conductance, and formally is defined as follows:

**Definition I.5.** *Let $\eta > 0$ and $h \in [H]$. Define*

$$\mathcal{B}_{h,\eta} := \left\{ y_{1:h} \in \mathcal{A}^h : \pi_{\mathsf{ref}}(y_{1:h})\widehat{V}(y_{1:h}) < \frac{\eta^2}{4\kappa^2} \cdot \sum_{y_{h+1:H} \in \mathcal{A}^{H-h}} \pi_{\mathsf{ref}}(y_{1:H})\widehat{V}(y_{1:H}) \right\}.$$

**Lemma I.4.** *For any $\eta > 0$ and $h \in [H]$ it holds that*

$$\pi^\star(\mathcal{B}_{h,\eta}) \leq \eta.$$

**Proof.** We know that $\pi^\star(y_{1:H}) = \frac{1}{V_{\mathsf{tilt}}^\star(\varnothing)}\pi_{\mathsf{ref}}(y_{1:H})V_{\mathsf{tilt}}^\star(y_{1:H})$, and so

$$\pi^\star(y_{1:h}) = \frac{1}{V_{\mathsf{tilt}}^\star(\varnothing)} \sum_{y_{h+1:H} \in \mathcal{A}^{H-h}} \pi_{\mathsf{ref}}(y_{1:H})V_{\mathsf{tilt}}^\star(y_{1:H}) = \frac{1}{V_{\mathsf{tilt}}^\star(\varnothing)}\pi_{\mathsf{ref}}(y_{1:h})V_{\mathsf{tilt}}^\star(y_{1:h})$$

by definition of $V_{\mathsf{tilt}}^\star$ (Eq. (3)). Thus, for any fixed $y_{1:h} \in \mathcal{A}^h$,

$$\frac{\sum_{y_{h+1:H} \in \mathcal{A}^{H-h}} \pi_{\mathsf{ref}}(y_{1:H})\widehat{V}(y_{1:H})}{\pi_{\mathsf{ref}}(y_{1:h})\widehat{V}(y_{1:h})} = \frac{\sum_{y_{h+1:H} \in \mathcal{A}^{H-h}} \pi^\star(y_{1:H})\frac{\widehat{V}(y_{1:H})}{V_{\mathsf{tilt}}^\star(y_{1:H})}}{\pi^\star(y_{1:h})\frac{\widehat{V}(y_{1:h})}{V_{\mathsf{tilt}}^\star(y_{1:h})}}$$

$$= \frac{V_{\mathsf{tilt}}^\star(y_{1:h})}{\widehat{V}(y_{1:h})}\mathbb{E}^{\pi^\star}\left[\frac{\widehat{V}(y_{1:H})}{V_{\mathsf{tilt}}^\star(y_{1:H})} \mid y_{1:h}\right].$$

We conclude that

$$\Pr_{y_{1:h} \sim \pi^\star}[y_{1:h} \in \mathcal{B}_{h,\eta}] = \Pr_{y_{1:h} \sim \pi^\star}\left[\frac{V_{\mathsf{tilt}}^\star(y_{1:h})}{\widehat{V}(y_{1:h})}\mathbb{E}^{\pi^\star}\left[\frac{\widehat{V}(y_{1:H})}{V_{\mathsf{tilt}}^\star(y_{1:H})} \mid y_{1:h}\right] > \frac{4\kappa^2}{\eta^2}\right]$$

$$\leq \Pr_{y_{1:h} \sim \pi^\star}\left[\frac{V_{\mathsf{tilt}}^\star(y_{1:h})}{\widehat{V}(y_{1:h})} > \frac{2\kappa}{\eta}\right] + \Pr_{y_{1:h} \sim \pi^\star}\left[\mathbb{E}^{\pi^\star}\left[\frac{\widehat{V}(y_{1:H})}{V_{\mathsf{tilt}}^\star(y_{1:H})} \mid y_{1:h}\right] > \frac{2\kappa}{\eta}\right]$$

$$\leq \frac{\eta}{2\kappa}\mathbb{E}^{\pi^\star}\left[\frac{V_{\mathsf{tilt}}^\star(y_{1:h})}{\widehat{V}(y_{1:h})}\right] + \frac{\eta}{2\kappa}\mathbb{E}^{\pi^\star}\left[\frac{\widehat{V}(y_{1:H})}{V_{\mathsf{tilt}}^\star(y_{1:H})}\right]$$

$$\leq \eta$$

where the final two inequalities are by Markov's inequality and Eq. (44) respectively. $\square$

Recall the definition of Markov chain $\mathbb{P}$ from Definition I.3. It will often be convenient to interpret $\mathbb{P}$ as a $\mathcal{T} \times \mathcal{T}$ matrix, i.e. with $\mathbb{P}_{xy} := \mathbb{P}(y \mid x)$. We will denote by $\nu_0$ the initial distribution which puts all the mass on the root node $\varnothing$, i.e. $\nu_0(\varnothing) = 1$. Denote by $\nu_t := \nu_0\mathbb{P}^t$ the distribution of the Markov chain after $t$ steps starting from $\nu_0$. Recall, from Lemma I.1, that the Markov chain $\mathbb{P}$ is reversible with stationary distribution $\mu$ defined by

$$\mu(v) := \frac{1}{Z_f}\sum_{w \in \mathcal{T}:w \sim v} f(v, w) \tag{45}$$

where $Z_f := \sum_{v,w \in \mathcal{T}:w \sim v} f(v, w)$. Furthermore, we have the following lower bounds on (1) the mass that $\mu$ places on the root node $\varnothing$, and (2) the mass that $\mu$ places on the set of leaves $\mathcal{A}^H \subseteq \mathcal{T}$. Lemma I.5 is the analogue of Lemma I.2 for the average-case setting.

**Lemma I.5.** *The stationary distribution $\mu$ satisfies $\mu(\varnothing) \geq 1/(2\kappa^2 H)$ and $\mu(\mathcal{A}^H) \geq 1/(2\kappa^2 H)$. Furthermore, it holds that $Z_f \leq 2V_{\mathsf{tilt}}^\star(\varnothing)H\kappa$.*

**Proof.** First, we observe that

$$Z_f = \sum_{v \in \mathcal{T}}\sum_{w \in \mathcal{T}:w \sim v} f(v, w) \leq 2\sum_{h=1}^H \sum_{y_{1:h} \in \mathcal{A}^h} \pi_{\mathsf{ref}}(y_{1:h})\widehat{V}(y_{1:h})$$

$$= 2V_{\mathsf{tilt}}^\star(\varnothing)\sum_{h=1}^H \mathbb{E}^{\pi^\star}\left[\frac{\widehat{V}(y_{1:h})}{V_{\mathsf{tilt}}^\star(y_{1:h})}\right]$$

$$\leq 2V_{\mathsf{tilt}}^\star(\varnothing)H\kappa$$

by Eq. (44). Next,

$$\sum_{y_{1:H} \in \mathcal{A}^H} f(y_{1:H}, y_{1:H-1}) = \sum_{y_{1:H} \in \mathcal{A}^H} \pi_{\text{ref}}(y_{1:H}) \widehat{V}(y_{1:H})$$

$$= V_{\text{tilt}}^\star(\varnothing) \, \mathbb{E}^{\pi^\star} \left[ \frac{\widehat{V}(y_{1:H})}{V_{\text{tilt}}^\star(y_{1:H})} \right]$$

$$\geq \frac{V_{\text{tilt}}^\star(\varnothing)}{\mathbb{E}^{\pi^\star} \left[ \frac{V_{\text{tilt}}^\star(y_{1:H})}{\widehat{V}(y_{1:H})} \right]}$$

$$\geq \frac{V_{\text{tilt}}^\star(\varnothing)}{\kappa}$$

by Jensen's inequality ($r \mapsto 1/r$ is convex on $(0, \infty)$) and Eq. (44). It follows from this bound and the preceding bound on $Z_f$ that

$$\mu(\mathcal{A}^H) = \sum_{y_{1:H} \in \mathcal{A}^H} \frac{f(y_{1:H}, y_{1:H-1})}{Z_f} \geq \frac{1}{2\kappa^2 H}.$$

Similarly,

$$\sum_{y_1 \in \mathcal{A}} f(\varnothing, y_1) = \sum_{y_1 \in \mathcal{A}} \pi_{\text{ref}}(y_1) \widehat{V}(y_1) = V_{\text{tilt}}^\star(\varnothing) \, \mathbb{E}^{\pi^\star} \left[ \frac{\widehat{V}(y_1)}{V_{\text{tilt}}^\star(y_1)} \right] \geq \frac{V_{\text{tilt}}^\star(\varnothing)}{\kappa},$$

which, together with the bound on $Z_f$, implies that $\mu(\varnothing) \geq 1/(2\kappa^2 H)$. $\qquad \square$

For the analysis, we will look at the action of the Markov chain on the space of functions $g : \mathcal{T} \to \mathbb{R}$. Specifically, we will work with the space $L^2(\mu)$, which has inner product defined as follows.

**Definition I.6** (Inner product with respect to the stationarity distribution). *For any functions $g, g' : \mathcal{T} \to \mathbb{R}$, we define the $\mu$-inner product $\langle g, g' \rangle_\mu$ by $\langle g, g' \rangle_\mu := \mathbb{E}_\mu[gg'] = g^\top D g'$ (where $D = \text{diag}(\mu)$).*

Next, we recall the notion of the Dirichlet form corresponding to a Markov chain, which can be seen as a generalization of the Laplacian. This is a measure of how a function varies locally with respect to the transitions of the Markov chain.

**Definition I.7** (Dirichlet form). *For any functions $g, g' : \mathcal{T} \to \mathbb{R}$, we define the* Dirichlet form *of $g, g'$ (with respect to kernel $\mathbb{P}$ and stationary distribution $\mu$) to be*

$$\mathcal{E}(g, g') := \sum_{u, v \in \mathcal{T}} \mu(u) \mathbb{P}(v \mid u)(g(u) - g(v)) \cdot (g'(u) - g'(v)).$$

The following lemma gives a standard expression for the Dirichlet form. The proof can be found in (Montenegro & Tetali, 2005, Section 1.1).

**Lemma I.6.** *For any function $g : \mathcal{T} \to \mathbb{R}$ it holds that*

$$\mathcal{E}(g, g) = 2 \langle g, (I - \mathbb{P}) g \rangle_\mu$$

*where $I$ is the identity operator.*

Rather than directly working with $\mathbb{P}$ as a matrix, it is more convenient to work with the following similar matrix—which is symmetric and PSD due to the laziness of the random walk, as shown in Lemma I.7.

**Definition I.8.** *Define $Q \in \mathbb{R}^{\mathcal{T} \times \mathcal{T}}$ by $Q_{xy} := \mathbb{P}(y \mid x) \sqrt{\mu(x)/\mu(y)}$, so that $Q = D^{1/2} \mathbb{P} D^{-1/2}$ where $D = \text{diag}(\mu)$.*

**Lemma I.7.** *The matrix $Q$ is symmetric and positive semi-definite.*

**Proof.** The first claim follows from reversibility of $\mathbb{P}$ (Lemma I.1):

$$Q_{xy} = \mathbb{P}(y \mid x)\mu(x)\sqrt{\frac{1}{\mu(x)\mu(y)}} = \mathbb{P}(x \mid y)\mu(y)\sqrt{\frac{1}{\mu(x)\mu(y)}} = Q_{yx}.$$

Since $Q$ is similar to $\mathbb{P}$, it has the same eigenvalues as $\mathbb{P}$ (which are real-valued since $Q$ is symmetric). By construction, we can write $\mathbb{P} = \frac{1}{2}(I + \mathbb{P}')$ where $I$ is the identity matrix and $\mathbb{P}'$ is the Markov kernel corresponding to the non-lazy random walk (that is, the Markov chain conditioned on moving to a neighboring state at each step). All eigenvalues of any Markov kernel (and in particular, $\mathbb{P}'$) have magnitude at most 1, so all eigenvalues of $\mathbb{P}$ lie in $[0, 1]$. Thus, $Q$ is positive semi-definite. $\qquad\square$

The next lemma is a key technical ingredient in the proof of Theorem 4.2; it is a *local stationarity / metastability*-type bound. It shows that the Dirichlet energy $\mathcal{E}(\nu_r/\mu, \nu_r/\mu)$ must be small at average timesteps $r$, by showing that this form measures the change in $\chi^2$-divergence. See Balasubramanian et al. (2022); Liu et al. (2024b) for similar bounds and additional references.[24]

**Lemma I.8.** *For any $T \in \mathbb{N}$, it holds that*

$$\sum_{r=0}^{T-1} \mathcal{E}\left(\frac{\nu_r}{\mu}, \frac{\nu_r}{\mu}\right) \leq 4H\kappa^2.$$

**Proof.** Since $Q$ is positive semi-definite (Lemma I.7), the square-root of $Q$ is well-defined. Define matrix $A := D^{-1/2}Q^{1/2}D^{1/2}$ and, for each $0 \leq t \leq 2T$, define function $g_t : \mathcal{T} \to \mathbb{R}$ by $g_t := A^t\frac{\nu_0}{\mu}$ (i.e. for any node $v$, $g_t(v) := e_v^\top A^t \frac{\nu_0}{\mu}$). Then, by Lemma I.6,

$$\begin{aligned}
\frac{1}{2}\mathcal{E}(g_t, g_t) &= \langle g_t, (I - \mathbb{P})g_t\rangle_\mu \\
&= \langle g_t, g_t\rangle_\mu - \langle Ag_t, Ag_t\rangle_\mu \\
&= \langle g_t, g_t\rangle_\mu - \langle g_{t+1}, g_{t+1}\rangle_\mu
\end{aligned}$$

where the second equality uses the fact that $g_t^\top D\mathbb{P}g_t = g_t^\top D^{1/2}QD^{1/2}g_t = g_t^\top A^\top DAg_t$. Therefore

$$\frac{1}{2}\sum_{t=0}^{2T-1} \mathcal{E}(g_t, g_t) = \langle g_0, g_0\rangle_\mu - \langle g_{2T}, g_{2T}\rangle_\mu \leq \langle g_0, g_0\rangle_\mu - 1 = D_{\chi^2}(\nu_0 \parallel \mu)$$

by telescoping and the fact that

$$\langle g_{2t}, g_{2t}\rangle_\mu = (\frac{\nu_0}{\mu})^\top (A^{2t})^\top DA^{2t}\frac{\nu_0}{\mu} = (\frac{\nu_0}{\mu})^\top D\mathbb{P}^{2t}\frac{\nu_0}{\mu} = \nu_0^\top \mathbb{P}^t D^{-1}(\mathbb{P}^t)^\top \nu_0 = D_{\chi^2}(\nu_t \parallel \mu) + 1$$

for any nonnegative integer $t$. The penultimate equality uses the fact that $D\mathbb{P} = \mathbb{P}^\top D$ by reversibility. Next, since $\mathcal{E}(g_t, g_t) \geq 0$ for all $t$, it holds that

$$\sum_{r=0}^{T-1} \mathcal{E}(g_{2r}, g_{2r}) \leq 2D_{\chi^2}(\nu_0 \parallel \mu).$$

Now observe that for any even $t = 2r$,

$$g_t = A^{2r}\frac{\nu_0}{\mu} = A^{2r}D^{-1}\nu_0 = D^{-1/2}Q^r D^{-1/2}\nu_0 = D^{-1}(\mathbb{P}^r)^\top \nu_0 = \frac{\nu_r}{\mu}.$$

Here, again we used the reversibility via $Q^r = (Q^\top)^r$. Thus,

$$\sum_{r=0}^{T-1} \mathcal{E}\left(\frac{\nu_r}{\mu}, \frac{\nu_r}{\mu}\right) \leq 2D_{\chi^2}(\nu_0 \parallel \mu).$$

---

[24]These works use KL-divergence as the potential rather than $\chi^2$-divergence, and so they get that $\mathcal{E}(\nu_r/\mu, \log(\nu_r/\mu))$ is small on average rather than $\mathcal{E}(\nu_r/\mu, \nu_r/\mu)$. For our purposes, it will be slightly more convenient to have bounds on the latter quantity.

To complete the proof we observe that $D_{\chi^2}(\nu_0 \parallel \mu) \leq \frac{1}{\mu(\varnothing)} \leq 4H\kappa^2$ by Lemma I.5 and the fact that $\nu_0$ only puts mass on the root node $\varnothing$. $\qquad\square$

We will use Lemma I.8 to show that at a typical timestep $t$, the density ratio $\nu_t/\mu$ is roughly constant on (most of) the autoregressive tree $\mathcal{T}$—namely, on the subgraph carved out around the root node by the "bad" sets $\mathcal{B}_{h,\eta}$. The following lemma formally relates the deviations in the density ratios on this subgraph to the Dirichlet energy at a particular timestep.

**Lemma I.9.** *Let $\eta > 0$ and fix any $r \in [T]$. Then,*

$$\sum_{y_{1:H} \in \mathcal{A}^H} \mu(y_{1:H}) \left[ \left( \frac{\nu_r(y_{1:H})}{\mu(y_{1:H})} - \frac{\nu_r(\varnothing)}{\mu(\varnothing)} \right)^2 \mathbb{1}[\forall h \leq H : y_{1:h} \notin \mathcal{B}_{h,\eta}] \right] \leq \frac{8H\kappa^2}{\eta^2} \mathcal{E}\left( \frac{\nu_r}{\mu}, \frac{\nu_r}{\mu} \right).$$

**Proof.** For any $h \in [H]$ and $y_{1:h} \in \mathcal{A}^h \setminus \mathcal{B}_{h,\eta}$, write $v = y_{1:h}$ and $u = y_{1:h-1}$. Then we have

$$\frac{\mu(y_{1:h})\mathbb{P}(y_{1:h-1} \mid y_{1:h})}{\sum_{y_{h+1:H} \in \mathcal{A}^{H-h}} \mu(y_{1:H})} = \frac{f(y_{1:h}, y_{1:h-1})/(2Z_f)}{\sum_{y_{h+1:H} \in \mathcal{A}^{H-h}} f(y_{1:H}, y_{1:H-1})/Z_f}$$

$$= \frac{\pi_{\mathsf{ref}}(y_{1:h})\widehat{V}(y_{1:h})}{2\sum_{y_{h+1:H} \in \mathcal{A}^{H-h}} \pi_{\mathsf{ref}}(y_{1:H})\widehat{V}(y_{1:H})}$$

$$\geq \frac{\eta^2}{8\kappa^2} \qquad (46)$$

where the final inequality is by definition of $\mathcal{B}_{h,\eta}$ (Definition I.5). Thus,

$$\sum_{y_{1:H} \in \mathcal{A}^H} \mu(y_{1:H}) \left( \frac{\nu_r(y_{1:H})}{\mu(y_{1:H})} - \frac{\nu_r(\varnothing)}{\mu(\varnothing)} \right)^2 \mathbb{1}[\forall h \leq H : y_{1:h} \notin \mathcal{B}_{h,\eta}]$$

$$\leq H \sum_{y_{1:H} \in \mathcal{A}^H} \mu(y_{1:H}) \sum_{k=1}^{H} \left( \frac{\nu_r(y_{1:k})}{\mu(y_{1:k})} - \frac{\nu_r(y_{1:k-1})}{\mu(y_{1:k-1})} \right)^2 \mathbb{1}[y_{1:k} \notin \mathcal{B}_{k,\eta}]$$

$$= H \sum_{k=1}^{H} \sum_{y_{1:k} \in \mathcal{A}^k} \left( \sum_{y_{k+1:H} \in \mathcal{A}^{H-k}} \mu(y_{1:H}) \right) \left( \frac{\nu_r(y_{1:k})}{\mu(y_{1:k})} - \frac{\nu_r(y_{1:k-1})}{\mu(y_{1:k-1})} \right)^2 \mathbb{1}[y_{1:k} \notin \mathcal{B}_{k,\eta}]$$

$$\leq \frac{8H\kappa^2}{\eta^2} \sum_{k=1}^{H} \sum_{y_{1:k} \in \mathcal{A}^k} \mu(y_{1:k})\mathbb{P}(y_{1:k-1} \mid y_{1:k}) \left( \frac{\nu_r(y_{1:k})}{\mu(y_{1:k})} - \frac{\nu_r(y_{1:k-1})}{\mu(y_{1:k-1})} \right)^2 \mathbb{1}[y_{1:k} \notin \mathcal{B}_{k,\eta}]$$

$$\leq \frac{8H\kappa^2}{\eta^2} \cdot \mathcal{E}(\frac{\nu_r}{\mu}, \frac{\nu_r}{\mu})$$

where the second inequality is by Eq. (46). $\qquad\square$

The preceding lemma shows that (at timesteps where the Dirichlet energy is small), the density ratios $\nu_r/\mu$ are near-constant on a certain subgraph of the tree (which, via Lemma I.4, we will show is "most" of the tree, under the measure of $\pi^\star$). However, ultimately we wish to show that $\nu_r/\mu$ is *not too small* on most of the tree (which we will use to lower bound $\nu_r/\pi^\star$). The following lemma ensures this by lower bounding the density ratio at the root node $\varnothing$.

Specifically, it makes formal the intuition that if (1) the random walk starts at $\varnothing$, and (2) the stationary distribution puts reasonable mass on $\varnothing$, then the marginal law of the random walk at any particular timestep should put reasonable mass on the root—even before the random walk mixes to the stationary distribution. This, of course, requires the walk to be sufficiently lazy, which implicitly comes into the proof via the fact that $\mathbb{P}$ is similar to a PSD matrix $Q$ (Lemma I.7).

**Lemma I.10.** *For any non-negative integer $t$, it holds that $\frac{\nu_t(\varnothing)}{\mu(\varnothing)} \geq 1$.*

**Proof.** Define $g : \mathcal{T} \to \mathbb{R}$ by $g(x) := \frac{\nu_0(x) - \mu(x)}{\sqrt{\mu(x)}}$. Then since $Q$ is positive semi-definite (Lemma I.7), we have that $Q^t$ is positive semi-definite and hence

$$
\begin{aligned}
0 &\leq g^\top Q^t g \\
&= (\nu_0 - \mu)^\top D^{-1/2} Q^t D^{-1/2} (\nu_0 - \mu) \\
&= (\nu_0 - \mu)^\top \mathbb{P}^t D^{-1} (\nu_0 - \mu) \\
&= \nu_t^\top D^{-1} \nu_0 - \mu^\top D^{-1} \nu_0 - \nu_t^\top D^{-1} \mu + \mu^\top D^{-1} \mu \\
&= \mathop{\mathbb{E}}_{y \sim \nu_t} \frac{\nu_0(y)}{\mu(y)} - \mathbb{1}^\top \nu_0 - \nu_t^\top \mathbb{1} + \mu^\top \mathbb{1} \\
&= \mathop{\mathbb{E}}_{y \sim \nu_t} \frac{\nu_0(y)}{\mu(y)} - 1
\end{aligned}
$$

where the third equality uses the fact that $\mu = \mathbb{P}^\top \mu$. Rearranging terms and observing that $\nu_0(y) = \mathbb{1}[y = \varnothing]$ completes the proof. $\qquad\square$

To formally compare distributions on $\mathcal{T}$ with $\pi^\star$ (which is a distribution on the leaves of $\mathcal{T}$), it is convenient to zero-extend $\pi^\star$ to $\mathcal{T}$:

**Definition I.9** (Extension of $\pi^\star$ to $\mathcal{T}$). *Let $\mu^\star \in \Delta(\mathcal{T})$ be the distribution on $\mathcal{T}$ that agrees with $\pi^\star$ on the leaves and is zero on the internal nodes.*

We show that $\mu^\star$ is not too far from $\mu$ in $\chi^2$-divergence, which is equivalent to an average-case density ratio bound:

**Lemma I.11.** *It holds that*

$$
1 + D_{\chi^2}(\mu^\star \parallel \mu) \leq 2H\kappa^2.
$$

**Proof.** We have

$$
\begin{aligned}
1 + D_{\chi^2}(\mu^\star \parallel \mu) &= \mathop{\mathbb{E}}_{y_{1:H} \sim \pi^\star} \left[ \frac{\pi^\star(y_{1:H})}{\mu(y_{1:H})} \right] \\
&= Z_f \mathop{\mathbb{E}}_{y_{1:H} \sim \pi^\star} \left[ \frac{\pi^\star(y_{1:H})}{f(y_{1:H}, y_{1:H-1})} \right] \\
&= Z_f \mathop{\mathbb{E}}_{y_{1:H} \sim \pi^\star} \left[ \frac{\pi^\star(y_{1:H})}{\pi_{\mathsf{ref}}(y_{1:H}) \widehat{V}(y_{1:H})} \right] \\
&= \frac{Z_f}{V_{\mathtt{tilt}}^\star(\varnothing)} \mathop{\mathbb{E}}_{y_{1:H} \sim \pi^\star} \left[ \frac{V_{\mathtt{tilt}}^\star(y_{1:H})}{\widehat{V}(y_{1:H})} \right] \\
&\leq 2H\kappa^2
\end{aligned}
$$

where the fourth equality is by definition of $V_{\mathtt{tilt}}^\star$, and the final inequality is by Lemma I.5 and Eq. (44). $\qquad\square$

We can now formally conclude the proof of Theorem 4.2.

**Proof of Theorem 4.2.** Observe that the distribution of $y^{(t)}$ in Algorithm 1 is precisely $\nu_t$ for each $0 \leq t \leq T$, and moreover the distribution of the output is $\nu := \frac{1}{T} \sum_{t=1}^T \nu_t$. Set $\eta := \delta/(6H)$ and $\delta' := \delta/(48H\kappa^2)$. Fix any $r \in [T]$. Define events $\mathcal{F}^1, \mathcal{F}^2, \mathcal{F}^3 \subseteq \mathcal{A}^H$ by

$$
\mathcal{F}^1 := \{y_{1:H} : \exists h \leq H, y_{1:h} \in \mathcal{B}_{h,\eta}\},
$$

$$
\mathcal{F}^2 := \{y_{1:H} : \nu_r(y_{1:H})/\mu(y_{1:H}) < 1/2\},
$$

$$
\mathcal{F}^3 := \{y_{1:H} : \mu(y_{1:H})/\pi^\star(y_{1:H}) < 4\delta'\}.
$$

By Lemma I.4 and a union bound over $h \in [H]$, we have

$$
\mathop{\Pr}_{y_{1:H} \sim \pi^\star} [y_{1:H} \in \mathcal{F}^1] \leq H\eta. \tag{47}
$$

Next, for any $y_{1:H} \in \mathcal{F}^2 \setminus \mathcal{F}^1$, observe that $y_{1:H}$ satisfies

$$\left( \frac{\nu_r(y_{1:H})}{\mu(y_{1:H})} - \frac{\nu_r(\varnothing)}{\mu(\varnothing)} \right)^2 \mathbb{1}[\forall h \leq H : y_{1:h} \notin \mathcal{B}_{h,\eta}] > \left( \frac{1}{2} - 1 \right)^2 = \frac{1}{4}$$

by definition of the events $\mathcal{F}^1, \mathcal{F}^2$ and by Lemma I.10. It follows from Lemma I.9 that

$$\Pr_{v \sim \mu} \left[ (|v| = H) \wedge (v \in \mathcal{F}^2 \setminus \mathcal{F}^1) \right] \leq \frac{32 H \kappa^2}{\eta^2} \mathcal{E} \left( \frac{\nu_r}{\mu}, \frac{\nu_r}{\mu} \right). \tag{48}$$

Therefore

$$\begin{aligned}
\Pr_{y_{1:H} \sim \pi^\star}[y_{1:H} \in \mathcal{F}^2 \setminus \mathcal{F}^1] &= \Pr_{v \sim \mu^\star} \left[ (|v| = H) \wedge (v \in \mathcal{F}^2 \setminus \mathcal{F}^1) \right] \\
&= \mathbb{E}_{v \sim \mu} \left[ \frac{\mu^\star(v)}{\mu(v)} \cdot \mathbb{1} \left[ |v| = H \wedge v \in \mathcal{F}^2 \setminus \mathcal{F}^1 \right] \right] \\
&\leq \sqrt{(1 + D_{\chi^2}(\mu^\star \| \mu)) \Pr_{v \sim \mu}[(|v| = H) \wedge (v \in \mathcal{F}^2 \setminus \mathcal{F}^1)]} \\
&\leq \frac{8 H \kappa^2}{\eta} \sqrt{\mathcal{E} \left( \frac{\nu_r}{\mu}, \frac{\nu_r}{\mu} \right)} \tag{49}
\end{aligned}$$

where the first inequality is by Cauchy-Schwarz and the second inequality is by Lemma I.11 and Eq. (48). Finally,

$$\Pr_{y_{1:H} \sim \pi^\star}[y_{1:H} \in \mathcal{F}^3] = \Pr_{y_{1:H} \sim \pi^\star} \left[ \frac{\pi^\star(y_{1:H})}{\mu(y_{1:H})} > \frac{1}{4\delta'} \right] \leq 8 H \kappa^2 \delta' \tag{50}$$

by Lemma I.11 and Markov's inequality. In the event that neither of $\mathcal{F}^2, \mathcal{F}^3$ occur, we have $\nu_r(y_{1:H}) \geq 2\delta' \pi^\star(y_{1:H})$. Thus, by Eqs. (47), (49) and (50),

$$\Pr_{y_{1:H} \sim \pi^\star} \left[ \frac{\nu_r(y_{1:H})}{\pi^\star(y_{1:H})} < 2\delta' \right] \leq H\eta + \frac{8 H \kappa^2}{\eta} \sqrt{\mathcal{E} \left( \frac{\nu_r}{\mu}, \frac{\nu_r}{\mu} \right)} + 8 H \kappa^2 \delta'.$$

Recall that $r \in [T]$ was chosen arbitrarily. Since $\nu = \frac{1}{T} \sum_{r=1}^{T} \nu_r$, it follows from Lemma F.3, applied to the random variables $Z_i := \nu_i(y_{1:H})/\pi^\star(y_{1:H})$ (for $y_{1:H} \sim \pi^\star$), that

$$\begin{aligned}
\Pr_{y_{1:H} \sim \pi^\star} \left[ \frac{\nu(y_{1:H})}{\pi^\star(y_{1:H})} < \delta' \right] &\leq 2H\eta + 16 H \kappa^2 \delta' + \frac{16 H \kappa^2}{\eta T} \sum_{r=1}^{T} \sqrt{\mathcal{E} \left( \frac{\nu_r}{\mu}, \frac{\nu_r}{\mu} \right)} \\
&\leq 2H\eta + 16 H \kappa^2 \delta' + \frac{16 H \kappa^2}{\eta T} \sqrt{T \cdot \sum_{r=1}^{T} \mathcal{E} \left( \frac{\nu_r}{\mu}, \frac{\nu_r}{\mu} \right)} \\
&\leq 2H\eta + 16 H \kappa^2 \delta' + \frac{16 H \kappa^2}{\eta} \sqrt{\frac{4 H \kappa^2}{T}} \\
&\leq \delta
\end{aligned}$$

where the third inequality is by Lemma I.8, and the fourth inequality is by choice of $\eta$ and $\delta'$, and holds so long as $T \geq C H^5 \kappa^6 \delta^{-4}$ for a sufficiently large constant $C$. Substituting in the choice of $\delta' := \delta/(48 H \kappa^2)$ and our definition of $\kappa := 1 + \varepsilon_V$ into the left-hand side of the inequality completes the proof of Eq. (7). The claim that the runtime of VGB is $O(T \cdot |\mathcal{A}|)$ is immediate from inspection of Algorithm 1. $\qquad \square$

## J LLM USAGE STATEMENT

We used LLMs to help us write and refine small fragments of code—specifically, code for programmatically generating Matplotlib and TikZ figures—and as a general-purpose aide for learning syntax for specific Python libraries.

