# OpenReview forum: "Taming Imperfect Process Verifiers: A Sampling Perspective on Backtracking"
_ICLR.cc/2026/Conference — ICLR 2026 Poster_

### Official Review · Reviewer_s7CV · 2025-10-16

**Soundness:** 4
**Presentation:** 3
**Contribution:** 3
**Rating:** 8
**Confidence:** 3

**Summary:**

In test-time scaling, process verifiers guide generation towards high-reward outputs. However, process verifiers may be inaccurate. This paper shows that process verification errors can "cascade" in the sense of growing with the number of decoding steps. It then presents an alternative test-time scaling algorithm, VGB, which is shown to have positive theoretical properties with regard to multiplicative errors in the process reward:

- Given a uniform bound on the multiplicative process reward error, VGB is guaranteed to sample from a distribution $\hat{\pi}$ with bounded total variation distance from the reward-optimal sampling distribution $\pi^*$, and to output a "leaf node" (a complete generation) in time $\tilde{O}(H^3)$.
- Given a bound on the expected ratio of the estimated and true process rewards, the output distribution is similar to the reward-optimal sampling distribution with high probability. Specifically the ratio $\pi*/\hat{\pi} < \delta^{-1} 48H(1+\epsilon_V)^2$, with $H$ indicating the decoding length and $(1+\epsilon_V)$ the multiplicative bound on the expected ratio of estimated and true process rewards. The runtime in this setting grows as $\tilde{O}(H^5)$.

The paper makes several other theoretical contributions, including showing that other error conditions are insufficient to achieve the desired guarantees (appendix D).

The empirical side focuses mainly on minimal synthetic language tasks, such as Dyck grammars.

**Strengths:**

The paper addresses an important practical issue in test-time scaling. It introduces useful theoretical machinery, which to my knowledge is novel in this domain. The discussion of "pitfalls" of existing techniques is clear and convincing, as is the presentation of the theoretical guarantees offered by the approach.

The empirical results shown in the main paper mostly support the main claims, albeit in toy settings: fig 1 demonstrates that VGB dominates block best-of-N and block rejection sampling in the Dyck grammar task, and fig 2 shows that the KL-divergence error grows more slowly with $H$ in the ABC task.

Overall, while I doubt that the specific algorithm here is likely to be of practical value, the paper seems like a significant theoretical advance that will catalyze more research that may ultimately make test-time scaling more effective and/or efficient.

The writing quality is excellent throughout the paper.

**Weaknesses:**

While the paper claims to address the problem of cascading errors, it seems like the bound in theorem 4.2 still grows in $H$. Similarly, the KL-divergence errors seem to grow with $H$ in Figure 2, although they do grow more slowly than with action reject sampling. Thus it seems more like the proposed approach *mitigates* this issue rather than solving it.

I think the proposed algorithm is unlikely to be adopted due to the polynomial run-time scaling with $H$ (cubic with the uniform guarantee, and quintic with the expectation condition). But as noted above, that's not a deal-breaker to me because the paper may stimulate research that results in more efficient algorithms. The authors do acknowledge this issue in the paper.

I am not a specialist in this area, and I found it pretty hard to understand how the algorithm actually works. In particular, the paper did a poor job explaining the search space explored by the backtracking algorithm, and would have benefited from a better figure than fig 1 (right).

**Questions:**

Please feel free to comment on the points raised above: (1) whether errors still increase with $H$ in VDB, and (2) whether and how the algorithm can be practical given the apparently poor polynomial scaling with $H$.

---

> ### Author Response · Authors · 2025-11-20
> **Response to Reviewer s7CV**
>
> We thank the reviewer for their time and attention! Overall, we are glad that the reviewer appreciated our work, and we totally agree with the reviewer’s assessment on scalability. While VGB itself may not be suitable for large-scale applications (indeed, our experiments already modify it slightly so that it empirically scales quadratically in the horizon rather than cubically – see Appendix E.1), we believe that our work and our theoretical framing will hopefully stimulate further research in this area, as the reviewer suggests.
>
> Regarding the reviewer’s other questions/concerns:
>
> **[W1] (Solving vs. mitigating the problem of cascading errors).** Broadly speaking, we agree with the reviewer that our approach “mitigates” rather than “solves” the issue of error amplification. Of course, there are several nuances, which we discuss below:
>
> First, in the setting of Theorem 4.1 (the guarantee for VGB under a **worst-case** error bound), we would argue that VGB completely solves the error amplification issue: the error can be made arbitrarily small (and independent of H), albeit at the cost of increasing the runtime by poly(H). This is the backing for e.g. our claims in line 53. Of course, the worst-case error bound is likely unrealistic in practice.
>
> Second, in the setting of Theorem 4.2 (the guarantee for VGB under an **average-case** error bound), we would tend to agree that the issue of error amplification is only “mitigated”, since the density ratio bound scales linearly in H. Note that the baseline is action-level rejection sampling, for which the density ratio bound may scale exponentially in H, so this is a substantial improvement. Under the average-case error bound, we believe that the linear scaling in H is likely necessary for any poly(H)-time algorithm that seeks to cover the true distribution. However, as shown in Appendix B, if the goal is only reward maximization, then one can once again trade computation for error via best-of-N, and achieve an arbitrarily small (and horizon-independent) regret bound.
>
> Third, in the experiments (e.g. Figure 2), we agree that the errors of VGB do seem to grow with H, but more slowly than action-level sampling. This is in line with our intuition that Assumption 4.2 is likely more realistic than Assumption 4.1.
>
> **[W2] (Polynomial run-time scaling).** As discussed above, we agree that a primary contribution of our work is to propose a clean yet fruitful theoretical framing, and hopefully stimulate (and guide) further empirical research. We make two additional remarks about scaling (see also our response to Reviewer ZxSD).
>
> First, our implementation of VGB (Appendix E.1) seems to empirically scale roughly quadratically in $H$, since we run the Markov chain once, until it hits a leaf, rather than restarting repeatedly. It is also straightforward to modify the theory to get an $O(H^2)$ overall time guarantee, by increasing the self-loop transition probabilities at the leaves; this technique was originally proposed by (Hayes & Sinclair, 2010) in the context of the original work of Sinclair and Jerrum – we will mention this in the final version of the paper.
>
> Second, in long-horizon tasks, LLM search methods are typically applied at the block level rather than the token level, and the number of blocks is often on the order of $16$ to $32$. This could be done for VGB as well (and the theory would carry over unchanged), and the runtime would then scale roughly as $O(HK)$ where $K$ is the number of blocks, i.e. only a $K$ factor worse than action-level sampling. We believe this is not out of the realm of practicality, though certainly a better runtime would be preferable.
>
> **[W3] (Intuition for algorithm)** We hope that the high-level idea of the algorithm (delete tokens if the verifier says they are bad) is fairly intuitive, but we agree that the precise choice of random walk probabilities may be less intuitive, and we are very open to comments on how it might be better explained.
>
> To try to clarify the “search space explored by the backtracking algorithm”: the search space is a “prefix tree” (i.e. a trie). The root node is the empty sequence. Each other node is some non-empty sequence, and the children of a node are the sequences obtained by appending one token to the current sequence.
>
> We also have a more detailed figure (Figure 21) which is in the appendices due to space constraints; we will move it to the main body in the final version of the paper.
>
> Please let us know if there are any specific confusions that we can help clarify. Though we think that the generation tree is a good mental model, we agree that it is abstract and perhaps could be better instantiated in particular settings.

---

### Official Review · Reviewer_HyCr · 2025-10-28

**Soundness:** 3
**Presentation:** 3
**Contribution:** 3
**Rating:** 8
**Confidence:** 2

**Summary:**

The paper proposes a generalization of the Sinclair-Jerrum random walk to reward-guided generation of language models when the value function is not perfect.
The proposed method (VGB) has a strong theoretical motivation, and where possible, formula derivations are delegated to relevant work.

**Strengths:**

+ Solid theoretical foundation backed up by experimental results
+ Simple examples that illustrate the problem of a more greedy approach (ActionLevelRS)
+ Thought out collection of evaluation tasks (such as parity that is not in $\mathrm{AC}^0$)
+ Polynomial (in generation length, action space, value function error, and target $D_{\mathrm{TV}}$) algorithm for sampling from an optimal policy for a KL regularized objective

**Weaknesses:**

+ Minor: Relatively simple generation tasks (Dyck grammar, Python test generation, and letter avoidance with output length 32 and  0.5B parameter model). To be more precise, inclusion of simple tasks benefits the work; however, inclusion of more complex tasks should further help to assess the practicality of the method.
+ Minor: The paper delegates a lot of proofs to the Appendix. To be more precise, the Appendix consists of around 55 pages, and around 25 of them are proofs. This is a highly dense, in particular for a conference where "Authors may use as many pages of appendices (after the bibliography) as they wish, but reviewers are not required to read the appendix.". I must say I haven't managed to check all the proofs of the propositions and theorems. However, I noted that some of them seem to miss some (relatively obvious so this is a minor problem) assumptions or have some typos regarding the parentheses (see proof of proposition G.3).
+ Minor: the algorithm requires a lot of sampling to achieve theoretical guarantees

**Questions:**

Do authors have any intuition or results regarding more complex problems, especially those that have a value function that is much harder to learn/approximate?
To be more precise, the algorithm still requires a good, on average, value function for efficient sampling, and it is not clear whether, in more complex cases, it wouldn't be cheaper to tune the policy with GRPO instead.

---

> ### Author Response · Authors · 2025-11-20
> **Response to Reviewer HyCr**
>
> We thank the reviewer for their time and attention! We will take a look for any typos and missing references to assumptions; thanks for pointing out the missing parenthesis. Overall, we are glad that the reviewer appreciated our work – we hope that our work will inspire further research into more scalable methods that can be applied to complex tasks. Below, we respond the the reviewer’s individual questions/concerns:
>
> **[W1] (Relatively simple experimental tasks).** While our experimental tasks are indeed relatively small scale, we have done our best to at least include a diverse range of tasks/models. We emphasize that our main goal is to demonstrate that the algorithm design space of verifier-guided generation is (both theoretically and empirically) rich and interesting due to the challenge of mitigating error amplification – and we believe our theory and experiments achieve this goal.
>
> **[W3] (Runtime).** Indeed, we believe it is an important open question to understand whether more efficient algorithms can achieve comparable guarantees to VGB. Our paper lays the groundwork to study this question.
>
> **[Q1] (Intuition about more complex problems / comparison with GRPO)** This is a good question, and raises many subtleties. We are by no means the first to experiment with using value functions to guide generation. Our general belief, from the burgeoning empirical literature on this topic, is that there seems to be some tangible empirical benefit of verifier-guided generation over just fine-tuning (though this is by no means settled science, and we’re not aware of e.g. a thorough compute-controlled study). Of course, even if this is the case, the theoretical tradeoff remains essentially unexplored – we believe this is an interesting open question. We make several more concrete remarks:
>
> First, learning a perfect value function is in some sense as hard as learning a perfect policy, but in practice one might hope that even a fairly “weak” verifier for some specific task can improve performance of a general-purpose language model (on that task). This is one motivation for our model that allows imperfections in the value function. We believe that even under our assumptions (multiplicative error etc.), there are likely settings where learning an imperfect (but still good enough for VGB) value function is computationally easier than directly fine-tuning a good policy. In particular, we believe that (something like) the **parity** task should provide one concrete example, where the ability of VGB to backtrack enables avoiding the need to solve a computationally hard task. Of course, this is a synthetic example, and we believe this direction is worth exploring further (theoretically and empirically).
>
> Second, there are some practical frameworks where training a verifier is possible but fine-tuning the generator is not: for instance, if we only have an API that allows query access to the generator. This corresponds to the fact that verifier-guided generation is an inference-time method whereas fine-tuning is not.
>
> We believe our work provides a useful perspective for investigating these tradeoffs.

---

> ### Comment · Reviewer_HyCr · 2025-11-25
>
> Thank you for the comments. I agree with the authors and reviewer s7CV that this is an important theoretical contribution that paves the way for future research. As my concerns have been addressed, I maintain my positive assessment of the paper, albeit with increased confidence.

---

### Official Review · Reviewer_ZxSD · 2025-10-31

**Soundness:** 3
**Presentation:** 3
**Contribution:** 2
**Rating:** 4
**Confidence:** 3

**Summary:**

This paper investigates the robustness of process-guided test-time alignment in generative language models, focusing on settings where imperfect value functions (verifiers) are used to steer generation. The authors identify error amplification pitfalls associated with standard action-level sampling when value-function estimates are used, and propose a new algorithm, Value-Guided Backtracking (VGB), grounded in Markov Chain Monte Carlo theory to counteract these weaknesses. They provide theoretical analyses and guarantees for VGB—even under average-case error regimes—as well as empirical validation across synthetic, code generation, and constrained text generation benchmarks.

**Strengths:**

- The authors provide an explicit theoretical understanding of how small, seemingly benign errors in learned process verifiers can catastrophically degrade the distributional fidelity of generation under standard action-level sampling.
- VGB is a principled modification to existing sampling strategies, and the authors convincingly root its correctness and performance in established Markov chain (Sinclair-Jerrum walk) theory.
- The paper benchmarks VGB against both sampling (Block Best-of-N, Block Rejection Sampling) and search-style (beam search, locally constrained decoding) baselines across multiple axes.

**Weaknesses:**

- While the theoretical grounding is new, its real-world applicability and improvement magnitude over clever heuristics or simple extensions (e.g., adding "undo" moves) is not novel, especially since the best practical variant of VGB may require hyperparameter tuning for the backtracking probability, which the authors acknowledge but do not fully explore.
- The computational cost of VGB, especially for long horizons (scaling as $\tilde{O}(H^3)$ for guaranteed leaf sampling under uniform error, and worse under average-case errors), could render it prohibitive for many realistic LLM tasks beyond toy or controlled environments.
- Most evaluation focuses on (relatively) small-scale or synthetic tasks—such as the Dyck grammar and ABC tasks. Despite one code generation task, there are no results on multi-turn question answering, math and code, or agentic reasoning where process verifiers are essential and imperfect. This leaves open how VGB would fare in "hard" settings.

**Questions:**

- Can the authors rigorously clarify the scalability of VGB for long-horizon, real-world LLM settings, both in terms of query count and wall-clock time, beyond what is presented in synthetic or small-scale experiments? For example, can VGB be realistically run for open-ended generation and complex code synthesis tasks at scale, or do computational bottlenecks dominate in these regimes?
- Would the main conclusions regarding error amplification and backtracking hold if the value function $\widehat{V}$ were implemented with more complex architectures (e.g., transformer-based verifiers) or noisy/biased sampling of partial completions? Are the present synthetic experiments sufficiently representative?
- Is there a principled way to tune the backtracking parameter (probability or weight), especially in the absence of oracle rewards or in scenarios where the true error profile of $\widehat{V}$ is unknown?
- Do the authors have insight into failure modes for VGB in more diverse or adversarial settings (e.g., prompts specifically constructed to trip up process verifiers), and could VGB be made robust to such cases?

---

> ### Author Response · Authors · 2025-11-20
> **Response to Reviewer ZxSD (Part 1 of 2)**
>
> We thank the reviewer for their time and attention! Below we respond to the reviewer’s concerns and questions:
>
> **[W1] (Undo moves are not novel).** While backtracking strategies have been explored empirically, we emphasize that the main contribution of our work is to show for the first time that a *principled* strategy based on backtracking leads to provable benefits, thereby putting the study of such heuristics on solid foundations for future work. We also remark that the practical version of VGB, with a **trained** value function, **does not require hyperparameter tuning** (see response Q3 below), in contrast with prior ad-hoc backtracking strategies. The “backtracking weight” $\alpha$ in the constrained text generation task is a parameter of the **untrained** value function that is given to VGB (and in any case our result in Figure 3 demonstrates that the performance of VGB is robust to the choice of this parameter).
>
> **[W2/W3/Q1] (Scalability of VGB to complex settings).** First, we view our work as primarily theoretical in nature, and we believe that the theory stands on its own; the experiments demonstrate that the theoretically-designed algorithms have promise beyond the formal theoretical assumptions. While the specific algorithm that we propose may be unlikely to be adopted “at scale”, the key contribution of our work is to demonstrate a theoretical model for imperfect process verifiers in which non-trivial algorithmic guarantees are possible – and which therefore can usefully guide algorithm design. As Reviewer s7CV suggests, we believe this lays the groundwork for future research into scaling up similar algorithms.
>
> Second, to clarify the scalability of VGB specifically, we found experimentally that **VGB (as implemented) scales roughly quadratically in the number of “generation steps”**. Compared to the theory, this is because we run the Markov chain continuously until it hits a leaf, rather than restarting after T steps if a leaf was not reached. It is also straightforward to improve the **theoretical runtime to $O(H^2)$** by increasing the weights of the self-loops at the leaves, a technique originally introduced by (Hayes & Sinclair, 2010); we will mention this in the final version.
>
> For the experiments in our paper, we took each “generation step” to be a single token, but for long-horizon LLM tasks, one could just as easily apply our algorithm with **a single step corresponding to a chunk of tokens**. This is commonly done in LLM search methods such as beam search – even for very long-horizon tasks, the number of chunks is often chosen to be as small as K=16 or 32. In such cases, we expect that VGB will only pay a factor of ~K (over baseline action-level rejection sampling) in query count and wall-clock time. This could enable smoothly improving scalability at the cost of (potentially) worse errors.
>
> Third, our experiments also demonstrate (Figures 5 and 6) that VGB can sometimes be **more efficient** than baselines, measured in terms of wall-clock time, when the goal is to find a reward-1 response. While this experiment is synthetic, we believe that the underlying phenomenon (value being easier to estimate at some steps than others) is  very natural and may appear in much more complex, real-world tasks.
>
> **[Q2] (Generality of conclusions).** We emphasize that our theoretical results are **architecture-agnostic**. If the approximate value is multiplicatively close to the true value, then the theoretical conclusions hold regardless of architecture and how the value function was learned. Our experimental results, while somewhat small scale, validate this assumption across **a diverse collection of tasks and architectures**: the Python test case generation experiment uses a transformer-based value function; the synthetic tasks use an MLP-based value function; and the constrained text generation task uses the most common “explicit” value function used in the literature (a heuristic where we set the value to 1 if the current prefix satisfies the given constraint, and 0 otherwise).

---

> ### Author Response · Authors · 2025-11-20
> **Response to Reviewer ZxSD (Part 2 of 2)**
>
> **[Q3] (Hyperparameters).** We emphasize that the main version of our algorithm implemented in our experiments (Algorithm 2) is **entirely hyperparameter free** in small-alphabet settings (in large-alphabet settings, a “batch size” hyperparameter K is needed, but all inference-time algorithms have such a parameter; see Appendix E.1). This actually **contrasts favorably with prior heuristics** for inference-time backtracking (Botta et al., 2025), which did require several hyperparameters to tune the backtracking likelihoods (albeit in a setting with a slightly different type of process verifier).
>
> The “backtracking probability” parameter $\alpha$ in Section 5.3 should be interpreted as something that can lead to **additional improvements beyond what is suggested by the theory**, not as a key parameter required for the algorithm to succeed at all. Specifically, this parameter only appears in the constrained text generation setting (Section 5.3) where one has some measure of the “correctness of past tokens” (rather than training an estimate of the “expected correctness of future tokens”, which is the focus of most of the paper). In this setting, we believe one should choose $\alpha$ roughly as the “probability of not making an error at each step”. However, our experiment shows that the performance of VGB is quite robust to the choice of $\alpha$ (with benefits for any $\alpha \in [0.1, 0.5]$).
>
> **[Q4] (Adversarial settings).** Many of our experiments are explicitly designed to explore different ways in which the prompts or training procedure might “trip up” the process verifier, and **the results demonstrate robustness of VGB**. In the ABC task, we found that the benefits of VGB are robust to training the verifier with very few samples (Figure 2). In the parity task, we found that the benefits of VGB extend to settings where learning the true verifier is hard for gradient descent (Appendix E.3). In the Dyck grammar task, we specifically chose prompts on which the base model has very low probability of success (Table 1, bottom entry), and we found that VGB does well in this scenario.

---

> > ### Comment · Reviewer_ZxSD · 2025-11-27
> >
> > Thank you for the detailed response. The clarification regarding the computational complexity and the implementation details has addressed my primary concerns. I have raised my score to 8 to reflect the strength of the theoretical contribution.
> >
> > Regarding the suggestion to use "chunking" to handle long horizons, I agree it is a viable path for deployment. However, would such an approach re-introduce approximation errors over larger action spaces compared to the token-level approach?

---

> > > ### Author Response · Authors · 2025-11-28
> > >
> > > Thanks for the response and the question! Yes, there are definitely tradeoffs involved in increasing the chunk size.
> > >
> > > From a theoretical perspective, while the effective horizon decreases, the action-level coverage coefficient (line 187) may increase (since it will correspond to how well the base model covers the target model on chunks rather than on individual tokens), which factors linearly into the runtime (Theorem 4.1). Thus, the tradeoff depends on the "strength" of the base model.
> > >
> > > From a practical perspective, one could avoid this runtime factor by fixing the number of rejection sampling iterations K at each step of the algorithm (e.g. just use K=32 as in Section E.1), but this could lead to additional errors if the action-level coverage coefficient exceeds K. We remark that other practical algorithms (including e.g. action-level rejection sampling) likely face the same issue when applied at the chunk level.

---

### Official Review · Reviewer_VY1h · 2025-10-31

**Soundness:** 3
**Presentation:** 2
**Contribution:** 3
**Rating:** 6
**Confidence:** 3

**Summary:**

The authors study "process verifiers" for LLMs within a theoretical framework. The notion of value function is introduced. It is supposed to align LLM generation with a predetermined reward. The "true" value function is defined, as well as its approximate counterpart. The central question of the paper is: how can we sample from the LLM using signal from the approximate verifier so that the resulting distribution of responses remain close to the optimal one achieved with the "true" value function?

A VGB sampling algorithm based on Sinclair-Jerrum random walk is proposed and its theoretical guarantees are proved under certain assumptions.

Additionally, some small-scale experiments are conducted showing the practical usefulness of the method.

**Strengths:**

* I find the topic of the paper interesting and timely.
* The authors put effort into introducing a formal framework of sampling from LLMs with verifiers and seem to prove significant theoretical results.
* The presented VGB algorithm is conceptually simple.

**Weaknesses:**

1. My main objection is that it's really unclear to what extent the presented theoretical results are relevant in practice. The authors motivate the paper practically, mentioning LLM-based PRMs from recent literature. However, I'm not sure if the introduced theoretical framework captures these practical PRMs and their usage. More specifically, I'm not sure if Assumptions 4.1 or 4.2 will be satisfied in practice with real-life verifiers.
2. The presentation in some places is not polished. Some symbols are not defined (for instance delta in l. 64, A^H in l. 65, D_TV and omega in l. 209), which sometimes makes reading / understanding more difficult than necessary.
3. It is good that the authors include some number of experiments, but they are small-scale and targeting simple problems. I think they are not convincing of practical relevance of VGB and its theoretical properties.

**Questions:**

1. In your framework you fix the length of generation to be H. Can the framework be reformulated to assume variable length generation?
2. In VGB the probability of staying in the current node is 1/2. Is this the only constant preserving the properties of VGB, or it can be different?

---

> ### Author Response · Authors · 2025-11-20
> **Response to Reviewer VY1h**
>
> We thank the reviewer for their time and attention! Below we respond to the reviewer’s concerns and questions:
>
> **[W1] (Theoretical framework capturing practice).** We believe our theoretical results stand on their own merits: the broader goal (which we believe our work accomplishes) is to highlight that the algorithm design space for inference-time search methods is rich and interesting, meriting future exploration. Moreover, we believe our **experimental results**, while relatively small scale, address the reviewer’s concern about practical relevance. Our value functions/PRMs are **trained with regression as in practice**. Across a diverse range of tasks/models (see our response Q4 to Reviewer ZxSD), even if the theoretical assumptions may not be exactly satisfied, our algorithms nevertheless have demonstrable empirical benefits when applied with these trained PRMs (whether MLP-based or transformer-based).
>
> Broadly, we would think of Assumptions 4.1 and 4.2 as **yardsticks for guiding algorithm design**. The experiments demonstrate that they are useful yardsticks, and we hope that they will inspire refinements in future work.
>
> **[W2] (Presentation).** Thanks for pointing this out; while these symbols are standard in our subarea of theoretical machine learning, we recognize that they are not universal, and will be sure to add additional background in the Preliminaries section. For example: $\Delta(Y)$ is the space of distributions on set $Y$. $A^H$ is the space of sequences of length $H$ where each character lies in $A$ (we will also clarify that $A$ is the “action space”). $D_{TV}$ is total variation distance. $\Omega$ is from asymptotic big-O notation, so e.g. $\Omega(1)$ refers to “a quantity that is bounded away from 0”.
>
> **[W3] (Experiments).** We agree with the reviewer that VGB itself has not yet been demonstrated for large-scale applications. However, we believe that the theory and experiments do amply demonstrate that there is value in our framing of the problem of dealing with imperfect process verifiers, and that current practical methods leave something “on the table” – i.e., that there is room for algorithmic improvement. As Reviewer s7CV comments, we believe that our paper will stimulate research into more practical and scalable algorithms with similar guarantees. See also our response to W1.
>
> **[Q1] (Variable length of generation).** This is a good question. There are simple modifications to the current setup that could handle variable length (such as padding the length and adding a EOS token). Another natural adaptation would be to run VGB for a fixed number of “generation steps”, but where each “generation step” can be a variable length (e.g. one line of text); this accommodates variable length as measured in # of tokens. Our results would give accuracy guarantees and (upper bounds) on the running time in these settings as well. It is an interesting direction for future work to adapt the running time upper bounds to incorporate the variability in the horizon in a natural way.
>
> **[Q2] (Probability of staying in current node).** Good question – any nonzero (constant) probability would be sufficient for the main theoretical guarantee. Any positive self-loop probability ensures that VGB’s Markov chain is ergodic (and hence converges to the stationary distribution eventually), and any positive constant probability lets us quantitatively upper bound the mixing time. We are happy to explain this in more detail. We also remark that our experiments set the self-loop probability to zero (providing empirical validation that the choice of this probability is not too important).

---

### Meta-Review · Area_Chair_QjDn · 2026-01-02

**Summary:**

This paper focuses on test-time algorithms that combine the generative power of language models with process verifiers that assess the quality of partial generations. It proposes a generalization to reward-guided generation of language models when the value function is not perfect. It shows that process verification errors can "cascade" in the sense of growing with the number of decoding steps. Authors proposed an alternative test-time scaling algorithm - the method (VGB) - that has a strong theoretical motivation, and where possible, formula derivations are delegated to relevant work.

Based on initial reviews:

Reviewer VY1h identifies as strengths: paper topic interesting and timely, introduction of a formal framework for LLMs with verifies with a priori significant theoretical results, the VGB algo is conceptually simple. On the other hand: the relevance of theoretical results in practice lack justification, presentation not polished, experiments limited (small scale).

Reviewer ZxSD identifies as strengths: explicit theoretical understanding, VGB is a principled modification of existing sampling strategies, benchmarking of VGB. On the other hand: real world applicability is not new and could be better explored, high computational cost, small-scale experiments.

HyCr - strengths: solid theory backed up by experimental results, simple illustrative examples, thought out collection of evaluation tasks, polynomial algorithm. On the other hand (minor), relative simple generation tasks, large and dense appendix with typos and missing assumptions, lot of sampling needed (complexity).

s7CV - Strengths: important practical issue, useful novel  theoretical machinery, discussion and presentation convincing. Empirical results support the work. Significant theoretical advances. Writing quality excellent. On the other hand, bounds provided not that good, proposed algorithm not very efficient, difficulties for understanding the algorithm.

Overall, all reviewers underlined the theoretical contribution and its usefulness. If there are some weaknesses (complexity, small-case experiments, ...), the contribution appears significant and 3 reviewers gave an accept.
I propose then acceptance.

**Reviewer Concerns:**

For Reviewer VY1h, authors provided honest answers to all questions, recognizing that the theoretical setting may not be exactly satisfied in some cases but can help to design algorithms and that the experiments are rather small scale.

For Reviewer ZxSD, authors argued that for the first time that a principled strategy based on backtracking leads to provable benefits and that their work is primarily theoretical. they answer on the computational cost and other points. The reviewer was satisfied and increased his score.

For HyCr, authors mentioned that they will correct typos. authors are on the same basis as the other reviewer. Reviewer mentioned that authors have addressed his concerns and indicated thatches an important theoretical contribution.

For s7CV, authors discussed their bound and comment on the average-case error bound, discussed the polynomial scaling and provide more intuitive explanation.

**Reviewer Scores:**

Reviewer VY1h gave a 6. It unclear if he would have increased his score since authors did not provide that much new elements but he would at least keep his score.

Reviewer ZxSD gave a 4 but he mentioned in his answer that he would have provided an 8 to reflect the strength of the theoretical contribution.

HyCr gave a 8, and he mentioned that he maintained his positive score.

s7CV gave a 8, I guess he would have kept his positive assessment.

---

### Decision · Program_Chairs · 2026-01-26

Accept (Poster)